# NIRVANA: STRUCTURED PRUNING REIMAGINED FOR LARGE LANGUAGE MODELS COMPRESSION

## ABSTRACT

Structured pruning of large language models (LLMs) offers substantial efficiency improvements by removing entire hidden units, yet current approaches often suffer from significant performance degradation, particularly in zero-shot settings, and necessitate costly recovery techniques such as supervised fine-tuning (SFT) or adapter insertion. To address these critical shortcomings, we introduce **NIRVANA**, a novel pruning method explicitly designed to balance immediate zero-shot accuracy preservation with robust fine-tuning capability. Leveraging a first-order saliency criterion derived from the Neural Tangent Kernel under Adam optimization dynamics, NIRVANA provides a theoretically grounded pruning strategy that respects essential model training behaviors. To further address the unique challenges posed by structured pruning, NIRVANA incorporates an *adaptive sparsity allocation* mechanism across layers and modules (attention vs. MLP), which adjusts pruning intensity between modules in a globally balanced manner. Additionally, to mitigate the high sensitivity of pruning decisions to calibration data quality, we propose a simple yet effective *KL divergence-based calibration data selection* strategy, ensuring more reliable and task-agnostic pruning outcomes. Comprehensive experiments conducted on Llama3, Qwen, and T5 models demonstrate that NIRVANA outperforms existing structured pruning methods under equivalent sparsity constraints, providing a theoretically sound and practical approach to LLM compression.

## 1 INTRODUCTION

Transformer-based (Vaswani et al., 2017) large language models (LLMs) have revolutionized natural language processing, achieving unprecedented performance across diverse tasks. However, this remarkable capability comes at the cost of enormous computational resources, making these models increasingly inaccessible to the broader community. For instance, a typical 7-billion-parameter model requires approximately 14GB of GPU memory at 16-bit precision, incurring prohibitive costs for training and inference. This computational barrier not only restricts widespread adoption but also impedes the democratization of AI, reinforcing misconceptions that advanced AI tools are inherently exclusive to resource-rich institutions (Ai et al., 2025a; Zou et al., 2024; Dang et al., 2025; Wei et al., 2023; Tian et al., 2025).

To alleviate this critical bottleneck, model compression techniques—particularly pruning (LeCun et al., 1989)—emerge as an essential strategy, aiming to create lighter, more accessible models without substantially compromising their effectiveness. Current pruning approaches generally fall into three categories: **(1) Unstructured pruning** methods (e.g., SparseGPT (Frantar and Alistarh, 2023), Wanda (Sun et al., 2023)[1]) achieve near-lossless zero-shot accuracy by removing individual weights (Frantar and Alistarh, 2023; Sun et al., 2023), but fail to deliver practical speedups due to irregular sparsity patterns incompatible with hardware accelerators (Cheng et al., 2024). **(2) Semi-structured pruning** (e.g., 2:4 block sparsity (Zheng et al., 2024)) addresses this limitation by enforcing fixed sparsity patterns optimized for NVIDIA sparse tensor cores (Yang et al., 2023). However, such approaches still struggle during supervised fine-tuning (SFT), as optimizer updates inevitably disrupt the predefined structures, limiting their end-to-end usability. **(3) Structured pruning** methods, such as LLM-Pruner (Ma et al., 2023) and FLAP (An et al., 2024), further improve hardware compatibility by removing entire neurons or layers, offering acceleration across

---

[1]These two can also be applied for semi-structured pruning.

both inference and training. Orthogonal to pruning, quantization further compresses models by reducing their numerical precision, presenting a similar dilemma between computational gains and performance preservation (Lin et al., 2024; Frantar et al., 2023; Xiao et al., 2023; Zhao et al., 2024).

While structured pruning holds the greatest promise for practical deployment, existing methods face several critical challenges: **(1) Inefficient recovery tuning lacking alignment with fine-tuning dynamics:** To recover from pruning-induced performance drops, existing methods rely heavily on costly fine-tuning procedures like LoRA adapters (Hu et al., 2021a) or extensive SFT (Xia et al., 2024). However, these recovery methods do not explicitly account for how pruning decisions influence the model's fine-tuning capability, resulting in inefficient resource usage and sub-optimal results. **(2) Ignoring layer- and module-specific characteristics:** Current methods typically apply pruning uniformly across all layers or modules (attention and MLP), disregarding their distinct roles within the network. This oversight often results in suboptimal pruning choices, impairing model performance. **(3) Neglect of calibration data influence:** Pruning decisions depend critically on the calibration dataset used; yet, existing approaches rarely discuss or optimize this crucial factor, leaving pruning outcomes vulnerable to suboptimal data selection.

To address these critical gaps, we introduce **NIRVANA** (**N**TK-**I**nfo**R**med adapti**V**e neuron & **A**ttentio**N** he**A**d pruning), a novel structured pruning method that tightly integrates pruning decisions with model fine-tuning dynamics through the lens of the Neural Tangent Kernel (NTK) (Jacot et al., 2018). By aligning pruning criteria with the NTK spectrum under Adam—the de facto optimizer for LLMs—NIRVANA uniquely balances immediate accuracy preservation and long-term fine-tuning adaptability. Additionally, NIRVANA employs an adaptive sparsity allocation strategy across layers and modules, complemented by a calibration data selection mechanism based on KL divergence, making pruning both theoretically grounded and practically effective. Our primary contributions are:

- **NIRVANA**: A novel NTK-guided structured pruning method explicitly designed to preserve zero-shot accuracy while maintaining fine-tuning capability, connecting pruning decisions to fundamental training dynamics.
- An **adaptive sparsity allocation** strategy that dynamically adjusts pruning ratios across layers and modules, explicitly addressing overlooked disparities in existing pruning methodologies.
- A **KL-divergence-driven calibration data selection** strategy ensuring pruning robustness by identifying optimal subsets that minimize post-pruning output discrepancies, effectively decoupling pruning quality from calibration dataset size.
- **Comprehensive experiments** on prominent LLMs (Llama3 family, Qwen and T5) demonstrating that NIRVANA significantly outperforms state-of-the-art structured pruning methods in perplexity and downstream task accuracy under similar sparsity budgets, while seamlessly integrating into standard fine-tuning pipelines without requiring additional modifications.

## 2 RELATED WORK

**Unstructured and semi-structured pruning.** Recent unstructured pruning methods, such as SparseGPT (Frantar and Alistarh, 2023) and Wanda (Sun et al., 2023), prune individual weights using local criteria, but produce irregular sparsity patterns that are inefficient for current hardware. Semi-structured methods address this by imposing fixed patterns (e.g., 2:4 sparsity (Fang et al., 2024; Zheng et al., 2024)), yet still struggle to support efficient training and require specialized hardware.

**Structured pruning.** Structured pruning removes entire neurons, attention heads, or layers, enabling practical speedups. Methods such as LLM-Pruner (Ma et al., 2023), Sheared Llama (Xia et al., 2024), and SlimGPT (Ling et al., 2024) prune components based on local pruning scores(Lee et al., 2021; Salama et al., 2019), and typically treat attention and MLP uniformly, ignoring their distinct features. Recent approaches such as Adapt-Pruner (Wang et al., 2025), FLAP (An et al., 2024), and ShortGPT (Men et al., 2024) introduce global or layer-wise pruning strategies, yet do not explicitly address the imbalance between modules. SliceGPT (Ashkboos et al., 2024) applies PCA-based transformations per block, but remains highly sensitive to calibration data, reflecting a broader limitation: most methods overlook the influence of calibration data on pruning outcomes.

**Calibration data.** Pruning methods often depend on a small *calibration* set to estimate activation or gradient statistics for scoring and pruning decisions. Recent works (Williams and Aletras, 2024) highlight that the *selection* of calibration data plays a critical role in the pruning outcome, with factors

such as data quality, diversity, and alignment with the model's pretraining distribution shown to significantly influence pruning effectiveness (Bandari et al., 2024; Ji et al., 2024). A more detailed version of related work can be found in Section A.

## 3 PRELIMINARY & PROBLEM FORMULATION

### 3.1 PRELIMINARY OF MODEL ARCHITECTURE AND NTK

We use lowercase letters to denote scalars, boldface lowercase letters to denote vectors, and boldface uppercase letters to denote matrices. The element-wise product is denoted by $\odot$. The neural network is denoted by $f$, parameterized by $\mathbf{W}$, and $\mathbf{x}$ represents the input data. See the full notation in Table 4. Since most of the current LLMs are based on SwiGLU Shazeer (2020) structure, we focus on pruning the Attention & MLP sub-layer inside a SwiGLU Transformer block. For the MLP block, it is parameterized by three weight matrices: $\mathbf{W}_{\text{gate}} \in \mathbb{R}^{d \times m}$, $\mathbf{W}_{\text{up}} \in \mathbb{R}^{d \times m}$, and $\mathbf{W}_{\text{down}} \in \mathbb{R}^{m \times d}$. For an input token $\mathbf{x} \in \mathbb{R}^d$, the MLP output is computed as:

$$\mathbf{H}(\mathbf{x}) = \Big(\sigma(\mathbf{x}\mathbf{W}_{\text{gate}}) \odot \mathbf{x}\mathbf{W}_{\text{up}}\Big)\mathbf{W}_{\text{down}},$$

where the activation function Swish $\sigma(\cdot)$ is applied elementwise, and biases are omitted for simplicity (either because modern designs often exclude them or their contribution is marginal (Dubey et al., 2024)). As for the Multi-Head Attention (MHA) block with $\mathbf{Q}_a = \mathbf{x}\mathbf{W}_a^Q$, $\mathbf{K}_a = \mathbf{x}\mathbf{W}_a^K$, $\mathbf{V}_a = \mathbf{x}\mathbf{W}_a^V$, $a = 1, \ldots, h$, where $\mathbf{W}_a^Q, \mathbf{W}_a^K, \mathbf{W}_a^V \in \mathbb{R}^{d \times d_h}$ and $d_h = d/h$, the attention for each head $a$ is $\text{head}_a = \text{softmax}\left(\frac{\mathbf{Q}_a \mathbf{K}_a^\top}{\sqrt{d_h}}\right)\mathbf{V}_a$, and the MHA output [2] is

$$\text{MHA}(\mathbf{x}) = [\text{head}_1, \ldots, \text{head}_h]\mathbf{W}^O, \quad \mathbf{W}^O \in \mathbb{R}^{h d_h \times d}.$$

Neural Tangent Kernel (NTK) (Jacot et al., 2018) provides a kernel-based framework for analyzing the training dynamics of neural networks by approximating their behavior as linear models in function space, particularly in the infinite-width limit. In our paper, we use Adam-based NTK with the form of

$$\Theta(\mathbf{x}, \mathbf{x}) = \nabla_{\mathbf{W}} f(\mathbf{x}; \mathbf{W}) \, \text{sign}\big(\nabla_{\mathbf{W}} f(\mathbf{x}; \mathbf{W})\big)^\top = \big\langle \nabla_{\mathbf{W}} f(\mathbf{x}; \mathbf{W}), \text{sign}\big(\nabla_{\mathbf{W}} f(\mathbf{x}; \mathbf{W})\big)\big\rangle.$$

See the details of the derivation in Section A.7

### 3.2 PROBLEM FORMULATION OF PRUNING PROCESS

We assume that the output $f(\mathbf{x}; \mathbf{W})$ is a single value, which is common in classification or next-token generation tasks.[3] Given a sparsity level $v$, the target of pruning can be then written in the form of:

$$\text{argmin}_{\hat{\mathbf{W}}, \mathbf{M}} \mathcal{L}\Big(f(\mathbf{x}; \mathbf{W}), \, f(\mathbf{x}; \hat{\mathbf{W}} \odot \mathbf{M})\Big),$$

where $\mathbf{M}$ is the mask matrix that has the same shape as $\mathbf{W}$. Directly solving this joint optimization over $\hat{\mathbf{W}}$ and $\mathbf{M}$ is NP-hard. Consequently, popular practices include fixing the weights (i.e., setting $\hat{\mathbf{W}} = \mathbf{W}$) and searching for $\mathbf{M}$ only (one-shot pruning (Frankle and Carbin, 2019; Frantar and Alistarh, 2023; Sun et al., 2023; Chen et al., 2021)), or selecting $\mathbf{M}$ first and then optimizing $\hat{\mathbf{W}}$, which typically requires further fine-tuning or re-training (Kwon et al., 2022; Ma et al., 2023). General pruning approaches define a *saliency score* $S_{i,j}$ for each weight, which estimates the impact of removing the connection $\mathbf{W}_{i,j}$. A general form of the saliency score is:

$$S_{i,j} = \frac{\partial \mathcal{I}}{\partial \mathbf{W}_{i,j}} \cdot \mathbf{W}_{i,j}, \tag{1}$$

where $\mathcal{I}$ is a function that measures the importance of the weight $\mathbf{W}$ to the network $f$. Once these scores are computed, the mask is obtained by selecting the top $\kappa\%$ of weights:

$$\mathbf{M}_{i,j} = \text{Top}_\kappa(S)_{i,j} = \begin{cases} 1, & \text{if } S_{i,j} \text{ is among the top } \kappa\%, \\ 0, & \text{otherwise.} \end{cases}$$

This binary mask is then applied to the weights for pruning.

---

[2]In Llama3's implementation, which employs Grouped Query Attention (GQA), multiple query heads share the same key-value (KV) head. When calculating group saliency scores, we align query (Q) and output (O) heads with their corresponding KV head groupings through index mapping.

[3]Without loss of generality, our analysis can be extended to the vector-output case.

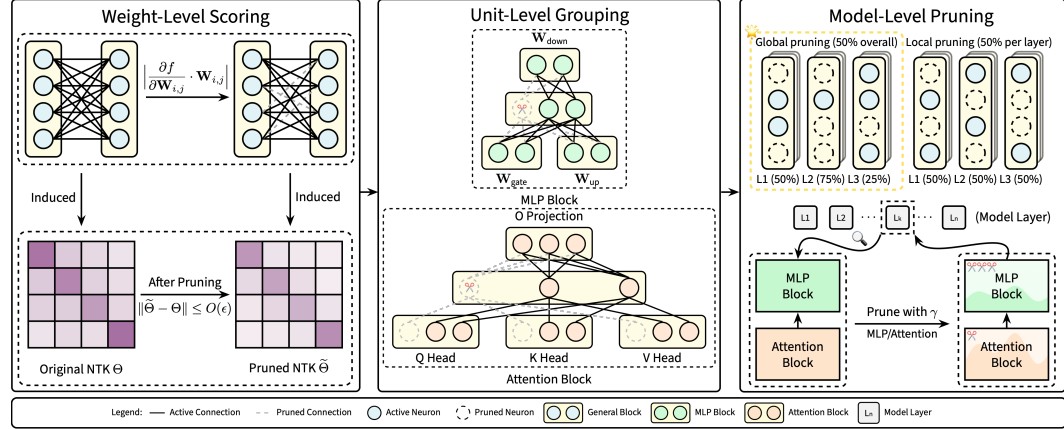

Figure 1: Illustration of our proposed **NIRVANA** framework. Left: We compute weight-level saliency scores using the NTK-guided score. Middle: We aggregate the scores into structured units. Right: We perform global pruning with an adaptive sparsity allocation strategy that adjusts pruning ratios both across layers and between MLP and attention modules with $\gamma$.

## 4 PROPOSED METHOD: NIRVANA

In this section, we present the details of NIRVANA. The corresponding pseudocode is provided in Algorithm 1, and an illustrative diagram is shown in Figure 1. Our method is built around four core components, each addressing a different level of the pruning process:

- **Weight-level saliency scoring:** quantifying the importance of individual weights based on their impact on the model's output (Section 4.1).
- **Neuron/head-level structured grouping:** aggregating weight saliencies into group-wise scores for MLP neurons and attention heads to enable structured pruning (Section 4.2).
- **Model-level adaptive sparsity allocation:** applying a global sparsity strategy and balancing pruning between MLP and attention via the adaptive ratio $\gamma$ (Section 4.3).
- **Calibration data selection:** identifying high-quality calibration data that provides the most informative gradients for reliable pruning (Section 4.4).

### 4.1 SALIENCY SCORE AND ITS CONNECTION TO NTK - WEIGHT LEVEL

We begin by deriving the saliency score, quantifying the *importance* of a weight in terms of its direct impact on the model output, i.e. $\mathcal{I} = f(\mathbf{x}; \mathbf{W})$, where $f(\mathbf{x}; \mathbf{W})$ denote the network output for input $\mathbf{x}$ and weights $\mathbf{W}$. Consider removing a single weight $\mathbf{W}_{i,j}$ by setting it to zero while keeping all other weights unchanged. We define the perturbation as: $\Delta\mathbf{W}_{i,j} = -\mathbf{W}_{i,j}$, so that the new weight matrix becomes: $\mathbf{W}' = \mathbf{W} + \Delta\mathbf{W}$, where $\Delta\mathbf{W}$ is zero everywhere except at entry $(i,j)$. The resulting change in output is: $\Delta f = f(\mathbf{x}; \mathbf{W}') - f(\mathbf{x}; \mathbf{W})$. Applying a first-order Taylor expansion around $\mathbf{W}$, we obtain: $f(\mathbf{x}; \mathbf{W} + \Delta\mathbf{W}) \approx f(\mathbf{x}; \mathbf{W}) + \frac{\partial f}{\partial\mathbf{W}_{i,j}}\Delta\mathbf{W}_{i,j}$, which is simplified to:

$$\Delta f \approx -\frac{\partial f(\mathbf{x}; \mathbf{W})}{\partial\mathbf{W}_{i,j}}\mathbf{W}_{i,j}.$$

Thus, the change in output when pruning $\mathbf{W}_{i,j}$ is approximately proportional to its gradient scaled by its magnitude. We define the saliency score as the absolute value of this first-order effect:

$$S(\mathbf{W}_{i,j}) = \left|\frac{\partial f(\mathbf{x}; \mathbf{W})}{\partial\mathbf{W}_{i,j}} \cdot \mathbf{W}_{i,j}\right|. \tag{2}$$

This score quantifies how much the output is expected to change if the weight is pruned, providing a principled criterion for pruning decisions.

**Bridging to the training dynamics via NTK.** While our saliency score in Equation (2) quantifies immediate, first-order changes to the model's outputs, it does not directly inform how pruning affects

the model's longer-term training behavior. Ideally, pruning decisions should not only minimize immediate output degradation but also preserve the model's behavior during fine-tuning. Unlike prior work that relies on empirical fine-tuning for recovery (Ma et al., 2023; Ashkboos et al., 2024), we introduce the NTK to provide theoretical insight. NTK characterizes how a model's predictions evolve under gradient-based updates; thus, if the NTK remains stable after pruning, the pruned model will preserve a similar optimization trajectory, ensuring that performance can be recovered efficiently through fine-tuning. Recall the NTK for input $\mathbf{x}$ is defined as:

$$\Theta(\mathbf{x}, \mathbf{x}) = \langle \nabla_{\mathbf{W}} f(\mathbf{x}; \mathbf{W}), \text{sign}(\nabla_{\mathbf{W}} f(\mathbf{x}; \mathbf{W})) \rangle,$$

where the gradient $\nabla_{\mathbf{W}} f(\mathbf{x}; \mathbf{W})$ reflects the sensitivity of the model output with respect to each parameter. For a given weight $\mathbf{W}_{i,j}$, its contribution to the NTK is proportional to $\left| \partial f / \partial \mathbf{W}_{i,j} \right|$. However, relying solely on sensitivity is insufficient to assess the true impact of pruning, as it ignores the current contribution of the weight itself to the output. Consider two weights: $\mathbf{W}_{i,j}$ with a large magnitude but a small gradient, and $\mathbf{W}_{i',j'}$ with a small magnitude but a large gradient. Pruning $\mathbf{W}_{i,j}$ may cause significant output distortion due to its large weight, despite its small gradient; in contrast, pruning $\mathbf{W}_{i',j'}$ would induce substantial changes in the NTK, as the gradient term dominates, even if its immediate contribution to the output is minor. This illustrates the need to balance both the parameter value and its sensitivity to avoid disrupting either the output or the training dynamics captured by the NTK. Accordingly, our saliency score incorporates both aspects, ensuring that pruning decisions simultaneously minimize output perturbations and preserve NTK stability. We formalize this with the following NTK stability bound:

**Proposition 4.1** (Short version of Theorem E.1). *For a Transformer model pruned with NIRVANA, let $\widetilde{\Theta}$ denote the NTK after pruning. For sufficiently small $\epsilon > 0$, we have:*

$$\|\widetilde{\Theta} - \Theta\| \leq O(\epsilon). \tag{3}$$

This result justifies that pruning based on the saliency score in (2) preserves both immediate model outputs (via first-order Taylor approximation) and long-term fine-tuning potential (via NTK stability). As a result, NIRVANA achieves both strong zero-shot approximation and robust fine-tuning recovery, as illustrated in Figure 1 (left).

## 4.2 STRUCTURED PRUNING VIA GROUPING - NEURON/ATTENTION HEAD LEVEL

To achieve practical efficiency through structured pruning, we group weight saliency scores by their corresponding hidden units. This allows us to prune entire units at once, directly reducing the model's computational dimensions in a strategy that aligns with dependency graph approaches in (Fang et al., 2023). In the MLP sub-layer, we treat each hidden unit $u \in \{1, \ldots, m\}$ as a group, which consists of all weights in the $u$-th column of $\mathbf{W}_{\text{gate}}$ and $\mathbf{W}_{\text{up}}$, and the $u$-th row of $\mathbf{W}_{\text{down}}$. For each hidden unit, we compute a cumulative saliency score by summing the saliency scores of its associated weights:

$$S(u) = \sum_{i=1}^{d} \left| \frac{\partial f}{\partial (\mathbf{W}_{\text{gate}})_{i,u}} \cdot (\mathbf{W}_{\text{gate}})_{i,u} \right| + \sum_{i=1}^{d} \left| \frac{\partial f}{\partial (\mathbf{W}_{\text{up}})_{i,u}} \cdot (\mathbf{W}_{\text{up}})_{i,u} \right| + \sum_{i=1}^{d} \left| \frac{\partial f}{\partial (\mathbf{W}_{\text{down}})_{u,i}} \cdot (\mathbf{W}_{\text{down}})_{u,i} \right|.$$

For MHA, we aggregate the scores for each attention head $a$, which consists of all weights in $\mathbf{W}_a^Q, \mathbf{W}_a^K, \mathbf{W}_a^V$ and the corresponding part $\mathbf{W}^O[a] \in \mathbb{R}^{d_h \times d}$:

$$S(a) = \sum_{i=1}^{d} \sum_{j=1}^{d_h} \left| \frac{\partial f}{\partial (\mathbf{W}_a^Q)_{i,j}} \cdot (\mathbf{W}_a^Q)_{i,j} \right| + \sum_{i=1}^{d} \sum_{j=1}^{d_h} \left| \frac{\partial f}{\partial (\mathbf{W}_a^K)_{i,j}} \cdot (\mathbf{W}_a^K)_{i,j} \right|$$

$$+ \sum_{i=1}^{d} \sum_{j=1}^{d_h} \left| \frac{\partial f}{\partial (\mathbf{W}_a^V)_{i,j}} \cdot (\mathbf{W}_a^V)_{i,j} \right| + \sum_{i=1}^{d_h} \sum_{j=1}^{d} \left| \frac{\partial f}{\partial (\mathbf{W}^O[a])_{i,j}} \cdot (\mathbf{W}^O[a])_{i,j} \right|$$

After computing these scores, we rank all hidden units by their aggregated saliency scores. Units falling below a global threshold (determined by our target sparsity level $v$, detailed further in Section 4.3) are pruned. Specifically, this involves zeroing out the corresponding column in $\mathbf{W}_{\text{gate}}$ and $\mathbf{W}_{\text{up}}$, the corresponding row in $\mathbf{W}_{\text{down}}$, and the entire attention head $a$. This grouping-based pruning ensures that the network's dependency structure is respected and that entire units are removed together, leading to real efficiency gains. See Figure 1 (middle) for a visual illustration.

## 4.3 Adaptive Sparsity Allocation between Attention and MLP - Model Level

Having established the unit-level saliency scores, we now address model-level pruning, focusing on how units are ranked globally and how sparsity is allocated between MLP neurons and attention heads. Most prior works (Sun et al., 2023; Frantar and Alistarh, 2023; Ma et al., 2023; Ashkboos et al., 2024; Ling et al., 2024) adopt local sparsity strategies by applying fixed pruning ratios to each layer or module, implicitly treating all units equally regardless of their functional role. However, recent studies (Wang et al., 2025; An et al., 2024) have also noted the challenge of balancing pruning across modules due to scale mismatches, though their approaches primarily rely on heuristic metric normalization. In contrast, we adopt a principled global sparsity strategy where all units across layers and modules are ranked jointly by their saliency scores, ensuring the overall sparsity target is achieved. To prevent layer collapse (Tanaka et al., 2020; Vysogorets and Kempe, 2023), we introduce a simple safeguard that retains at least one unit per layer. Furthermore, motivated by prior observations that MLP layers tend to store factual knowledge more efficiently than attention heads (Nichani et al., 2024), we introduce an explicit sparsity allocation parameter $\gamma$ to control the relative pruning rates between MLP neurons and attention heads. Unlike heuristic standardization, this parameterized allocation allows us to systematically adjust pruning aggressiveness between the two components, providing a more flexible and interpretable balance:

$$v_{\text{Attn}} = \frac{v(\#_{\text{MLP}} + \#_{\text{Attn}})}{\#_{\text{Attn}} + \gamma \cdot \#_{\text{MLP}}}, \quad v_{\text{MLP}} = \gamma \cdot v_{\text{Attn}}, \tag{4}$$

where $v$ denotes the overall target sparsity, $v_{\text{MLP}}$ and $v_{\text{Attn}}$ denote the applied sparsity to MLP neurons and attention heads, respectively, and $\#_{\text{MLP}}$ and $\#_{\text{Attn}}$ represent their total parameter counts. Details on how $\gamma$ is determined are provided in Section F. This global ranking and adaptive allocation process is illustrated in Figure 1 (right).

## 4.4 Calibration Data Selection via KL Divergence

As pruning transitions from individual weights as in Equation (2) to larger groups (neurons/heads), the variance in the aggregated saliency scores across these groups can be more pronounced compared to weight-level pruning. This makes it critical to carefully select calibration data that accurately reflects the model's behavior. Contrary to existing work (Williams and Aletras, 2024; Bandari et al., 2024; Ji et al., 2024), we empirically find that metrics like data quality, diversity, or quantity do not consistently correlate with post-pruning performance, as illustrated by examples in Section H. Therefore, we adopt a lightweight yet effective approach. We propose using the KL divergence to measure the output discrepancy between the pruned model $\hat{f}$ and the original model $f$, serving as a proxy for selecting calibration data. Minimizing KL divergence ensures calibration data induces pruning decisions that preserve the original model's output distribution, avoiding biased gradients from unrepresentative samples. We define the calibration dataset $\mathcal{C}^*$ as:

$$\mathcal{C}^* = \arg\min_{\substack{\mathcal{C} \subset \mathcal{D} \\ |\mathcal{C}| = n}} \frac{1}{n} \sum_{\mathbf{x} \in \mathcal{C}} \text{KL}\big(f(\mathbf{x}) \,\|\, \hat{f}(\mathbf{x})\big).$$

In detail, we randomly sample multiple candidate batches from the dataset and prune the model using each batch. For each pruned model, we compute the KL divergence on a fixed held-out evaluation set. The batch that yields the lowest KL divergence is the calibration data $\mathcal{C}^*$. This selection process is highly efficient, as each trial requires only a single backward pass on a small data subset, adding minimal computational overhead. The pseudocode of this process can be found in Algorithm 2.

## 5 Experiments

We begin by outlining the experimental setup, followed by comparative evaluations against several baselines on Llama3.1-8B, conclude with an ablation study. Additional experiments on Llama3.2-3B, Qwen2.5-7B, and fine-tuning on T5-base are provided in Section B.

### 5.1 Experimental Settings

Table 1: Evaluation results of Llama3.1-8B. **Bold** indicates the best results while underline indicates the second-best. We report both the zero-shot results and the performance after recovery fine-tuning. The down-arrow notation (↓) indicates that a lower metric represents better performance. The results are obtained through three runs.

| Sparsity | Method | WikiT ↓ | PTB ↓ | LambD ↓ | ARC-e | WinoG | HellaS | SVAMP | MBPP* | Avg | #Param |
|---|---|---|---|---|---|---|---|---|---|---|---|
| 0% | Llama3.1-8B | 8.50 | 14.02 | 20.09 | 81.52 | 73.56 | 78.90 | 72.67 | 48.60 | 71.05 | 8.03B |
| 20% | LLM-Pruner | 17.45 | 27.89 | 28.33 | **69.57** | 66.06 | 65.89 | 22.33 | 4.40 | 45.65 | 6.73B |
|  | SliceGPT | 21.10 | 118.79 | 252.46 | 53.70 | 61.61 | 51.40 | 0.00 | 0.00 | 33.34 | 6.57B |
|  | FLAP | 20.88 | 31.35 | 31.72 | 60.35 | 66.22 | 59.03 | 16.33 | 3.40 | 41.07 | 6.61B |
|  | NIRVANA (ours) | **13.38** | **19.77** | **26.20** | 68.52 | 67.40 | 66.75 | 49.00 | 23.80 | **55.09** | 6.62B |
| w/ tune | LLM-Pruner | 12.57 | 19.77 | 25.00 | 74.92 | **68.27** | 74.06 | 33.33 | 25.20 | 55.16 | 6.75B |
|  | SliceGPT | 49.81 | 87.25 | 81.64 | 61.53 | 61.56 | 59.91 | 0.00 | 0.60 | 36.72 | 6.58B |
|  | FLAP | 16.14 | 25.00 | 30.63 | 70.71 | 63.54 | 67.58 | 34.33 | 15.00 | 50.23 | 6.63B |
|  | NIRVANA (ours) | **12.37** | **18.58** | **25.00** | **76.98** | 66.54 | 74.36 | 56.67 | 33.00 | **61.51** | 6.64B |
| 40% | LLM-Pruner | 98.86 | 196.62 | 105.24 | 37.00 | 51.93 | 32.18 | 11.33 | 0.00 | 26.49 | 5.27B |
|  | SliceGPT | 61.63 | 421.14 | 647.20 | 35.61 | 50.99 | 32.20 | 0.00 | 0.00 | 23.76 | 5.44B |
|  | FLAP | 45.44 | 69.02 | 49.23 | 42.85 | 58.72 | 43.60 | 9.66 | 0.00 | 30.97 | 5.22B |
|  | NIRVANA (ours) | **28.33** | **38.72** | **43.19** | 45.79 | 58.33 | 45.85 | 11.67 | 2.20 | **32.77** | 5.23B |
| w/ tune | LLM-Pruner | 33.12 | 74.63 | 59.96 | 56.90 | 53.59 | 51.51 | 11.67 | 0.20 | 34.77 | 5.29B |
|  | SliceGPT | 156.76 | 224.54 | 214.26 | 48.19 | 53.83 | 43.68 | 0.00 | 0.00 | 29.14 | 5.44B |
|  | FLAP | 20.09 | 30.63 | 32.60 | 58.80 | **61.17** | 57.33 | 20.33 | 5.80 | 40.69 | 5.25B |
|  | NIRVANA (ours) | **19.17** | **27.45** | **31.64** | 62.33 | 57.93 | 58.74 | 22.33 | 11.00 | **42.47** | 5.25B |
| 50% | LLM-Pruner | 215.94 | 356.02 | 196.62 | 31.19 | 49.01 | 28.82 | 7.33 | 0.00 | 23.27 | 4.55B |
|  | SliceGPT | 93.24 | 612.76 | 870.90 | 32.37 | 49.57 | 29.70 | 0.00 | 0.00 | 22.33 | 4.57B |
|  | FLAP | 68.21 | 97.71 | 61.87 | 36.45 | 53.56 | 37.20 | 8.33 | 0.00 | 27.11 | 4.50B |
|  | NIRVANA (ours) | **48.94** | **67.95** | 70.11 | 37.54 | 54.38 | 37.72 | 9.33 | 0.20 | **27.83** | 4.51B |
| w/ tune | LLM-Pruner | 45.26 | 92.87 | 70.12 | 48.32 | 52.41 | 43.51 | 4.67 | 0.00 | 29.78 | 4.55B |
|  | SliceGPT | 198.93 | 246.61 | 248.55 | 40.19 | 52.49 | 36.57 | 0.00 | 0.00 | 25.85 | 4.59B |
|  | FLAP | 28.33 | 42.52 | 43.87 | 50.76 | **58.09** | 49.90 | 15.67 | 0.00 | 34.88 | 4.52B |
|  | NIRVANA (ours) | **25.79** | **36.94** | **42.52** | 57.49 | 56.27 | 49.91 | 23.00 | 3.40 | **38.01** | 4.53B |

* Pass@1. 3-shot.

**Models and Baselines.** We conduct our main experiments on **Llama3.1-8B**, with additional evaluations on Llama3.2-3B, Qwen2.5-7B, and T5-base. We compare NIRVANA against representative structured pruning methods: LLM-Pruner (Ma et al., 2023), SliceGPT (Ashkboos et al., 2024), and FLAP (An et al., 2024).

**Evaluation Protocol.** We assess model performance across three dimensions: (1) zero-shot perplexity on standard corpora; (2) zero-shot accuracy on commonsense reasoning and math tasks; and (3) code generation capability. All experiments use NVIDIA A100 GPUs. Please refer to Section B.1 for the complete experimental setup, including dataset details and calibration settings.

## 5.2 Results on Llama3.1-8B

Table 1 summarizes the evaluation results across various sparsity levels and benchmarks. Overall, NIRVANA consistently outperforms existing pruning baselines, with the performance gap widening notably under higher sparsity. This demonstrates the effectiveness of our approach in maintaining model capabilities while enabling aggressive compression. Baselines exhibit clear weaknesses: (1) LLM-Pruner is effective only at low sparsity, as its local approach fails to capture complex dependencies at higher compression rates. (2) SliceGPT is highly sensitive to calibration data, leading to unstable performance. (3) FLAP performs well at high sparsity (40-50%) but poorly at low sparsity (20%). In sharp contrast, NIRVANA consistently achieves SoTA results across all sparsity levels. These results validate our method's core design choices: NTK-guided scoring, adaptive sparsity allocation, and robust calibration data selection.

Additional experiments on varying model sizes (Llama3.2-3B), model families (Qwen), and model architectures (T5-base) further support our findings; see Section B for details.

## 5.3 Ablation Study

Our method includes several key components: (1) an NTK-inspired saliency score; (2) an adaptive sparsity allocation strategy that combines global sparsity ranking with the pruning ratio $\gamma$ to balance

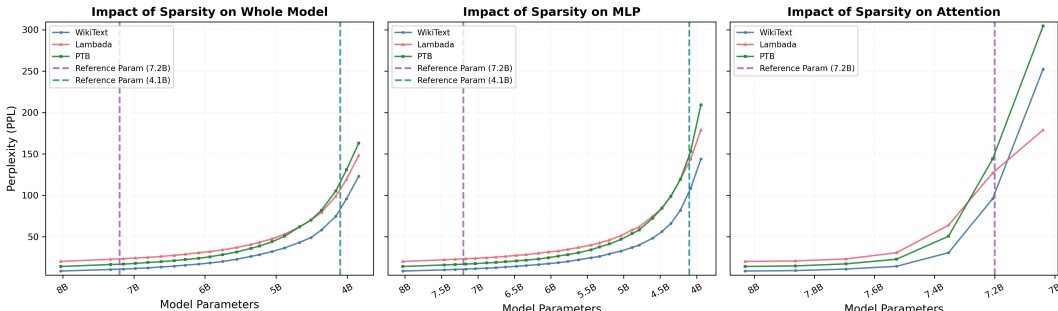

Figure 2: Comparison of pruning impact on perplexity across different pruning scopes. We evaluate pruning applied to the entire model, MLP modules only, and attention heads only. Dashed vertical lines indicate selected reference parameter points (7.2B and 4.1B) for visual comparison.

MLP units and attention heads; and (3) a calibration data selection strategy. We conduct an ablation study to assess the effects of these components, as shown in Table 2. Specifically, we compare NIRVANA with magnitude scoring, local pruning, without different ratio between MLP and attention ($\gamma = 1$), and without KL-selected calibration data. We find that using magnitude-based scores for pruning leads to extreme performance collapse, highlighting the inadequacy of naive importance metrics in the LLM pruning context. Similarly, applying local pruning instead of global pruning also leads to a performance drop, likely due to its inability to account for cross-layer importance differences.

**Impact of $\gamma$.** To better understand the impact of $\gamma$, we further investigate the impact of pruning scope by analyzing how perplexity changes as a function of the remaining model parameters. Figure 2 presents a detailed comparison where pruning is applied only to attention heads, only to MLP units, or jointly across the whole model using our proposed global strategy. We observed that performance degrades non-linearly as the model is pruned. Targeting only attention heads, which are critical for long-range dependencies, leads to a catastrophic performance collapse

Table 2: Ablation study on Llama3.1-8B with 50% sparsity on three PPL datasets.

|                  | Wikitext      | PTB           | Lambada       |
|------------------|---------------|---------------|---------------|
| NIRVANA          | 48.94         | 70.11         | 70.11         |
| Magnitude score  | $\approx 10^6$ | $\approx 10^5$ | $\approx 10^6$ |
| NTK-SAP score    | 58.12         | 72.33         | 72.33         |
| local            | 90.01         | 142.85        | 132.42        |
| w/o ratio $\gamma$ | 102.00      | 139.42        | 123.04        |
| w/o selected data | 72.33        | 115.58        | 95.82         |

even at moderate sparsity. While pruning only the more numerous MLP parameters is more stable, performance still drops sharply below a critical threshold (see the reference line of 4.1B). In contrast, our global strategy yields a significantly smoother perplexity curve and consistently outperforms both specialized approaches at any given parameter budget. This demonstrates that balancing pruning across different components is crucial for maintaining model robustness.

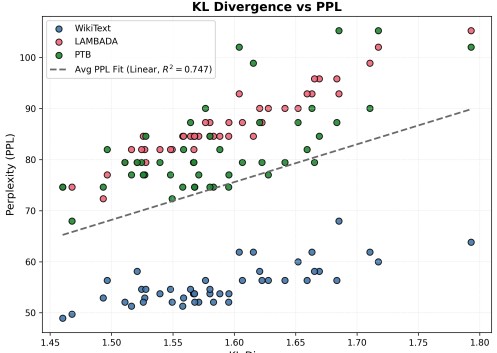

Figure 3: Relationship between KL divergence and perplexity (PPL) across different datasets over 50 runs. The observed linear correlation supports the effectiveness of using KL divergence as a proxy for data selection in pruning calibration.

**Calibration data selection.** We use calibration data from BookCorpus to maintain the zero-shot setting, with 32 examples of length 128 tokens. To identify optimal calibration data, we use the KL divergence between the pruned model and the original model as a proxy: specifically, we first prune the model using calibration data sampled with a random seed, then evaluate the KL divergence on a fixed validation set (also from BookCorpus) between the original and pruned models. We also test the pruned model on three perplexity benchmarks: WikiText, LAMBADA, and PTB, to assess whether performance correlates with the KL value. As shown in Figure 3, we observe a roughly linear relationship between KL divergence and downstream perfor-

Table 3: Post-pruning statistics during inference, tested on 20 rounds of Wikitext (batch size 32).

| Sparsity | Method | #Param | FLOPs (G) | Latency (s/batch) ↓ | Throughput (tok/s) ↑ | Mem (GB) |
|---|---|---|---|---|---|---|
| 0% | Llama3.1-8B | 8.03B | 6.98 | 0.33 | 12412.12 | 17.95 |
| 20% | LLM-Pruner | 6.73B | 5.75 | 0.41 | 9990.24 | 15.46 |
| | SliceGPT | 6.57B | 5.59 | 0.38 | 10736.57 | 15.54 |
| | FLAP | 6.61B | 5.56 | 0.50 | 8192.00 | 15.33 |
| | NIRVANA | 6.62B | 5.61 | 0.28 | 14628.57 | 15.29 |
| 40% | LLM-Pruner | 5.27B | 4.42 | 0.52 | 7876.92 | 12.52 |
| | SliceGPT | 5.44B | 4.21 | 0.31 | 13429.51 | 12.30 |
| | FLAP | 5.22B | 4.19 | 0.39 | 10502.56 | 12.73 |
| | NIRVANA | 5.23B | 4.26 | 0.23 | 17808.70 | 12.65 |
| 50% | LLM-Pruner | 4.55B | 3.76 | 0.20 | 20480.00 | 10.95 |
| | SliceGPT | 4.57B | 3.52 | 0.27 | 15340.82 | 10.53 |
| | FLAP | 4.50B | 3.52 | 0.30 | 13653.33 | 11.32 |
| | NIRVANA | 4.51B | 3.55 | 0.21 | 19504.76 | 11.27 |

mance. Based on this, we select the calibration data that yields the smallest KL value for subsequent experiments.

## 5.4 EFFICIENCY ANALYSIS

To evaluate the practical performance gains of structured pruning, we benchmark the inference efficiency of all methods on a single A100. The results are presented in Table 3. A key observation is the stark difference in how parameter reduction translates into practical speedups. While all methods successfully reduce peak memory usage, their impact on latency and throughput varies significantly. NIRVANA demonstrates a consistent and predictable trend: as sparsity increases, latency steadily decreases, and throughput improves accordingly. Baseline methods, however, exhibit erratic and non-monotonic behavior. For instance, LLM-Pruner's latency unexpectedly deteriorates at 40% sparsity, despite a 34% reduction in parameters. This suggests that merely reducing FLOPs is insufficient for guaranteed acceleration. These inconsistent speedups are caused by a hardware-software mismatch. Modern GPUs leverage specialized compute kernels (e.g., Tensor Cores) that are highly optimized for matrix dimensions that are multiples of 8 (Sarge et al., 2019; NVIDIA Corporation). When pruning creates dimensions not aligned to this constraint, the GPU must fall back to slower, general-purpose kernels, which can negate the performance gains from sparsity. See Section B.6 for the detailed analysis. Armed with this insight, we designed NIRVANA to be explicitly *hardware-aware*, which includes a dimension alignment step that ensures all remaining hidden dimensions are multiples of 8.

## 6 CONCLUSION

We introduced **NIRVANA**, a novel structured pruning approach guided by the NTK, aiming to balance zero-shot accuracy preservation with fine-tuning adaptability. By explicitly linking the pruning criterion to NTK dynamics under Adam optimization, NIRVANA ensures pruning decisions that maintain both immediate output stability and fine-tuning recovery. Furthermore, our method integrates an adaptive sparsity allocation mechanism and a KL-divergence-based calibration data selection strategy, enhancing pruning robustness and efficiency. Extensive experiments demonstrate that NIRVANA consistently outperforms existing structured pruning baselines in both perplexity and downstream task performance under equivalent sparsity.

## ETHICS STATEMENT

As a model compression method, NIRVANA primarily targets reducing the resource demands of large language models without altering their behavior or capabilities. We do not foresee direct societal risks or misuse arising uniquely from this work. However, as with any efficiency technique, care should be taken to ensure that improved accessibility to large models does not inadvertently facilitate harmful or unintended applications.

## REPRODUCIBILITY STATEMENT

To ensure our work is fully reproducible, we've provided extensive details on our experimental setup, hardware specifications, and algorithms, complete with pseudocode and illustrative figures in both the main body and in the appendix. For direct validation and easy adoption, a complete, runnable, and easy-to-integrate code implementation is available in the supplementary materials.

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

# A    RELATED WORK

While Section 2 and Section 3.2 provide a high-level overview of LLM structured pruning, this appendix offers a more comprehensive discussion. The landscape of modern AI is fundamentally shaped by the Transformer architecture (Zou et al., 2025b; Chen et al.; Zhang et al., 2024b; 2023; Cui et al., 2025; Ai et al., 2025b), whose scaling capabilities have driven unprecedented progress across different tasks (Zou et al., 2025a;c; Chen et al., 2025; He et al., 2025; Ning et al., b;a). This success, however, has also necessitated a strong focus on computational efficiency to manage the growing resource demands of large models (Zhao et al., 2023; 2025b;a; Li et al., 2025b). Against this backdrop, we now dive into a more detailed analysis of existing pruning methodologies.

## A.1    NOTATIONS

We use lowercase letters to denote scalars, boldface lowercase letters for vectors, and boldface uppercase letters for matrices. The element-wise product is denoted by $\odot$. Let $f(\mathbf{x}; \mathbf{W} \odot \mathbf{M})$ denote the neural network function, where $\mathbf{x}$ are the inputs, $\mathbf{W}$ the weights (or connections), and $\mathbf{M}$ the sparse mask matrix with sparsity $v$ (density $d = 1 - v$). Additionally, let $\mathcal{L}$ be the loss function, and let

$$D = \{(\mathbf{x}_k, \mathbf{y}_k)\}_{k=1}^N \subset \mathbb{R}^n \times \mathbb{R}^m$$

denote the dataset with $N$ data points (with $\mathbf{x}$ as the input features and $\mathbf{y}$ as the labels). Finally, let $\mathcal{A}$ be the optimizer that, given the initial weights of the network before supervised fine-tuning (SFT) $\mathbf{W}^{(0)}$, returns the weights after SFT, i.e., $\mathbf{W}^{(\text{final})} = \mathcal{A}(\mathbf{W}^{(0)})$. See the full notation in Table 4.

## A.2    POST-TRAIN PRUNING

Post-train pruning compresses a fully trained dense model by removing unimportant weights or structures. A common formulation minimizes the discrepancy between the outputs of the uncompressed and pruned layers. Given an input $\mathbf{x}$, the objective is to solve:

$$\text{argmin}_{\hat{\mathbf{W}}, \mathbf{M}} \mathcal{L}\Big(\mathbf{W}^{(\text{final})}\mathbf{x}, \ (\hat{\mathbf{W}} \odot \mathbf{M})\mathbf{x}\Big)$$

$$= \text{argmin}_{\hat{\mathbf{W}}, \mathbf{M}} \frac{1}{N} \sum_{k=1}^N \mathcal{L}\Big(\mathbf{W}^{(\text{final})}\mathbf{x}_k, \ (\hat{\mathbf{W}} \odot \mathbf{M})\mathbf{x}_k\Big),$$

where $\mathbf{M}$ is a mask matrix enforcing a fixed sparsity $v$.

Directly solving this joint optimization over $\hat{\mathbf{W}}$ and $\mathbf{M}$ is NP-hard. Consequently, popular practices include fixing the weights (i.e., setting $\hat{\mathbf{W}} = \mathbf{W}$) and searching for $\mathbf{M}$ only (one-shot pruning (Frankle and Carbin, 2019; Frantar and Alistarh, 2023; Sun et al., 2023; Chen et al., 2021)), or selecting $\mathbf{M}$ first and then optimizing $\hat{\mathbf{W}}$ (which typically requires further fine-tuning or re-training (Kwon et al., 2022; Ma et al., 2023)).

## A.3    FORESIGHT PRUNING

Foresight pruning, also known as pruning before training, seeks to identify and eliminate redundant parameters at initialization, thereby reducing both training and inference costs. For a neural network $f$ parameterized by $\mathbf{W}$ and data $(\mathbf{x}, \mathbf{y})$, the objective is formulated as:

$$\min_{\mathbf{M}} \ \mathcal{L}\Big(f\big(\mathbf{x}; \mathcal{A}(\mathbf{W}^{(0)} \odot \mathbf{M})\big), \mathbf{y}\Big)$$

$$= \min_{\mathbf{M}} \ \frac{1}{N} \sum_{k=1}^N \mathcal{L}\Big(f\big(\mathbf{x}_k; \mathcal{A}(\mathbf{W}^{(0)} \odot \mathbf{M})\big), \mathbf{y}_k\Big), \tag{5}$$

where $\mathcal{A}$ returns the final weights $\mathbf{W}^{(\text{final})}$. As in the post-train case, solving Equation (5) exactly is NP-hard because it involves joint optimization over the mask and the model parameters.

Table 4: Notation.

| Symbol | Definition and Description |
|---|---|
| $\mathbf{x}$ | Input token, $\mathbf{x} \in \mathbb{R}^d$ |
| $f(\mathbf{x}; \mathbf{W} \odot \mathbf{M})$ | Neural network function, with weights $\mathbf{W}$ and sparse mask $\mathbf{M}$ |
| $\mathbf{W}$ | Weight matrices (parameters) of the network |
| $\mathbf{M}$ | Binary mask matrix, $\mathbf{M} \in \{0,1\}^{\text{shape}(\mathbf{W})}$, indicating pruned weights |
| $\mathcal{D}$ | Dataset |
| $S$ | Saliency score function |
| $\odot$ | Element-wise (Hadamard) product |
| $\sigma(\cdot)$ | Swish activation function, applied elementwise |
| $v$ | Sparsity ratio |
| $d$ | Hidden size of the model |
| $m$ | Intermediate (FFN) dimension in the MLP |
| $h$ | Number of attention heads |
| $d_h$ | Per-head dimension, $d_h = d/h$ |
| $\mathbf{W}_a^Q, \mathbf{W}_a^K, \mathbf{W}_a^V$ | Query, Key, Value weight matrices for head $i$, each $\in \mathbb{R}^{d \times d_h}$ |
| $\mathbf{W}_a^O$ | Output weight matrices for head $i$, $\in \mathbb{R}^{d_h \times d}$ |
| $\mathbf{Q}_a, \mathbf{K}_a, \mathbf{V}_a$ | Query, Key, Value representations for head $i$ |
| $\mathbf{W}_{\text{gate}}, \mathbf{W}_{\text{up}}$ | Projection weights, $\in \mathbb{R}^{d \times m}$ |
| $\mathbf{W}_{\text{down}}$ | Down projection weight, $\in \mathbb{R}^{m \times d}$ |
| $\text{head}_a$ | Attention output for head $a$ |
| $\text{MHA}(\mathbf{x})$ | Multi-Head Attention output |
| $\mathbf{H}(\mathbf{x})$ | MLP block output |

To bridge this gap, popular approaches define a *saliency score* $S_{i,j}$ for each weight, which estimates the impact of removing the connection $\mathbf{W}_{i,j}$. A general form of the saliency score is:

$$S\big(\mathbf{W}_{i,j}^{(0)}\big) = \frac{\partial \mathcal{I}}{\partial \mathbf{W}_{i,j}^{(0)}} \cdot \mathbf{W}_{i,j}^{(0)}, \tag{6}$$

where $\mathcal{I}$ is a function that measures the importance of the weight $\mathbf{W}$ to the network $f$. Once these scores are computed, the mask is obtained by selecting the top $\kappa\%$ of weights:

$$\mathbf{M}_{i,j} = \text{Top}_\kappa(S)_{i,j} = \begin{cases} 1, & \text{if } S_{i,j} \text{ is among the top } \kappa\%, \\ 0, & \text{otherwise.} \end{cases}$$

### A.4 REVISITING SALIENCY METHODS

Several representative methods use different formulations of the saliency score to approximate weight importance. These approaches are predominantly explored in the context of computer vision tasks. In the following sections, we will also discuss the work in the LLM context.

**SNIP** (Lee et al., 2019c) proposes a data-dependent saliency:

$$S_{\text{SNIP}} = \left| \frac{\partial \mathcal{L}(\mathbf{x}; \mathbf{W})}{\partial \mathbf{W}_{i,j}} \cdot \mathbf{W}_{i,j} \right|.$$

**GraSP** (Wang et al., 2020) employs a second-order (Hessian) metric:

$$S_{\text{GraSP}} = -\left( \mathbf{H} \frac{\partial \mathcal{L}(\mathbf{x}; \mathbf{W})}{\partial \mathbf{W}_{i,j}} \right) \cdot \mathbf{W}_{i,j},$$

where $\mathbf{H}$ denotes the Hessian of the loss.

**SynFlow** (Tanaka et al., 2020) introduces a data-agnostic approach by defining saliency on constant inputs (e.g., $\mathbf{1}$) and absolute weights:

$$S_{\text{SynFlow}} = \left| \frac{\partial f(\mathbf{1}; |\mathbf{W}|)}{\partial |\mathbf{W}_{i,j}|} \right| \cdot |\mathbf{W}_{i,j}|.$$

Although SynFlow's formulation is similar to our approach, using absolute values may not fully capture the true gradient flow in the model.

**NTK-SAP** (Wang et al., 2023) adopts a data-agnostic perspective by injecting a small perturbation $\Delta \mathbf{W}_{i,j} \sim \mathcal{N}(0, \epsilon \mathbf{I})$:

$$S_{\text{NTK-SAP}} = \left| \frac{\partial \| f(\mathbf{z}; \mathbf{W}) - f(\mathbf{z}; \mathbf{W} + \Delta \mathbf{W}) \|_2^2}{\partial \mathbf{W}_{i,j}} \right|,$$

with $\mathbf{z}$ drawn from a standard normal distribution.

**PX-Pruning** (Iurada et al., 2024) introduces an auxiliary function $\mathcal{R}$ computed by two helper networks $g$ and $h$ (sharing the original architecture):

$$S_{\text{PX}} = \left| \frac{\partial \mathcal{R}(\mathbf{x}, \mathbf{W}, \mathbf{a})}{\partial (\mathbf{W}_{i,j}^2)} \cdot \mathbf{W}_{i,j}^2 \right|,$$

where

$$\mathcal{R}(\mathbf{x}, \mathbf{W}, \mathbf{a}) = g(\mathbf{x}^2, \mathbf{1}, \mathbf{a})\, h(\mathbf{1}, \mathbf{W}^2, \mathbf{1}),$$

and $\mathbf{a}$ tracks the activation status. Backpropagation through $\mathcal{R}$ yields a saliency score for each parameter.

While these foresight methods have shown promise, they are not commonly applied to LLMs due to the models' scale and behavior during fine-tuning. Moreover, the inherent nature of unstructured pruning in these methods often limits their ability to reduce training costs. In contrast, our work focuses on the MLP module in LLMs and develops an NTK-aware saliency score tailored to preserve training dynamics.

## A.5 DETAILED COMPARISON WITH NTK-SAP

- **Paradigm and Target:** NTK-SAP is an iterative, unstructured pruning method that finds sparse subnetworks at initialization. In contrast, NIRVANA is a one-shot, structured pruning method designed for post-training compression of LLMs (removing entire heads/neurons).

- **Saliency Derivation:** NTK-SAP aims to preserve the NTK spectrum; its saliency score is a finite-difference approximation of the NTK trace norm: $S(m_{ij}) = \left| \frac{\partial \mathbb{E}[\| f(z; W \odot m) - f(z; (W + \Delta W) \odot m) \|_2^2]}{\partial m_{ij}} \right|$, where $m$ represents the mask. NIRVANA's saliency score is derived from a first-order Taylor expansion to directly estimate the impact of weight removal on the model's output: $S(W_{ij}) = \left| \frac{\partial f}{\partial W_{ij}} \cdot W_{ij} \right|$, where $W$ represents the weight. We then analyze this score under an Adam-style NTK to justify that it preserves NTK stability during optimization.

- **Data and Weight Dependency:** NTK-SAP is data- and weight-agnostic (using random inputs $z$ and averaging over random initializations). NIRVANA is data- and weight-dependent, using real calibration data and specific pre-trained weights to compress a target model instance. This distinction is critical. Data- and weight-agnostic methods are blind to the specific knowledge encoded in pre-trained weights, do not leverage the specific structure of a given pre-trained checkpoint, which can make it overlook model-specific important directions. By conditioning on the actual weights, NIRVANA explicitly targets the preservation of these learned knowledges, whereas agnostic methods risk pruning essential capabilities.

## A.6 LLM-BASED PRUNING

Recent unstructured pruning methods, such as SparseGPT (Frantar and Alistarh, 2023) and Wanda (Sun et al., 2023), have proposed efficient one-shot pruning strategies that remove individual weights based on local reconstruction error or activation-aware criteria. While effective in preserving accuracy, their irregular sparsity patterns limit practical speedups on existing hardware accelerators. To bridge the gap between unstructured and structured sparsity, LLM-Barber (Su et al., 2025) proposes a block-aware rebuilder for semi-structured masks, aiming to balance flexibility and hardware efficiency.

In contrast, structured pruning methods aim to remove entire neurons, attention heads, or layers, enabling more hardware-friendly sparsity. Representative approaches include LLM-Pruner (Ma et al., 2023), Sheared Llama (Xia et al., 2024), and SlimGPT (Ling et al., 2024), which prune model components based on local importance scores. More recently, Olica (He and Lin, 2025) and SlimLLM (Guo et al., 2025) have pushed the boundaries of structured pruning by refining importance estimation metrics. However, these methods typically treat attention and MLP modules uniformly or may face challenges with the reduced redundancy in Grouped Query Attention (GQA) architectures.

Several recent works have attempted to address the imbalance across layers. For example, Adapt-Pruner (Wang et al., 2025) introduces layer-wise global scoring but still applies uniform pruning within modules. FLAP (An et al., 2024) further introduces a heuristic structure search to assign different sparsity levels across both layers and modules. Similarly, Týr-the-Pruner (Li et al., 2025a) optimizes global sparsity distribution via evolutionary search. While effective, such search-based methods can be computationally prohibitive (e.g., taking hours for a single run) compared to one-shot approaches. ShortGPT (Men et al., 2024) removes entire layers based on global layer importance, but does not differentiate between module types within layers.

SliceGPT (Ashkboos et al., 2024) instead prunes hidden dimensions rather than intermediate MLP or attention dimensions, leveraging computational invariance in RMSNorm-connected transformers. The heavy reliance on calibration data to compute these transformations makes it sensitive to the choice of calibration dataset. **Pruning with Fine-tuning.** Distinct from the aforementioned post-training pruning methods, another line of work, including LoRAP (Li et al., 2024b) and LoRAPrune (Zhang et al., 2024a), tightly couples structured compression with Low-Rank Adaptation (LoRA) specifically for fine-tuning tasks. Our work, in contrast, focuses on architecture compression for the general-purpose backbone and is orthogonal to subsequent parameter-efficient fine-tuning strategies.

In summary, while recent methods have explored global strategies or fine-tuning integration, gaps remain in effectively handling the distinct dynamics of attention heads versus MLP neurons and the critical role of calibration data selection. Our work addresses these gaps through a theoretically grounded approach that jointly considers optimization dynamics, adaptive sparsity allocation, and calibration-aware pruning.

## A.7 NEURAL TANGENT KERNEL (NTK)

**SGD Perspective.** Consider training via continuous-time gradient descent (GD) with learning rate $\eta$, where the parameter vector $\mathbf{W}_t$ evolves over time $t$. For a neural network $f(\mathbf{x}; \mathbf{W})$ with training loss $\mathcal{L}$, one can write(Lee et al., 2019a):

$$\dot{\mathbf{W}}_t = -\eta \nabla_{\mathbf{W}_t} f(\mathbf{x}; \mathbf{W}_t)^\top \nabla_{f(\mathbf{x};\mathbf{W}_t)} \mathcal{L},$$

$$\dot{\mathcal{L}} = \nabla_{f(\mathbf{x};\mathbf{W}_t)} \mathcal{L}^\top \nabla_{\mathbf{W}_t} f(\mathbf{x}; \mathbf{W}_t) \dot{\mathbf{W}}_t$$

$$= -\eta \nabla_f \mathcal{L}^\top \left[ \nabla_{\mathbf{W}} f(\mathbf{x}; \mathbf{W}_t) \nabla_{\mathbf{W}} f(\mathbf{x}; \mathbf{W}_t)^\top \right] \nabla_f \mathcal{L}.$$

The factor $\nabla_{\mathbf{W}} f(\mathbf{x}; \mathbf{W}_t) \nabla_{\mathbf{W}} f(\mathbf{x}; \mathbf{W}_t)^\top$ is called the *Neural Tangent Kernel (NTK)*(Jacot et al., 2018) under SGD, denoted

$$\widehat{\Theta}^{\mathrm{SGD}}(\mathbf{x}, \mathbf{x}) = \nabla_{\mathbf{W}} f(\mathbf{x}; \mathbf{W}) \nabla_{\mathbf{W}} f(\mathbf{x}; \mathbf{W})^\top = \langle \nabla_{\mathbf{W}} f(\mathbf{x}; \mathbf{W}), \nabla_{\mathbf{W}} f(\mathbf{x}; \mathbf{W}) \rangle.$$

**Adam (SignGD) Perspective.** Modern Transformer-based language models commonly use *Adam* rather than plain SGD. Considering Adam's exact analysis is more complicated, existing work (Li et al., 2024a; Zou et al., 2021; Kunstner et al., 2023; Wei et al., 2023) suggests that *Sign Gradient Descent* (SignGD) often behaves similarly in training dynamics. Hence, as a proxy for Adam, we consider a *sign-based* update:

$$\dot{\mathbf{W}}_t = -\eta \, \mathrm{sign}\left( \nabla_{\mathbf{W}} f(\mathbf{x}; \mathbf{W}_t)^\top \nabla_{f(\mathbf{x};\mathbf{W}_t)} \mathcal{L} \right).$$

By the chain rule,

$$\dot{\mathcal{L}} = \nabla_{f(\mathbf{x};\mathbf{W}_t)} \mathcal{L}^\top \nabla_{\mathbf{W}} f(\mathbf{x}; \mathbf{W}_t) \dot{\mathbf{W}}_t$$

$$= -\eta \nabla_f \mathcal{L}^\top \left[ \nabla_{\mathbf{W}} f(\mathbf{x}; \mathbf{W}_t) \, \mathrm{sign}\left( \nabla_{\mathbf{W}} f(\mathbf{x}; \mathbf{W}_t) \right)^\top \right] \nabla_f \mathcal{L}.$$

Following the NTK viewpoint, we define the *asymmetric SignGD kernel* as

$$\widehat{\Theta}^{\text{A-Sign}}(\mathbf{x}, \mathbf{x}) = \nabla_{\mathbf{W}} f(\mathbf{x}; \mathbf{W}) \, \text{sign}\big(\nabla_{\mathbf{W}} f(\mathbf{x}; \mathbf{W})\big)^{\top} = \big\langle \nabla_{\mathbf{W}} f(\mathbf{x}; \mathbf{W}), \text{sign}\big(\nabla_{\mathbf{W}} f(\mathbf{x}; \mathbf{W})\big)\big\rangle.$$

For simplicity, we write $\Theta = \widehat{\Theta}^{\text{A-Sign}}$ in what follows.

# B    ADDITIONAL RESULTS

## B.1    EXPERIMENTAL SETTINGS

**Model backbones.** We evaluate NIRVANA on three open-source large language models covering both decoder-only and encoder-decoder architectures: Llama3.1-8B, Llama3.2-3B (Dubey et al., 2024), Qwen2.5-7B (et al., 2025), and T5-base (Raffel et al., 2023). Our main experiments are conducted on Llama3.1-8B, with additional results for the other models provided in Section B.

**Baselines.** We compare NIRVANA with several state-of-the-art structured pruning baselines, including LLM-Pruner (Ma et al., 2023), SliceGPT (Xia et al., 2024), and FLAP (An et al., 2024). For LLM-Pruner, we use the `param_second` configuration, which we find performs better in our setting. All methods are compared at matched target model sizes for fairness.

**Evaluation tasks and datasets.** To comprehensively evaluate the effectiveness of our method, we conduct experiments on three types of tasks. First, we follow (Ma et al., 2023) to evaluate zero-shot perplexity on WikiText2 (Merity et al., 2016), PTB (Wagner et al., 2020), and Lambada (Paperno et al., 2016). Second, we evaluate zero-shot accuracy on a suite of commonsense reasoning benchmarks, including ARC-easy (Clark et al., 2018), Winogrande (Sakaguchi et al., 2021), and HellaSwag (Zellers et al., 2019), as well as SVAMP (math word problem) (Patel et al., 2021). Additionally, we assess 3-shot Pass@1 performance on MBPP (code generation) (Austin et al., 2021), all except SVAMP use the `lm-eval-harness` (Gao et al., 2024) framework. Finally, we conduct recovery tuning experiments using LoRA (Hu et al., 2021b) on Alpaca (Taori et al., 2023) following (Ma et al., 2023). All experiments are run on NVIDIA A100-SXM4-40GB GPUs.

**Calibration data.** All methods, except SliceGPT, use 32 samples from BookCorpus with a sequence length of 128 as calibration data to preserve the zero-shot setting. For SliceGPT, we follow the original setup using 128 samples from WikiText with a sequence length of 2048, as using BookCorpus leads to significant performance degradation.

**Gamma Value** The adaptive sparsity ratio, $\gamma$, is derived from a general analytic formula detailed in Section F. This formula is broadly applicable, but the resulting numerical value of $\gamma$ is inherently dependent on the specific architectural hyperparameters of each model (e.g., hidden size, number of heads, MLP intermediate dimension). While our main experiment on Llama3.1-8B yielded (which was empirically validated in Figure 4), this value is not universal. As requested, we provide the corresponding values used for the other models in our study, which were all derived from the same analytic formula by inputting each model's respective architectural parameters.

| Model | $\gamma$ |
|---|---|
| Llama3.1-8B | 3.36 |
| Llama3.2-3B | 3.02 |
| Qwen2.5-7B | 10.11 |

Table 5: Gamma values that are used in our experiments.

## B.2    ADDTIONAL ON LLAMA-3.2-3B

As shown in Table 6, similar to the main experiment in Table 1, our method consistently achieves the best performance across all sparsity levels and evaluation tasks, particularly demonstrating strong robustness in both perplexity and downstream zero-shot tasks. Compared to LLM-Pruner and SliceGPT, our method yields significantly lower perplexity and higher task accuracy under the same compression ratio, highlighting its effectiveness in preserving both language modeling capacity and generalization abilities after pruning.

Table 6: Evaluation results of Llama3.2-3B. **Bold** indicates the best results while underline indicates the second-best.

| Sparsity | Method | WikiT ↓ | PTB ↓ | LambD ↓ | ARC-e | WinoG | HellaS | MBPP* | #Param |
|---|---|---|---|---|---|---|---|---|---|
| 0% | Llama3.2-3B | 10.42 | 16.91 | 22.76 | 74.54 | 69.38 | 73.55 | 38.00 | 3.21B |
| 20% | LLM-Pruner | 24.61 | 58.12 | 37.52 | **59.81** | 58.25 | 54.55 | 3.60 | 2.70B |
| | SliceGPT | 23.85 | 144.41 | 220.20 | 46.59 | 57.85 | 44.06 | 0.00 | 3.31B |
| | NIRVANA | **17.73** | **26.20** | **33.12** | 59.64 | **59.35** | **55.40** | **16.40** | 2.65B |
| 40% | LLM-Pruner | 87.25 | 148.41 | 90.02 | 37.04 | 50.67 | 31.64 | 0.00 | 2.10B |
| | SliceGPT | 54.28 | 460.73 | 482.84 | 33.80 | 51.07 | 30.71 | 0.00 | 2.52B |
| | NIRVANA (ours) | **43.87** | **59.96** | **65.86** | **41.08** | **52.96** | **37.89** | **0.4** | 2.09B |
| 50% | LLM-Pruner | 252.46 | 367.33 | 179.02 | 29.17 | 49.17 | 26.70 | 0.00 | 1.80B |
| | SliceGPT | 83.25 | 670.36 | 691.64 | 29.63 | **50.83** | 28.72 | 0.00 | 2.15B |
| | NIRVANA (ours) | **79.44** | **108.58** | **119.25** | **34.68** | 50.59 | **33.46** | 0.00 | 1.81B |

*Pass@1. 3-shot.

Table 7: Evaluation results of Qwen2.5-7B. **Bold** indicates the best results.

| Sparsity | Method | WikiT ↓ | PTB ↓ | LambD ↓ | ARC-e | WinoG | HellaS | MBPP* | #Param |
|---|---|---|---|---|---|---|---|---|---|
| 0% | Qwen2.5-7B | 9.05 | 15.64 | 22.41 | 80.47 | 72.85 | 78.86 | 64.20 | 7.62B |
| 20% | LLM-Pruner | 17.73 | 33.12 | 34.17 | 60.44 | 54.93 | 60.30 | 12.4 | 6.27B |
| | NIRVANA (ours) | 17.73 | **29.22** | **32.10** | **68.98** | **65.69** | **69.24** | **32.6** | 6.31B |
| 40% | LLM-Pruner | **33.64** | **63.83** | **48.18** | 43.69 | 53.91 | 42.98 | 0.0 | 5.13B |
| | NIRVANA (ours) | 37.52 | 72.33 | 48.94 | **51.26** | **58.33** | **54.22** | **5.0** | 5.19B |
| 50% | LLM-Pruner | 8625.69 | 11789.92 | 8625.69 | 24.79 | 51.38 | 25.73 | 0.00 | 4.35B |
| | NIRVANA (ours) | **77.00** | **148.41** | **79.44** | **38.43** | **52.80** | **38.50** | 0.00 | 4.34B |

*Pass@1. 3-shot.

### B.3 ZERO-SHOT ON QWEN2.5-7B & QWEN2.5-14B

Table 7 shows the evaluation results on Qwen2.5-7B, and Table 8 shows the results on Qwen2.5-14B. Across different sparsity levels, our method consistently achieves the best overall performance. Notably, at the extreme sparsity level of 50%, LLM-Pruner encounters severe degradation and unstable outputs, while our approach still maintains reasonable perplexity and accuracy, demonstrating stronger robustness under high compression ratios.

### B.4 ADDITIONAL RESULTS WITH OLICA

NIRVANA significantly outperforms Olica across all metrics. Notably, we observe a performance collapse for Olica at 40% sparsity. We verified our reproduction settings and attribute this to a potential structural mismatch with Llama3.1's Grouped Query Attention (GQA). Unlike Multi-Head Attention (MHA), GQA has significantly fewer KV heads with lower redundancy. Olica's pruning metric likely over-prunes these critical KV heads at higher sparsity levels. In contrast, NIRVANA's adaptive ratio explicitly balances the head/neuron reduction, successfully preserving these sensitive components.

### B.5 FINE-TUNING ON T5

Due to the high computational cost of fine-tuning and our observation that attention heads retain more critical information than MLP neurons, we conduct experiments on T5-base (Raffel et al., 2023) by pruning only the MLP layers, while keeping the attention heads intact. All MLP layers are pruned to a global sparsity of 50%.

We compare our method with several baselines, including classic unstructured foresight pruning methods originally proposed for vision models—SNIP (Lee et al., 2019c), SynFlow (Tanaka et al., 2020), and NTK-SAP (Wang et al., 2023)—as well as LLM-specific pruning methods such as Wanda

Table 8: Evaluation results of Qwen2.5-14B. **Bold** indicates the best results.

| Sparsity | Method | WikiT ↓ | PTB ↓ | LambD ↓ | ARC-e | WinoG | HellaS | MBPP* |
|---|---|---|---|---|---|---|---|---|
| 0% | Qwen2.5-14B | 7.16 | 13.38 | 20.09 | 79.34 | 75.69 | 82.91 | 69.00 |
| 20% | LLM-Pruner | 12.18 | 21.05 | **25.39** | 70.41 | 66.46 | **73.15** | 8.20 |
| | NIRVANA (ours) | **11.99** | **19.18** | 26.20 | **73.40** | **68.19** | 72.03 | **41.60** |
| 40% | LLM-Pruner | 119.25 | 345.07 | 334.45 | 39.77 | 51.54 | 34.41 | 0.00 |
| | NIRVANA (ours) | **31.11** | **48.18** | **53.75** | **45.88** | **55.25** | **47.47** | **0.20** |
| 50% | LLM-Pruner | 3827.63 | 6310.69 | 5231.74 | 26.56 | 48.93 | 26.08 | 0.00 |
| | NIRVANA (ours) | **59.96** | **84.56** | **105.24** | **37.42** | **52.57** | **38.85** | **0.00** |

⋆ Pass@1. 3-shot.

Table 9: Evaluation results (accuracy) of t5-base on GLUE datasets. **Bold** indicates the best results while underline indicates the second-best.

| | MRPC | CoLA | SST2 | MNLI |
|---|---|---|---|---|
| t5-base | 91.42 | 83.89 | 94.84 | 86.27 |
| Magnitude | 84.80 | 71.81 | 92.43 | 83.74 |
| SNIP | 90.20 | 82.17 | **94.50** | 85.86 |
| SynFlow | 85.29 | 78.04 | 90.94 | 80.97 |
| NTK-SAP | 88.24 | 81.30 | 93.35 | 85.16 |
| Wanda | **90.69** | 82.17 | 93.81 | 85.67 |
| LLM-Pruner | 90.20 | 81.20 | 93.58 | 85.37 |
| NIRVANA (ours) | **90.69** | **82.45** | 94.27 | **85.99** |

(Sun et al., 2023) and LLM-Pruner (Ma et al., 2023). For foresight baselines, we adapt their saliency criteria within our framework to ensure a fair comparison, while for Wanda and LLM-Pruner we directly follow their official implementations, using the parameter-second variant for LLM-Pruner.

We note that the comparison here focuses exclusively on the fine-tuning setting. This is because structured pruning methods without recovery typically suffer from significant zero-shot degradation, making direct comparison with unstructured methods—which often retain better zero-shot performance—unfair. However, after sufficient supervised fine-tuning (SFT), the performance gap between structured and unstructured methods narrows considerably, allowing for a more meaningful comparison under the fine-tuning scenario. All experiments are conducted on a single V100-SXM2-32GB GPU.

Table 9 presents the fine-tuning performance of NIRVANA and baseline methods on selected GLUE tasks. We observe the following: (1) NIRVANA consistently outperforms the baseline methods. This empirically proves our theory that using NTK-based pruning criteria leads to better preservation of the model's learning dynamics compared to other baselines. This is especially impressive since NIRVANAis on par and even manages to surpass the performance of those unstructured pruning methods such as Wanda and SNIP. (2) While SynFlow considers gradients with respect to the model's output, its reliance on synthetic inputs leads to suboptimal results. Similarly, NTK-SAP approximates NTK using first-order Taylor expansion in a weight-agnostic way, weakening the performance under the pre-trained language model scenario.

## B.6 EFFICIENCY ANALYSIS

We provide the inference efficiency analysis on Llama3.1-8B in Section 5.4. Here, we provide additional experiments on T5 about the fine-tuning efficiency. We also provide experiments to support our conclusion that ensuring a dimension of multiples of 8 is critical to the model's latency and throughput.

Table 10: Comparison with the recent structured pruning method Olica on Llama3.1-8B. NIRVANA consistently outperforms Olica, particularly at higher sparsity ratios where Olica suffers from performance collapse due to the scarcity of KV heads in GQA.

| Model | Sparsity | WikiT ↓ | PTB ↓ | LambD ↓ |
|---|---|---|---|---|
| Llama3.1-8B | 0% | 8.50 | 14.02 | 20.09 |
| Olica | 20% | 20.28 | 67.95 | 27.53 |
| **NIRVANA (Ours)** | 20% | **13.38** | **19.77** | **26.20** |
| Olica | 40% | 1978.00 | 7207.03 | 1307.38 |
| **NIRVANA (Ours)** | 40% | **28.33** | **38.72** | **43.19** |

Table 11: Post-pruning statistics during inference, tested on 20 rounds of Wikitext (batch size 32).

| Model | #Params | FLOPs (G) | Latency (s/bch) ↓ | Throughput (tks/s) ↑ | Peak Memory (GB) |
|---|---|---|---|---|---|
| Llama-3.1-8B-14336 | 8.03 | 6.98 | 0.32 | 12800 | 17.95 |
| Uniform-11469(1) | 6.73 | 5.75 | 0.68 | 6024 | 15.46 |
| Uniform-11470(2) | 6.73 | 5.75 | 0.40 | 10240 | 15.46 |
| Uniform-11468(4) | 6.73 | 5.75 | 0.39 | 10503 | 15.46 |
| Uniform-11464(8) | 6.73 | 5.75 | 0.29 | 14124 | 15.46 |
| Uniform-11472(16) | 6.73 | 5.75 | 0.29 | 14124 | 15.46 |
| Uniform-11488(32) | 6.73 | 5.75 | 0.29 | 14124 | 15.46 |
| Layer-wise(8s) | 6.73 | 5.75 | 0.28 | 14629 | 15.45 |
| Attention-only | 6.73 | 6.15 | 0.27 | 15170 | 14.70 |
| MLP-only | 6.73 | 5.68 | 0.28 | 14629 | 15.54 |

**Fine-tuning Efficiency.** Table 12 shows the impact of pruning on computational cost on a single V100. NIRVANA reduces memory usage by 19.2% and achieves a 1.3× speedup in fine-tuning, demonstrating its practical advantage over dense models. In contrast, although SNIP utilizes a similar saliency score, it applies unstructured pruning, which is not easily exploitable by standard hardware or deep learning libraries for runtime acceleration. As a result, SNIP's fine-tuning time and memory usage remain nearly identical to the dense model.

Table 12: Post-pruning statistics during SFT, tested on one round of SST2 (batch size 16).

| | # Param | Time (s) | Mem (GB) |
|---|---|---|---|
| t5-base | 223.50M | 927.91 | 6.78 |
| SNIP | 166.87M | 928.63 | 6.79 |
| NIRVANA | 166.87M | 732.80 | 5.48 |

**Sanity-check Experiments on Dimension.** To validate our conclusion in Section 5.4, we focused on the MLP component (attention heads are pruned in groups and remain multiples of 8) with the same sparsity level. Specifically, we evaluated:

- Uniform-11468: MLP intermediate =11,468 (not divisible by 8)
- Uniform-11472: MLP intermediate =11,472 (divisible by 8)
- Layer-wise: MLP dims vary per layer (divisible by 8)

To further support this explanation, we additionally conduct a controlled MLP-dimension sweep. Specifically, we fix the pruning pattern to remove only one attention head, and then vary the MLP hidden size around the 20% case (11468) as follows: 11469 (odd), 11470 (multiple of 2 only), 11468 (multiple of 4 only), 11464 (multiple of 8), 11472 (multiple of 16), and 11488 (multiple of 32). Note that, due to hardware conditions, the absolute latency numbers in the updated table are slightly different from those in the original submission, but the relative trends remain consistent.

We also isolated pruning effects on each subcomponent by creating:

- Attention-only: pruning concentrated on attention heads (retaining only one head, adjusting MLP to match total params)
- MLP-only: pruning concentrated on the MLP layers (only prune MLPs)

These studies in Table 11 yield several key insights into the practical efficiency of structured pruning. First, the specific distribution of sparsity, whether applied uniformly across layers or on a layer-wise

Table 13: Performance of Llama3.1-8B with NIRVANAand SmoothQuant INT8 quantization. Lower perplexity (PPL) is better (↓). Higher throughput is better (↑).

| Method | WikiT ↓ | PTB ↓ | LambD ↓ | Latency (s/batch) ↓ | Throughput (tok/s) ↑ | Peak Memory (GB) ↓ |
|---|---|---|---|---|---|---|
| Llama3.1-8B (Base) | 8.50 | 14.02 | 20.09 | 0.35 | 11702.86 | 17.95 |
| Llama3.1-8B.int8 | 8.64 | 14.24 | 20.40 | 0.44 | 9309.09 | 11.50 |
| NIRVANA-0.2 | 13.38 | 19.77 | 26.20 | 0.49 | 8359.18 | 15.29 |
| NIRVANA-0.2.int8 | 13.59 | 20.09 | 26.60 | 1.15 | 3561.74 | 10.15 |
| NIRVANA-0.4 | 28.33 | 38.72 | 43.19 | 0.40 | 10240.00 | 12.65 |
| NIRVANA-0.4.int8 | 28.77 | 39.33 | 43.87 | 0.96 | 4266.67 | 8.79 |
| NIRVANA-0.5 | 48.94 | 70.11 | 70.11 | 0.36 | 11377.78 | 11.27 |
| NIRVANA-0.5.int8 | 49.71 | 70.11 | 70.11 | 0.75 | 5461.33 | 8.08 |

basis, exerts a negligible influence on inference latency. More critically, our results reveal that hardware-aware dimension alignment profoundly impacts runtime performance. For example, the Uniform-11468 model, with dimensions not divisible by 8, incurs a substantial latency penalty compared to its aligned Uniform-11472 counterpart, despite both models possessing an identical theoretical computational cost (FLOPs). Furthermore, we observe a trade-off based on the pruning target: applying sparsity to MLP layers yields more significant latency reductions, while pruning attention components results in a more pronounced decrease in the memory footprint. Collectively, these findings underscore the principle that dimension regularization is essential for translating theoretical FLOP reductions into tangible inference acceleration. This insight provides a clear directive for developing pruning methods that are not only accurate but also optimized for efficient, real-world deployment on modern hardware.

## C  SYNERGY WITH QUANTIZATION

This section explores the relationship between our structured pruning method, NIRVANA, and quantization (Xiao et al., 2023; Lin et al., 2024; Frantar et al., 2023), another prominent model compression technique. We demonstrate that they are *orthogonal* and can be combined for compounded efficiency gains.

### C.1  PRUNING VS. QUANTIZATION

Structured pruning and quantization enhance model efficiency through distinct mechanisms:

Structured Pruning modifies the model's architecture by systematically removing entire components like neurons or attention heads. This directly reduces the model's parameter count and, more importantly, the floating-point operations (FLOPs) required for both inference and training.

Quantization operates on the data type level, reducing the numerical precision of weights and/or activations (e.g., from 16-bit floating-point, FP16, to 8-bit integer, INT8). This primarily decreases the model's memory footprint and the memory bandwidth needed during inference.

### C.2  EXPERIMENTAL RESULTS

Given that these two methods are orthogonal, they can be effectively combined. To validate this, we conducted experiments applying post-training INT8 quantization using SmoothQuant (Xiao et al., 2023) to our NIRVANA-pruned models. The results in Table 1 confirm that our pruning approach is fully compatible with quantization, enabling users to leverage the benefits of both strategies simultaneously.

### C.3  DISCUSSION ON COMBINED EFFICIENCY

A key consideration for both pruning and quantization is their non-linear impact on performance. Pushing either technique to its extreme—very high sparsity for pruning or very low bit-depth for quantization—inevitably leads to a sharp decline in model accuracy. For instance, recent studies show that while 8-bit quantization is relatively stable, moving to 4-bit can cause severe performance degradation for methods like SmoothQuant (Huang et al., 2024).

This suggests that the most practical approach is not to maximize one method alone but to combine them in their respective "sweet spots." Our results support this strategy: moderate pruning with NIRVANA combined with stable INT8 quantization delivers superior all-around efficiency. This avoids the "performance cliff" associated with applying either method too aggressively in isolation.

Furthermore, a critical advantage of structured pruning over post-hoc quantization is its impact on training costs. Quantization is almost exclusively an inference-time optimization. In contrast, pruning creates an architecturally smaller model that is not only faster for inference but is also significantly cheaper to fine-tune or subject to recovery training. This unique ability to reduce the computational cost of the entire training lifecycle is a benefit that quantization alone cannot provide.

# D  PSEUDOCODE OF NIRVANA

## D.1  PSEUDOCODE

Algorithm 1 presents the pseudocode for NIRVANA. The algorithm operates on both MLP neurons and attention heads, taking as input the model weights, target overall sparsity $v$, and a set of calibration data $\mathcal{D}$. First, it computes NTK-guided saliency scores for all structured units (i.e., MLP neurons and attention heads) using gradients obtained from the calibration data. Then, it applies an adaptive sparsity allocation strategy, where the pruning rates for MLP and attention are balanced according to the ratio $\gamma$. Finally, pruning is performed via global ranking within each module type, and the corresponding weights are zeroed out. This procedure ensures that pruning decisions simultaneously consider both model-level sensitivity and module-specific characteristics. Algorithm 2 outlines the procedure for selecting calibration data using our KL-divergence-based strategy. Given a full candidate dataset $\mathcal{D}$, the algorithm samples multiple candidate batches, prunes the model using each batch, and evaluates the KL divergence between the pruned and original model outputs on a fixed evaluation set $\mathcal{V}$. The batch that minimizes the average KL divergence is selected as the final calibration set $\mathcal{C}^*$. This approach ensures that the selected data leads to minimal output distribution shift after pruning, providing a simple yet effective proxy for calibration data quality.

---

**Algorithm 1** NIRVANA: NTK-guided Global Structured Pruning with Adaptive Sparsity Allocation

---

1: **Input:** Model $f$, weights $\mathbf{W}$, target sparsity $v$, pruning data $\mathcal{D}$, MLP/Attention unit sets $\mathcal{U}_{\mathrm{MLP}}, \mathcal{U}_{\mathrm{Attn}}$, ratio $\gamma$
2: **Output:** Pruned model $f'$

3: $G \leftarrow \mathrm{compute\_NTK\_gradients}(f, \mathcal{D})$
4: **/\* Compute saliency scores for all units \*/**
5: **for all** MLP unit $u \in \mathcal{U}_{\mathrm{MLP}}$ **do**
6:     $S_{\mathrm{MLP}}(u) \leftarrow \mathrm{aggregate\_saliency}(G, \mathbf{W}(u))$
7: **end for**
8: **for all** Attention head $h \in \mathcal{U}_{\mathrm{Attn}}$ **do**
9:     $S_{\mathrm{Attn}}(h) \leftarrow \mathrm{aggregate\_saliency}(G, \mathbf{W}(h))$
10: **end for**

11: **/\* Adaptive sparsity allocation using $\gamma$ \*/**
12: Compute $v_{\mathrm{Attn}}, v_{\mathrm{MLP}}$ using Eq.(6)
13: **/\* Global ranking and pruning \*/**
14: $\mathcal{U}_{\mathrm{pruned,MLP}} \leftarrow \mathrm{select\_lowest}(S_{\mathrm{MLP}}, v_{\mathrm{MLP}})$
15: $\mathcal{U}_{\mathrm{pruned,Attn}} \leftarrow \mathrm{select\_lowest}(S_{\mathrm{Attn}}, v_{\mathrm{Attn}})$
16: **/\* Apply pruning masks \*/**
17: Zero out weights for $\mathcal{U}_{\mathrm{pruned,MLP}}$ and $\mathcal{U}_{\mathrm{pruned,Attn}}$

18: **return** pruned model $f'$

---

---

**Algorithm 2** KL-based Calibration Data Selection

---

1: **Input:** Full dataset $\mathcal{D}$, batch size $B$, number of trials $T$, model $f$, pruning method Prune$(\cdot)$, fixed evaluation set $\mathcal{V}$.
2: **Output:** Best calibration set $\mathcal{C}^*$.

3: KL_min $\leftarrow +\infty$
4: **for** $t = 1$ **to** $T$ **do**
5:    $\mathcal{C}_t \leftarrow$ sample_batch$(\mathcal{D}, B)$
6:    $\hat{f}_t \leftarrow$ Prune$(f, \mathcal{C}_t)$
7:    KL_value $\leftarrow 0$
8:    **for** each $\mathbf{x}$ in $\mathcal{V}$ **do**
9:       $p \leftarrow f(\mathbf{x})$
10:      $q \leftarrow \hat{f}_t(\mathbf{x})$
11:      KL_value $+= \sum_{l=1}^{K} p_l \log \frac{p_l}{q_l}$
12:    **end for**
13:    KL_value $\leftarrow$ KL_value$/|\mathcal{V}|$
14:    **if** KL_value $<$ KL_min **then**
15:      KL_min $\leftarrow$ KL_value
16:      $\mathcal{C}^* \leftarrow \mathcal{C}_t$
17:    **end if**
18: **end for**
19: **return** the final calibration dataset $\mathcal{C}^*$

---

## D.2 Computational Overhead

### D.2.1 Computational cost of the saliency score calculation

### D.2.2 Computational cost of the KL-based Selection

We measured the actual wall-clock time for the KL-based selection process on a single A100 GPU. As shown below, the cost is minimal:

| Model | Wallclock time |
|---|---|
| Llama3.1-8B | 4.12s |
| Qwen2.5-14B | 6.28s |

Table 14: KL-based selection wallclock time measured on a single A100.

- Efficiency by Design: This low latency is achieved because we use a small size of calibration sets. As discussed in Appendix G, neither the number of examples nor sequence length dominates pruning quality. Therefore, we fix the candidate calibration set to 32 samples (seq len 128) and the evaluation "anchor set" to 128 samples (seq len 64).

- Total Computational Overhead: The process consists of two steps:

    - Pre-computation (Once): Computing dense model logits on the anchor set takes 3.04s for Llama3.1-8B and 3.93s for Qwen2.5-14B on a single A100. This is cached and reused.

    - Selection Loop (Per Candidate): For each candidate, computing the pruned model's logits and the KL divergence takes 4.12s (Llama3.1-8B) and 6.28s (Qwen2.5-14B).

Combined with the fast saliency computation (approx. 2s), a complete search over 50 candidate sets for Llama3.1-8B takes less than 10 minutes on a single A100. This minimal one-time cost yields significant performance gains, making it a highly efficient trade-off.

# E  PROOF OF PROPOSITION 4.1

**Proposition E.1** (Full version of Theorem 4.1). *Consider a Transformer model $f$. Let $\Theta$ denote the Neural Tangent Kernel (NTK) w.r.t. $\mathbf{W}$. Suppose we prune the model according to the saliency measure*

$$S(\mathbf{W}_{i,j}) = \left| \frac{\partial f(\mathbf{X}; \mathbf{W})}{\partial \mathbf{W}_{i,j}} \cdot \mathbf{W}_{i,j} \right|.$$

*Denote the pruned parameter set as $\widetilde{\mathbf{W}}$ (i.e. we zero entire hidden columns/rows whose group saliency is below a threshold) and let $\widetilde{\Theta}$ be the resulting NTK. Then for a sufficiently small $\epsilon > 0$, if we prune only low-saliency units so that the total pruned saliency is $\epsilon$, we obtain*

$$\left| \Theta - \widetilde{\Theta} \right| \leq O(\epsilon).$$

*In other words, the NTK of the pruned MLP remains linearly within $\epsilon$ of the original.*

*Proof.* Recall $\Theta(\mathbf{X}, \mathbf{Z}) = \langle \nabla_{\mathbf{W}} f(\mathbf{X}), \text{sign}(\nabla_{\mathbf{W}} f(\mathbf{Z})) \rangle$. Thus

$$
\begin{aligned}
\left| \widetilde{\Theta} - \Theta \right| &= \left| \langle \nabla_{\widetilde{\mathbf{W}}} f(\mathbf{X}), \text{sign}(\nabla_{\widetilde{\mathbf{W}}} f(\mathbf{Z})) \rangle - \langle \nabla_{\mathbf{W}} f(\mathbf{X}), \text{sign}(\nabla_{\mathbf{W}} f(\mathbf{Z})) \rangle \right| \\
&\leq \left| \langle \left( \nabla_{\widetilde{\mathbf{W}}} f(\mathbf{X}) - \nabla_{\mathbf{W}} f(\mathbf{X}) \right), \text{sign}\left( \nabla_{\widetilde{\mathbf{W}}} f(\mathbf{Z}) \right) \rangle \right| + \left| \langle \nabla_{\mathbf{W}} f(\mathbf{X}), \left( \text{sign}(\nabla_{\widetilde{\mathbf{W}}} f(\mathbf{Z})) - \text{sign}(\nabla_{\mathbf{W}} f(\mathbf{Z})) \right) \rangle \right| \\
&\leq \underbrace{\|\nabla_{\widetilde{\mathbf{W}}} f(\mathbf{X}) - \nabla_{\mathbf{W}} f(\mathbf{X})\| \cdot \|\text{sign}(\nabla_{\widetilde{\mathbf{W}}} f(\mathbf{Z}))\|}_{\text{Term (i)}} + \underbrace{\|\nabla_{\mathbf{W}} f(\mathbf{X})\| \cdot \|\text{sign}(\nabla_{\widetilde{\mathbf{W}}} f(\mathbf{Z})) - \text{sign}(\nabla_{\mathbf{W}} f(\mathbf{Z}))\|}_{\text{Term (ii)}}
\end{aligned}
$$

**Term (i).** Note that $\text{sign}(\cdot)$ is a vector of $\pm 1$ in each coordinate (ignoring coordinates exactly at zero,) hence its Euclidean norm is at most $\sqrt{2dm}$.

$$
\begin{aligned}
(i) &= \|\nabla_{\widetilde{\mathbf{W}}} f(\mathbf{x}) - \nabla_{\mathbf{W}} f(\mathbf{x})\| \cdot \|\text{sign}(\nabla_{\widetilde{\mathbf{W}}} f(\mathbf{Z}))\| \\
&\leq \sqrt{2dm} \, \|\nabla_{\widetilde{\mathbf{W}}} f(\mathbf{x}) - \nabla_{\mathbf{W}} f(\mathbf{x})\| \\
&= \sqrt{2dm} \sum_{(i,j) \in \mathcal{P}} |\nabla_{\mathbf{W}_{i,j}} f(\mathbf{x})|
\end{aligned}
$$

Here, since we prune the parameters according to the salience score $S$, we assume that for the parameters that are pruned in the set $\mathcal{P}$, we have

$$\sum_{(i,j) \in \mathcal{P}} |\nabla_{\mathbf{W}_{i,j}} f(\mathbf{x})| \cdot |\mathbf{W}_{i,j}| \leq \epsilon$$

Assuming weights are bounded $|\mathbf{W}_{i,j}| \geq c$ for pruned parameters since the model is pre-trained, we then have

$$\sum_{(i,j) \in \mathcal{P}} |\nabla_{\mathbf{W}_{i,j}} f(\mathbf{x})| \leq \frac{\epsilon}{c}$$

Hence

$$(i) \leq \frac{\sqrt{2dm}}{c} \epsilon$$

**Term (ii).** Similarly, $\left\| \text{sign}(\nabla_{\widetilde{\mathbf{W}}} f(Z)) - \text{sign}(\nabla_{\mathbf{W}} f(Z)) \right\| = \left\| \text{sign}(\nabla_{\mathbf{W}_{\mathcal{P}}} f(Z)) \right\| \leq \sqrt{2dm}$. As for $\|\nabla_{\mathbf{W}} f(\mathbf{x})\|$, since the activation function $\sigma$ (usually ReLU) is 1-Lipschitz, and $\mathbf{W}$ is under-controlled with weight-decay in Adam, it is safe to assume $\|\nabla_{\mathbf{W}} f(\mathbf{x})\| \leq G$ for a small constant $G$. Consequently,

$$(ii) \leq G\sqrt{2dm}.$$

**Combine Term (i) & (ii).**

$$\left|\widetilde{\Theta} - \Theta\right| \leq \text{Term (i)} + \text{Term (ii)}$$
$$\leq \sqrt{2dm}(\frac{\epsilon}{c} + G)$$
$$= O(\epsilon)$$

$\diamond$

In additional to this theoretical proof, we provide the results of the Centered Kernel Alignment (CKA) (Kornblith et al., 2019) to empirically support Proposition 4.1. The results demonstrate that NIRVANA achieves the highest NTK alignment among all baselines. This provides empirical evidence that our NTK-driven saliency score effectively preserves the NTK structure during pruning, which is consistent with the state-of-the-art recovery fine-tuning performance shown in Table 1.

| Method | Centered Kernel Alignment |
|---|---|
| Magnitude | 0.9658 |
| LLM-Pruner | 0.9722 |
| FLAP | 0.9924 |
| **NIRVANA** | **0.9972** |

Table 15: CKA results of the pruned model and the dense model.

# F    MEASURE THE INFLUENCE OF REMOVING ONE ATTENTION HEAD AND ONE MLP NEURON

Let's start this with a toy example. For a simple one-layer Transformer model which contains one Attention block and one MLP block, lets make the following assumptions/approximations.

$$
\begin{array}{ll}
L & \text{sequence length (tokens)} \\
d & \text{hidden size (e.g. 4096)} \\
h & \text{number of heads (e.g. 32)} \\
d_h = d/h & \text{per-head size (e.g. 128)} \\
X \in \mathbb{R}^{L \times d} & \text{layer input} \\
\sigma_Q^2, \sigma_K^2, \sigma_V^2 & \text{variances of the head's } Q, K, V \text{ activations} \\
\sigma_W^2 & \text{variance of linear-layer weights} \\
\sigma_\phi^2 & \text{variance of FFN activation}
\end{array}
$$

## F.1    ATTENTION

WLOG, we use MHA to derive the form, which can be adapted to GQA. For the $Q_i = X\,W_i^Q \in \mathbb{R}^{L \times d_h}$, we write $A_i = \mathrm{softmax}\big(Q_i K_i^T / \sqrt{d_h}\big) \in \mathbb{R}^{L \times L}$, then $H_i = A_i V_i \in \mathbb{R}^{L \times d_h}.Y_i = H_i\,W_i^O \in \mathbb{R}^{L \times d}$. The output of one attention head $i$ will then be $\Delta_i(X) = Y_i = A_i V_i W_i^O \in \mathbb{R}^{L \times d}$. To define the influence of removing that head, we further define $I_i^{\mathrm{attn}}(X) = \big\|\Delta_i(X)\big\|_F^2 = \big\|A_i V_i W_i^O\big\|_F^2$. Now we want to compute $\mathbb{E}\big[I_i^{\mathrm{attn}}(X)\big] = \mathbb{E}\big[\|\Delta_i(X)\|_F^2\big]$.

$$
\begin{aligned}
\mathbb{E}\big[\|\Delta_i(X)\|_F^2\big] &= \mathbb{E}\big[\|H_i\,W_i^O\|_F^2\big] \\
&= \mathbb{E}\Big[\sum_{t=1}^{L} \|\,H_i[t]W_i^O\|_2^2\Big] \\
&= \sum_{t=1}^{L} \mathbb{E}\big[\|\,H_i[t]W_i^O\|_2^2\big] \text{ (token index } t) \\
&= \sum_{t=1}^{L} \mathbb{E}\left[\sum_{j=1}^{d}(H_i[t]W_i^O[:,j])^2\right] \\
&\overset{(1)}{=} d\sigma_W^2 \sum_{t=1}^{L} \mathbb{E}\big[\|H_i[t]\|_2^2\big] \\
&\overset{(2)}{=} sLdd_h\sigma_W^2\,\sigma_V^2
\end{aligned}
$$

(1) Conditioning on $H_i[t]$ and using the law of total expectation:

$$
\mathbb{E}\big[(H_i[t]W_i^O[:,j])^2 \mid H_i[t]\big] = \sigma_W^2\|H_i[t]\|_2^2
$$

Summing over $d$ dimensions and removing the conditioning:

$$
\mathbb{E}\left[\sum_{j=1}^{d}(H_i[t]W_i^O[:,j])^2\right] = d\sigma_W^2\,\mathbb{E}\big[\|H_i[t]\|_2^2\big]
$$

(2) For token position $t$ with attention weights $\alpha_{ts} = A_i[t,s]$:

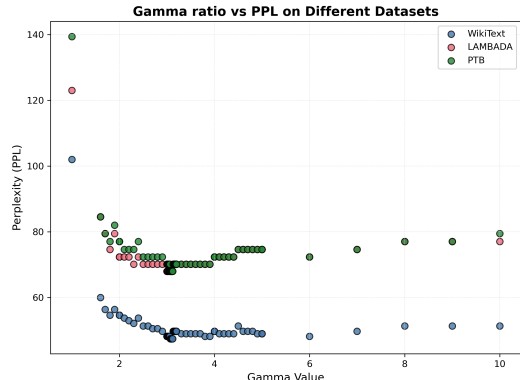

Figure 4: Empirical validation of the sparsity allocation ratio $\gamma$. The optimal value obtained via grid search closely matches the analytically derived ratio, confirming its practical effectiveness.

$$H_i[t] = \sum_{s=1}^{L} \alpha_{ts} V_i[s],$$

$$\mathbb{E}\left[\|H_i[t]\|_2^2\right] = d_h \mathbb{E}\left[\,[H_i[t]]_a^2\right]$$

$$= d_h \sum_s \alpha_{ts}^2 \underbrace{\mathbb{E}[V_i[s]_a^2]}_{\sigma_V^2}$$

$$= d_h \sigma_V^2 \sum_s \alpha_{ts}^2$$

$$= sLd_h\sigma_V^2,$$

where we obtain the last line because the rows $A_i[t]$ of the attention weight $A_i = \text{softmax}\left(Q_i K_i^\top / \sqrt{d_h}\right) = s \in [1/L, 1]$.

## F.2 MLP

$$\mathbb{E}\left[\|\Delta Y_{\text{neuron}}\|_F^2\right] = \mathbb{E}\left[\|\phi(XW_1[i])W_2[:,i]\|_F^2\right]$$
$$= L\,d\,\sigma_\phi^2\,\sigma_W^2$$

$$\Delta Y_{\text{neuron}} = \phi(XW^{\text{in}})W^{\text{out}}$$

Under similar Gaussian initialization assumptions ($W^{\text{in}} \sim \mathcal{N}(0, \sigma_W^2 \mathbf{I})$, $W^{\text{out}} \sim \mathcal{N}(0, \sigma_W^2 \mathbf{I})$):

$$\boxed{\mathbb{E}\left[\|\Delta Y_{\text{neuron}}\|_F^2\right] = Ld\sigma_\phi^2\sigma_W^2}$$

## F.3 Plugging in Model Details

We now instantiate our analytic ratio for a specific model.. Considering modern model's architecture employs Grouped Query Attention (GQA) with kv_groups = $h_{kv}$, the expected output ratio between an attention head and an MLP neuron can be written as:

$$\frac{I_{\text{attn}}}{I_{\text{mlp}}} = \frac{\mathbb{E}[Y_{\text{head}}]}{\mathbb{E}[Y_{\text{neuron}}]} = \frac{h_{kv}sLdd_h\sigma_W^2\,\sigma_V^2}{L\,d\,\sigma_\phi^2\,\sigma_W^2} = \frac{h_{kv}sd_h\sigma_V^2}{\sigma_\phi^2}.$$

Additionally, accounting for the parameter difference between an attention head and an MLP neuron, the adjusted pruning ratio $\gamma$ becomes:

$$\gamma = \frac{\frac{I_{\text{attn}}}{\#_{\text{attn}}}}{\frac{I_{\text{mlp}}}{\#_{\text{mlp}}}} = \frac{I_{\text{attn}} \cdot \#_{\text{mlp}}}{I_{\text{mlp}} \cdot \#_{\text{attn}}}.$$

Plugging in the model-specific values for Llama3.1-8B, we obtain $\gamma \approx 3.36$, which is used as the default setting in our method.

### F.4 EMPIRICAL VALIDATION

To verify the effectiveness of the pruning ratio $\gamma$, we conduct empirical experiments with varying $\gamma$ values, as shown in Figure 4. The best-performing ratio identified through grid search is 3.0, which aligns closely with our analytically derived value of 3.36. This consistency demonstrates that the analytically obtained ratio is well-justified and effective in practice.

# G FORMAL JUSTIFICATION OF THE KL-BASED CALIBRATION DATA SELECTION PROCESS

## G.1 THEORETICAL MOTIVATION: MONTE CARLO APPROXIMATION

The goal of pruning can be seen as to find a subnetwork $Q$ that minimizes the expected prediction error relative to the dense teacher $P$ over the true data distribution $\mathcal{D}$: $\min \mathbb{E}_{x \sim \mathcal{D}}[H(P(\cdot|x), Q(\cdot|x))]$.

By the information-theoretic identity $H(P, Q) = H(P) + D_{KL}(P\|Q)$, minimizing the cross-entropy (perplexity) is mathematically equivalent to minimizing the KL divergence. Since the true distribution $\mathcal{D}$ is intractable, we formally treat our anchor set $\mathcal{V}$ as a Monte Carlo estimator (Amini et al., 2025). The average KL divergence on $\mathcal{V}$ is an unbiased estimator of the true expected divergence:

$$\mathcal{L}_{proxy}(\mathcal{V}) = \frac{1}{|\mathcal{V}|} \sum_{x \in \mathcal{V}} D_{KL}(P(\cdot|x)\|Q(\cdot|x)) \approx \mathbb{E}_{x \sim \mathcal{D}}[D_{KL}(P\|Q)]$$

By selecting the calibration set $\mathcal{C}^*$ that minimizes this proxy, we are statistically maximizing the likelihood that the pruned model behaves like the teacher on the general distribution.

## G.2 EMPIRICAL EVIDENCE & ROBUSTNESS

We validate this estimator's effectiveness and stability in two ways:

1. **Correlation (Figure 3)**: We show a strong correlation between our proxy (KL on $\mathcal{V}$) and final downstream performance (PPL), confirming that the estimator is effective.

2. Stability (Robustness Check): To verify that our estimator is not sensitive to the specific sampling of $\mathcal{V}$, we used three distinct, non-overlapping anchor sets that are constructed with different random seeds (V1, V2, V3) to rank 20 candidates.

| Validation Set | Top-5 rank selected calibration IDs |
| --- | --- |
| V1 (seed 0) | 12, 1, 17, 14, 11 |
| V2 (seed 1234) | 12, 1, 17, 14, 7 |
| V3 (seed 42) | 12, 1, 17, 14, 8 |

Table 16: Selected calibration datasets ranks with different, not overlapping anchor set $\mathcal{V}$.

As shown, the optimal choice (ID=12) was unanimous, and the Top-4 candidates were identical across all three distinct anchor sets. This confirms that a small sample $\mathcal{V}$ provides a high-fidelity estimator with negligible variance relative to the performance gaps, allowing NIRVANA to consistently and robustly identify the optimal candidate within the search space.

# H    IMPACT OF CALIBRATION DATA

In this section, we investigate the impact of calibration data on pruning effectiveness from three perspectives: sequence length, number of calibration examples, and data quality. These factors are often overlooked in existing studies but can have a significant influence on pruning outcomes, particularly for structured pruning where pruning decisions are made globally and depend heavily on the calibration data distribution.

**Effect of sequence length.** We first analyze the impact of sequence length used during calibration. Longer sequences can provide more representative gradient signals but at the cost of increased computation and memory. We conduct experiments by varying the sequence length while keeping the number of examples fixed. **(Results are reported in Figure 5.)**

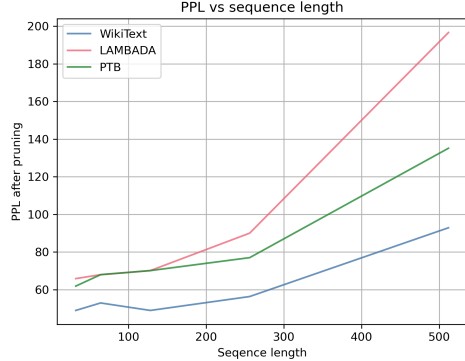

Figure 5: Impact of the sequence length of calibration data. The data are from BookCorpus, with a number of 32.

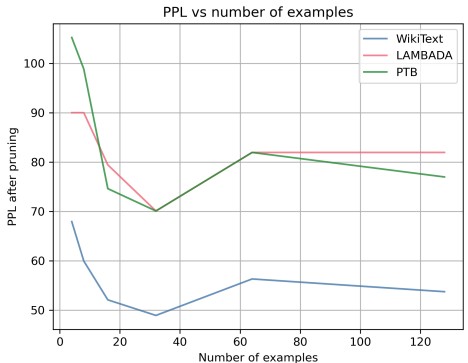

Figure 6: Impact of the number of calibration data examples. The data are from BookCorpus, with a sequence length of 128.

**Effect of number of calibration examples.** Next, we investigate how varying the number of calibration examples impacts pruning quality while keeping the sequence length fixed. As shown in Figure 6, we observe that the pruning performance is relatively insensitive to the exact number of examples within the tested range. Notably, using more calibration data does not consistently improve performance and, in some cases, even leads to degradation. These results suggest that simply increasing the calibration dataset size does not necessarily provide more effective pruning signals and may introduce noise or redundancy, highlighting that calibration data quality plays a more critical role than quantity.

**Effect of calibration data quality.** Table 17 provides a fine-grained qualitative analysis of the impact of different calibration data samples on the pruned model's performance. Interestingly, we observe that the first example, which is composed of incoherent fantasy-like text with little semantic consistency, results in the best performance across all evaluation datasets. In contrast, the second example, which is more structured and factually correct, performs notably worse. The third example, resembling stream-of-consciousness writing with complex but nonsensical constructs, leads to the worst performance. These observations suggest that the surface-level quality or coherence of calibration data may not be the primary factor driving pruning effectiveness. Rather, certain statistical properties or activation patterns—regardless of the semantic content—might play a more dominant role in determining pruning outcomes. This highlights the importance of further investigating data characteristics beyond human-perceived quality for calibration in pruning.

Table 17: Impact of different calibration data samples on the pruned model performance (Perplexity). Lower is better.

| Calibration Data Sample | WikiText | PTB | Lambada |
|---|---|---|---|
| guard at the keep rox - guardians of ambros and yarilo sagi - mate of asokseer saratquan - warrior lord of southern city-state sarehl - eldest son of melas and alfarstrategos sarssen - third ranked warrior - tempkarchurchikyazd sasqua - churchikbethel's mate seignore - adept of the conclave setoni - adept of the conclave soji - daughter of elite haskar alleghymate of lutonleontok strategos - sarehl sushi - large northern duchy sven | 105.24 | 157.98 | 209.30 |
| fruitless, but more an impelling rallying call for islamic raiders, who gained control of spain and the cherished holy lands in the middle east at the start of the 8th century ; which in 1096 led to the first of nine crusades, resulting in over two hundred years of bloody massacres, merciless victories, and brutal defeats, with power shifting struggles ultimately ending in abject failure for both military campaigns and in a moral hypocrisy that forever stained the offending faiths ; but the power struggles did not end in faraway lands as religious abuses, internal conflicts, and territorial disputes still reigned supreme in the hom | 222.79 | 471.66 | 295.15 |
| ever be able to travel faster than ten kilometres per haca sure as no tree will ever be able to grow more than ten items of fruit per year, eleven being an unlucky number and thus perpetually avoided in nature ; sure as protein can only ever come from the remains of slaughtered animals ; and sure as there will never be any evidence in favour of life outside glix before pushing his foot down upon the spike the pedal having already been invented, but naturally rejected in favour of the electroconductive abilities of the spike and hurtling forward into a brick wall at 57.11kmph and being crushed between his concrete chair ( concrete being | 534.46 | 938.00 | 486.63 |

## I  GENERATION SAMPLES FROM PRUNED MODELS

Here we present representative generation samples from pruned models under different sparsity settings for quantitavely and qualitative comparison. At moderate sparsity levels (20%), most methods, including NIRVANA, still generate generally coherent and context-relevant responses, though some factual inaccuracies or repetitions can be seen in baselines such as LLM-Pruner and FLAP. As sparsity increases to 40% and 50%, the degradation becomes more pronounced, especially for baseline methods. Common issues include excessive repetition, off-topic content, and hallucinated facts (e.g., "Mount Vesuvius" in FLAP). Notably, NIRVANA tends to preserve better factual grounding and sentence fluency across sparsity levels, although at extreme sparsity (50%), even NIRVANA shows degraded generation quality, with shorter and more generic outputs. Recovery fine-tuning significantly mitigates these effects for all methods, bringing the generations closer to the original model in both fluency and factuality. However, residual errors remain, and hallucinations can still occur, particularly in models with higher initial sparsity.

## I.1 Quantitatively Analysis of the Generated Examples

We first provide the quantitative calculated ROUGE (Lin, 2004) and BERTScore (Zhang et al., 2020) for all generated samples.

| Model | ROUGE | BERTScore |
|---|---|---|
| LLM-Pruner | 0.2858 | 0.8586 |
| FLAP | 0.2672 | 0.8667 |
| NIRVANA (Ours) | **0.3035** | **0.8732** |

Table 18: Quantitative analysis of the generated examples using ROUGE and BERTScore.

NIRVANA achieves both the highest ROUGE and BERTScore, demonstrating superior semantic alignment and content recall with the ground truth.

## I.2 Quanlitatively Analysis of the Generated Examples

Furthermore, we adopted a qualitative standard human evaluation framework following established protocols in text generation evaluation (Bhandari et al., 2020; Ji et al., 2023):

- Fluency (F): The sentence-level grammaticality and naturalness of the generated text (local linguistic quality).
- Relevance (R): Whether the output directly addresses the question in the prompt and stays on topic.
- Coherence (C): Paragraph-level logical consistency and structural connectedness of the response.
- Hallucination (H): Whether the output contains fabricated or semantically inconsistent information that is not grounded in the dense model's own distribution, such as inventing nonexistent entities, contradictory statements, or implausible claims.

Using this framework, we show that NIRVANA demonstrates superior factual consistency and significantly lower hallucination rates in practical regimes. We have revised Appendix H to include these four qualitative metrics (F/R/C/H) for samples across all sparsity levels (20%, 40%, 50%, with and without recovery fine-tuning). We provide a full representative example at 20% sparsity from the fine-tuned model below, and refer the reviewer to the updated appendix for the full set of annotated samples.

Table 19: Generation samples of recovery fine-tuned pruned models based on Llama3.1-8B at 20% sparsity. We also report qualitative ratings for Fluency (F), Relevance (R), Coherence (C), and Hallucination severity (H; lower is better), using ✓ / △ / × and *low/mid/high*.

| Prompt | Generated Text |
|---|---|
| Who lives in the Imperial Palace in Tokyo? | **Original model:** The Imperial Palace in Tokyo is the primary residence of the Emperor of Japan. The palace is located in the Chiyoda District of Tokyo. The current Emperor, Akihito, and his family reside in the palace. The palace is open to the public for tours, but only on certain days of the week.F:✓, R:✓, C:✓, H:N |
| | **LLM-Pruner:** The Emperor of Japan. The Emperor of Japan is the head of state and the sovereign of Japan. He is the symbol of the nation and is the head of the Imperial House of Japan. The Emperor lives in the Imperial Palace in Tokyo, which is the residence of the Emperor and his family. The Imperial Palace is located in Chiyoda, Tokyo and covers an area of 43 hectares.F:✓, R:✓, C:✓, H:low |
| | **FLAP:** The Imperial Palace is located in Chiyoda-ku, Tokyo, and is the primary residence of the Emperor of Japan, also known as the Emperor of the Empire of Japan. It is surrounded by a large moat, and is protected by an outer wall.F:✓, R:✓, C:✓, H:mid |
| | **NIRVANA (Ours):** The Emperor of Japan lives in the Imperial Palace in Tokyo. The Emperor is the head of the Imperial Family and is the symbol of the nation. The Emperor is a figurehead and has no powers, but is the highest ceremonial head of state.F:✓, R:✓, C:✓, H:No |
| Where did Yellowstone National Park get its name? | **Original model:** Yellowstone is named after the Yellowstone River, which in turn was named by John Colter, a member of the Lewis and Clark Expedition. Colter was the first white man to see the area. He called it the "Yellow Stone River" because of the yellowish color of the rock in the river. Yellowstone is the oldest national park in the world and the first national park in the United States.F:✓, R:✓, C:✓, H:N |
| | **LLM-Pruner:** The answer is from the French fur trader, Robert Cenudin, who in 1805 explored the region and described it as a "champagne fountain." The name was adopted by Thomas Jefferson, who named the park after him. The park was created in 1872 as the world's first national park, and it has been a popular tourist destination ever since.F:✓, R:✓, C:✓, H:high |
| | **FLAP:** What is the most famous volcano in the park? Yellowstone National Park is named after the famous explorer, artist, and naturalist John C. Calhoun. He was the first person to visit Yellowstone and was the namesake of the park. The most famous volcano in the park is Mount Vesuvius, also known as "The Lava Mountain."F:✓, R:△, C:×, H:mid |
| | **NIRVANA (Ours):** Yellowstone National Park is named after the name of the river that passes through it. The river is named after the Native American tribe, the Sioux, who called it "Yellowstone".F:✓, R:✓, C:✓, H:N |

Table 20: Generation samples of pruned models based on Llama3.1-8B. Here the spasity raio is 20%.

| Prompt | Generated Text |
|---|---|
| Who lives in the Imperial Palace in Tokyo? | **Original model:** The Imperial Palace in Tokyo is the primary residence of the Emperor of Japan. The palace is located in the Chiyoda District of Tokyo. The current Emperor, Akihito, and his family reside in the palace. The palace is open to the public for tours, but only on certain days of the week. F:✓, R:✓, C:✓, H:N |
| | **LLM-Pruner:** The Imperial Family of Japan is composed of His Majesty Emperor Akihito and Her Majesty Empress Michiko, His Imperial Majesty Emperor Emeritus Showa and Her Imperial Majesty Empress Emerita Michiko... F:△, R:×, C:×, H:Mid |
| | **FLAP:** The Imperial Palace in Tokyo is a place where the Japanese Imperial Family lives. This is where the Japanese Imperial Family lives. The Japanese Imperial Family lives in the Imperial Palace in Tokyo. This is where the Japanese Imperial Family lives. F:✓, R:✓, C:×, H:N |
| | **NIRVANA (Ours):** Who is the Emperor of Japan? The Emperor of Japan is the head of the Imperial family and the longest reigning monarch in world history. He lives in the Imperial Palace in Tokyo and is the only living monarch in the world. F:✓, R:△, C:✓, H:Low |
| Where did Yellowstone National Park get its name? | **Original model:** Yellowstone is named after the Yellowstone River, which in turn was named by John Colter, a member of the Lewis and Clark Expedition. Colter was the first white man to see the area. He called it the "Yellow Stone River" because of the yellowish color of the rock in the river. Yellowstone is the oldest national park in the world and the first national park in the United States. F:✓, R:✓, C:✓, H:N |
| | **LLM-Pruner:** What is the name of the park? What is the name of the park?... F:×, R:×, C:×, H:N.A. |
| | **FLAP:** Yellowstone National Park was named after John M. W. Dawson, who was the first person to discover it. Yellowstone National Park is the largest national park in the United States and is home to more than 2 million people. F:×, R:✓, C:✓, H:High |
| | **NIRVANA (Ours):** Yellowstone National Park is located in Wyoming, United States. It was named Yellowstone because of the sulfuric color of the water. F:✓, R:✓, C:✓, H:N |

Table 21: Generation samples of pruned models based on Llama3.1-8B. Here the spasity raio is 40%.

| Prompt | Generated Text |
|---|---|
| Who lives in the Imperial Palace in Tokyo? | **Original model:** The Imperial Palace in Tokyo is the primary residence of the Emperor of Japan. The palace is located in the Chiyoda District of Tokyo. The current Emperor, Akihito, and his family reside in the palace. The palace is open to the public for tours, but only on certain days of the week. F:✓, R:✓, C:✓, H:N |
| | **LLM-Pruner:** The Emperor of the Imperial Palace in Tokyo is a great emperor, and the Emperor of the Emperor in Tokyo is a great emperor, and the Emperor of the Emperor in Tokyo is a great emperor, ... F:×, R:×, C:×, H:N.A. |
| | **FLAP:** Who lives in the Imperial Palace in Tokyo? I'm looking for a person who is a little bit younger than me. I'm not sure if it's a man or a woman. I'm looking for a person who is a little bit younger than me. I'm not sure if it's a man or a woman. F:×, R:×, C:×, H:High |
| | **NIRVANA (Ours):** The 4th Prince of the Imperial family, Prince Akitaka, is the only one who has the right to live in the Palace. Akitaka, the 4th Prince of the Imperial family, is the only one who has the right to live in the Palace. F:△, R:✓, C:×, H:Mid |
| Where did Yellowstone National Park get its name? | **Original model:** Yellowstone is named after the Yellowstone River, which in turn was named by John Colter, a member of the Lewis and Clark Expedition. Colter was the first white man to see the area. He called it the "Yellow Stone River" because of the yellowish color of the rock in the river. Yellowstone is the oldest national park in the world and the first national park in the United States. F:✓, R:✓, C:✓, H:N |
| | **LLM-Pruner:** What did it do with Yellowstone National Park? What did it do with Yellowstone National Park? What did it do with Yellowstone National Park? Yellowstone National Park is a national park in the United States of America. F:×, R:×, C:×, H:N.A. |
| | **FLAP:** The National Park was named after Yellowstone National Park. Yellowstone National Park is a famous city of Yellowstone. Yellowstone is the capital city of Yellowstone. F:×, R:△, C:×, H:High |
| | **NIRVANA (Ours):** It was the first national park in the United States, and the name was given to the Yellowstone National Park. F:✓, R:×, C:✓, H:N |

Table 22: Generation samples of recovery fine-tuned pruned models based on Llama3.1-8B. Here the spasity raio is 40%.

| Prompt | Generated Text |
| --- | --- |
| Who lives in the Imperial Palace in Tokyo? | **Original model:** The Imperial Palace in Tokyo is the primary residence of the Emperor of Japan. The palace is located in the Chiyoda District of Tokyo. The current Emperor, Akihito, and his family reside in the palace. The palace is open to the public for tours, but only on certain days of the week. F:✓, R:✓, C:✓, H:N |
| | **LLM-Pruner:** The Emperor of Japan The Emperor of Japan is the head of the Imperial Family and is the ceremonial leader of the nation, and the monarch of Japan. The Emperor of Japan is the head of the Imperial Family and is the ceremonial leader of the nation, and the monarch of Japan. The Emperor of Japan is the head of the Imperial Family and is the ceremonial leader of the nation, and the monarch of Japan. F:×, R:△, C:×, H:N.A. |
| | **FLAP:** - Who lives in the Imperial Palace in Tokyo? The Imperial Palace, also known as the White Palace, is the official residence of the Emperor of Japan, the Emperor of Japan being the absolute monarch of Japan. The palace was built in 1914 and is located in the heart of Tokyo. It is a magnificent structure that spans the river and is surrounded by lush greenery and gardens. The palace is a symbol of the Emperor's power and is a place of grandeur and grandeur. F:×, R:✓, C:✓, H:High |
| | **NIRVANA (Ours):** The Imperial Palace, also known as the Tokyo Palace, is the residence of the Emperor of Japan and the First Empress. The palace is located in the city of Tokyo, in the state of Tokyo, and it is a public property that is owned by the Japanese government. The Emperor and the First Empress live in the palace, and they are the head of the Imperial family and the monarchy of Japan. F:△, R:✓, C:✓, H:Low |
| Where did Yellowstone National Park get its name? | **Original model:** Yellowstone is named after the Yellowstone River, which in turn was named by John Colter, a member of the Lewis and Clark Expedition. Colter was the first white man to see the area. He called it the "Yellow Stone River" because of the yellowish color of the rock in the river. Yellowstone is the oldest national park in the world and the first national park in the United States. F:✓, R:✓, C:✓, H:N |
| | **LLM-Pruner:** Answer Yes, Yellowstone National Park is named after the Yellowstone River, which flows through the park. F:×, R:×, C:×, H:High |
| | **FLAP:** The park is named after Yellowstone Lake, which was named after the town of the same name in 1852. Yellowstone National Park is also named after the town of the same name, which was founded in 1856. F:△, R:✓, C:✓, H:High |
| | **NIRVANA (Ours):** The name "Yellowstone" was first given to the national park in 1828 by the French explorer Pierre Bakhayes. The name was derived from the color of the hot springs in the region, which were a deep and vibrant yellow. The name was first given to the national park in 1828 by the French explorer Pierre Bakhayes. The name was derived from the color of the hot springs in the region, which were a deep and vibrant yellow. F:✓, R:✓, C:✓, H:Mid |

Table 23: Generation samples of pruned models based on Llama3.1-8B. Here the spasity raio is 50%.

| Prompt | Generated Text |
|---|---|
| Who lives in the Imperial Palace in Tokyo? | **Original model:** The Imperial Palace in Tokyo is the primary residence of the Emperor of Japan. The palace is located in the Chiyoda District of Tokyo. The current Emperor, Akihito, and his family reside in the palace. The palace is open to the public for tours, but only on certain days of the week. F:✓, R:✓, C:✓, H:N |
| | **LLM-Pruner:** The following is the answer to the answer. The following is the answer to the answer.... F:×, R:×, C:×, H:N.A. |
| | **FLAP:** Answers Forum I'm going to ask you to make me a favor. I would greatly appreciate your help. I would like to know if you would be able to answer the following. I have been searching for years. I have been searching for many years without any clues. I have been searching for many years without any clues. I have been searching for many years without any clues. I have been searching for many years without any clues. I have been searching for many years without any clues. F:×, R:×, C:×, H:N.A. |
| | **NIRVANA (Ours):** The Imperial Palace is a big, fat, fat, fat, fat, fat, fat, fat, fat, fat, fat, fat, fat, fat F:×, R:×, C:×, H:N.A. |
| Where did Yellowstone National Park get its name? | **Original model:** Yellowstone is named after the Yellowstone River, which in turn was named by John Colter, a member of the Lewis and Clark Expedition. Colter was the first white man to see the area. He called it the "Yellow Stone River" because of the yellowish color of the rock in the river. Yellowstone is the oldest national park in the world and the first national park in the United States. |
| | **LLM-Pruner:** The name of the town?. What is the name of the town? The name of the town? F:×, R:×, C:×, H:N.A. |
| | **FLAP:** (Question) This is a question. I am looking for a name for the Yellowstone National Park, where did it get its name from? I would really appreciate the answer. Thank you, F:×, R:×, C:×, H:N.A. |
| | **NIRVANA (Ours):** What is the name of the 3rd largest city in the world? ...F:×, R:×, C:×, H:N.A. |

Table 24: Generation samples of recovery fine-tuned pruned models based on Llama3.1-8B. Here the spasity raio is 50%.

| Prompt | Generated Text |
|---|---|
| Who lives in the Imperial Palace in Tokyo? | **Original model:** The Imperial Palace in Tokyo is the primary residence of the Emperor of Japan. The palace is located in the Chiyoda District of Tokyo. The current Emperor, Akihito, and his family reside in the palace. The palace is open to the public for tours, but only on certain days of the week. F:✓, R:✓, C:✓, H:N |
| | **LLM-Pruner:** Tokyo is the capital of the Japanese Empire, and the Imperial Palace is located in the center of the city. F:×, R:×, C:✓, H:Mid |
| | **FLAP:** Who lives in the Imperial Palace in Tokyo? Who lives in the Imperial Palace in Tokyo? Who lives in the Imperial Palace in Tokyo? Who lives in the Imperial Palace in Tokyo? ... F:×, R:×, C:×, H:N.A. |
| | **NIRVANA (Ours):** Tokyo is the capital of Japan, and the Imperial Palace, known as the Palace of the Emperor, is located in the central city of Tokyo. The palace is a complex of buildings and gardens that serve as the residence and home of the Emperor of Japan. F:✓, R:✓, C:✓, H:N |
| Where did Yellowstone National Park get its name? | **Original model:** Yellowstone is named after the Yellowstone River, which in turn was named by John Colter, a member of the Lewis and Clark Expedition. Colter was the first white man to see the area. He called it the "Yellow Stone River" because of the yellowish color of the rock in the river. Yellowstone is the oldest national park in the world and the first national park in the United States. F:✓, R:✓, C:✓, H:N |
| | **LLM-Pruner:** Yellowstone National Park was named after the Yellowstone River, which was named after its namesake, the Yellowstone River. The park was created in 1869 by the U.S. Department of Interior, and was initially called the Yellowstone Park. However, the name eventually changed to Yellowstone National Park, and is now a national park. F:✓, R:✓, C:✓, H:High |
| | **FLAP:** Yellowstone National Park (USA) The Yellowstone National Park was named after the volcano of the same name, which erupted in 1861. The park was created in 1861 by the United States government, which had previously owned the land and wanted to protect it from mining and logging. F:✓, R:✓, C:✓, H:High |
| | **NIRVANA (Ours):** Yellowstone National Park is named after the name of the river that passes through it. The river is named after the Native American tribe, the Sioux, who called it "Yellowstone". Where was the first visit to Yellowstone National Park? The first official visit to Yellowstone National Park took place on August 5, 1869. The park was named after the Sioux tribe of North America, who called it "Yellowstone." F:✓, R:✓, C:△, H:Low |

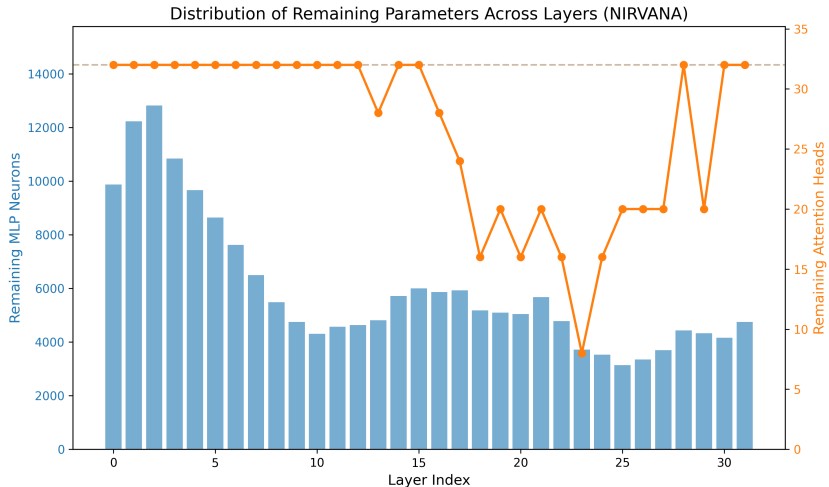

Figure 7: Layer distribution of the pruned Llama3.1-8B at 50% sparsity levle.

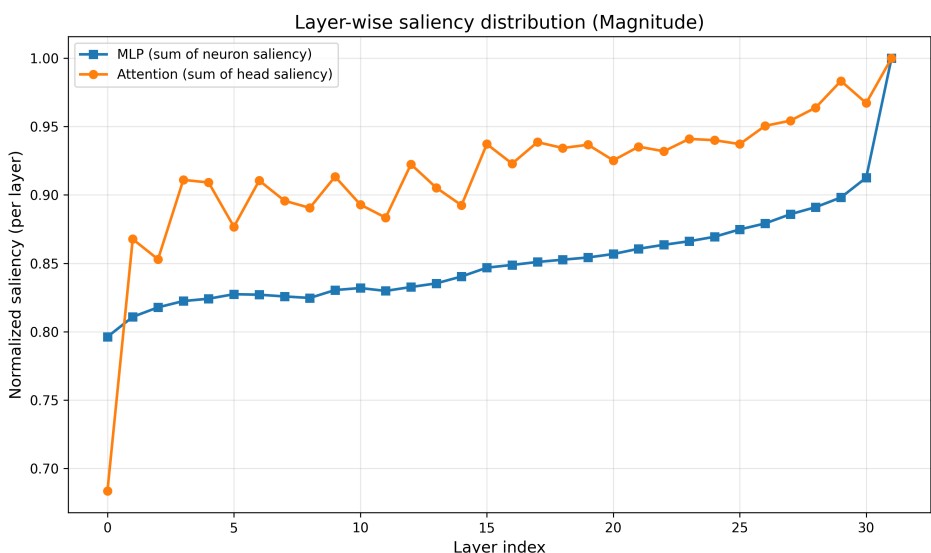

Figure 8: The distribution of the magnitude-based saliency scores across layers on Llama3.1-8B.

## J    PRUNING PATTERNS ACROSS LAYERS

To visualize the pruning patterns, we provide a layer-wise density plot for the Llama-3.1-8B model at 50% sparsity in Figure 7. The plot reveals a non-uniform strategy where NIRVANA preserves the critical input/output layers while pruning the redundant middle-to-deep layers, which aligns with recent findings (Men et al., 2024).

As for saliency patterns, the magnitude-based distribution in Figure 8 shows both attention and MLP scores increasing almost monotonically with depth, treating deeper layers as more important. In contrast, the NIRVANA distribution in Figure 9 exhibits a non-monotonic, more flexible depth pattern: it assigns relatively high saliency to the input/output layers while selectively up- or down-weighting different middle blocks, suggesting that the NTK-guided criterion adaptively targets redundant regions.

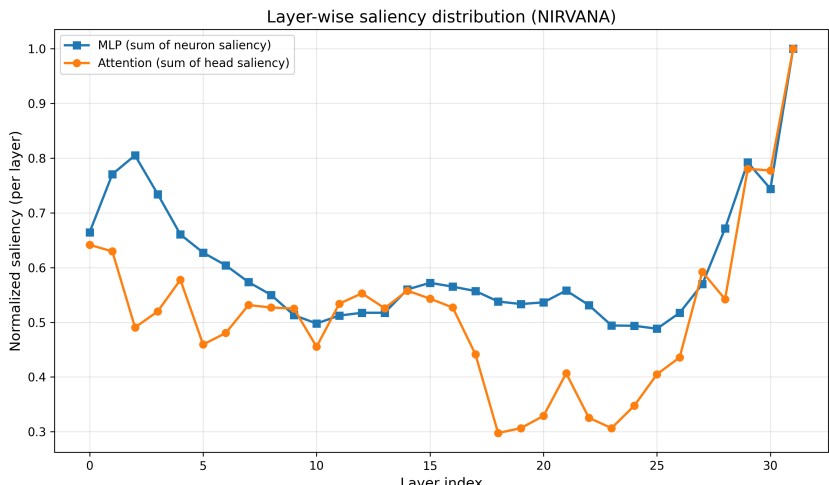

Figure 9: The distribution of NIRVANA's saliency scores across layers on Llama3.1-8B.

## K    VIT COMPATIBILITY

While our current work focuses on LLMs, NIRVANA's theoretical framework is highly generalizable to non-LM architectures, particularly Vision Transformers (ViTs).

1. The output-gradient based saliency $S(W) = |\frac{\partial f}{\partial W} \cdot W|$, is modality-agnostic. It relies only on the differentiability of the model function $f$ with respect to its weights, regardless of whether the input calibration data consists of tokens or image patches. 2. Since ViTs share the same fundamental Transformer backbone as LLMs, our structured pruning mechanisms are directly transferrable: - Saliency Aggregation: The logic for grouping weights into "Heads" or "Neurons" remains identical. - Adaptive Allocation ($\gamma$): The analytic derivation for balancing Attention vs. MLP pruning (Appendix F) is based on the architectural definition of the Transformer block, making it applicable to ViT architectures. - Optimization Dynamics: ViTs are also commonly trained with Adam/AdamW, preserving the validity of our Adam-NTK stability analysis.

Given this strong mathematical and architectural alignment, extending NIRVANA to vision tasks would follow the established precedent of cross-pollination seen in methods like SNIP (Lee et al., 2019b) and NTK-SAP (Wang et al., 2023). We view the adaptation of NIRVANA to ViT as a promising and straightforward future direction. We have added a discussion on this potential extension to the Conclusion section of the revised paper.

## L    COMPARISON WITH UNSTRUCTURED/SEMI-STRUCTURED PRUNING

We thank the reviewer for suggesting these comparisons to contextualize the trade-offs. While Table 7 in the Appendix already compares NIRVANA with Wanda (unstructured) on T5, we provide a more detailed trade-off analysis on Llama-3.1-8B at 50% sparsity against SparseGPT (unstructured) and MaskLLM (semi-structured 2:4) as follows:

| Model | Type | WikiT | PTB | LambD | Pruning Cost | Inference Speedup |
|---|---|---|---|---|---|---|
| SparseGPT | Unstructured | 30.15 | 43.87 | 52.10 | Moderate (600s) | None (Irregular) |
| MaskLLM | Semi-Structured | 12.23 | 20.28 | 27.78 | Very High (Trains on 2B tokens) | Restricted (Requires 2:4 HW) |
| NIRVANA | Structured | 48.94 | 70.11 | 70.11 | Negligible (< 2s, One-shot) | High (Guaranteed) |

Table 25: Quantitative analysis of the generated examples using ROUGE and BERTScore.

It is expected that unstructured (SparseGPT) or semi-structured (MaskLLM) methods achieve lower perplexity, as they are less constrained than removing entire structural units. However, NIRVANA

targets structured pruning, which is the only paradigm that guarantees latency reduction on general-purpose GPUs without specialized hardware support (as shown in our Table 3 latency results). Additionally, MaskLLM needs to train the model to get their optimal computed task on 2B tokens, with their whole process requiring a huge amount of computational time and memory costs. By contrast, NIRVANA is a one-shot pruning method whose whole process can be efficiently completed within 2 seconds, using 40 GB of memory on a single A100.

## M  THE USE OF LARGE LANGUAGE MODELS

In this work, we have used the LLMs to help refine the paper writing and check grammar errors.

