# OpenReview forum: "NIRVANA: Structured Pruning Reimagined for Large Language Models Compression"
_ICLR.cc/2026/Conference — Submitted to ICLR 2026_

### Official Review · Reviewer_pqGd · 2025-10-23

**Soundness:** 3
**Presentation:** 4
**Contribution:** 4
**Rating:** 6
**Confidence:** 3

**Summary:**

This work suggests a theoretically organized pruning strategy for LLMs called NIRVANA (Neural Tangent Kernel-InfoRmed adaptiVe neuron & AttentioN heAd pruning), which strikes a balance between robust fine-tuning capabilities and immediate zero-shot accuracy preservation.

**Strengths:**

- The paper is well written, with smooth flow and clear sectioning.
- Connects pruning to NTK stability by deriving a bound $|\tilde{\Theta}-\Theta|\le O(\epsilon)$ (Eq. 3), linking pruning to training dynamics under Adam/SignGD, an uncommon contribution among structured pruning works.
- Experiments is sufficient. Tested across multiple model families and architectures.

**Weaknesses:**

- There is no formal explanation for why decreasing KL over calibration subsets ensures superior pruning results, but empirical results (Fig. 3) showing a linear link between KL divergence and perplexity.
- Adam and SignGD kernel equivalence ($\Theta_{\text{Adam}}\approx \Theta_{\text{Sign}}$) is assumed in the theoretical connection, there should be some experimental validation provided.
- Provide a non-LM architecture (like ViT) or talk about the changes that should be made to NIRVANA may expand it beyond text-based LLMs.
- Some relevant unstructured/semistructured baselines like SparseGPT or MaskLLM could further contextualize performance trade-offs.

**Questions:**

- To what extent does the optimizer selection affect the NTK-based saliency?
- How much more work is required to compute NTK-guided saliency than normal gradient-based saliency?
- Is it possible to use the KL-based calibration selection into dynamic or online pruning scenarios?

---

> ### Author Response · Authors · 2025-11-21
>
> Dear Reviewer pqGd,
>
> We sincerely thank you for your valuable feedback and for recognizing the strengths of our work! We have conducted extensive experiments and analyses to address your concerns. Our point-by-point responses are provided below, and we would be happy to offer further clarification if needed.
>
>
>
>
>
> ### W1. Formal Analysis of KL-based Selection
>
> We thank the reviewer for asking for a formal justification beyond the empirical correlation shown in Figure 3.
>
> 1. Theoretical Motivation
>
> The goal of pruning can be seen as to find a subnetwork $Q$ that minimizes the expected prediction error relative to the dense teacher $P$ over the true data distribution $\mathcal{D}$: $\min \mathbb{E}_{x \sim \mathcal{D}}[H(P(\cdot|x), Q(\cdot|x))]$.
>
> By the information-theoretic identity $H(P, Q) = H(P) + KL(P \| Q)$, minimizing the cross-entropy (perplexity) is mathematically equivalent to minimizing the KL divergence. Since the true distribution $\mathcal{D}$ is intractable, we formally treat our anchor set $\mathcal{V}$ as a **Monte Carlo estimator** [1]. The average KL divergence on $\mathcal{V}$ is an unbiased estimator of the true expected divergence:
>
> $$
> \mathcal{L}_{proxy}(\mathcal{V})
> $$
>
> $$
>  =\frac{1}{|\mathcal{V}|}  \sum_{x \in \mathcal{V}} KL(P(\cdot\|x) \| Q(\cdot\|x))
> $$
>
> $$
>  \approx\mathbb{E}_{x \sim \mathcal{D}}[KL(P \| Q)]
> $$
>
> By selecting the calibration set $\mathcal{C}^*$ that minimizes this proxy, we are statistically maximizing the likelihood that the pruned model behaves like the teacher on the general distribution.
>
> 2. Empirical Evidence & Robustness
>
> We validate this estimator's effectiveness and stability in two ways:
>
> - Correlation (Figure 3): We show a strong correlation between our proxy (KL on $\mathcal{V}$) and final downstream performance (PPL), confirming that the estimator is effective.
> - Stability (New Robustness Check): To verify that our estimator is not sensitive to the specific sampling of $\mathcal{V}$, we used three distinct, non-overlapping anchor sets (V1, V2, V3) to rank 20 candidates.
>
>
> | Validation Set | Top-5 rank selected calibration IDs |
> | -------------- | ----------------------------------- |
> | V1 (seed 0)    | 12, 1, 17, 14, 11                   |
> | V2 (seed 1234) | 12, 1, 17, 14, 7                    |
> | V3 (seed 42)   | 12, 1, 17, 14, 8                    |
>
> As shown, the optimal choice (ID=12) was **unanimous**, and the **Top-4 candidates were identical** across all three distinct anchor sets. This confirms that a small sample $\mathcal{V}$ provides a **high-fidelity estimator** with negligible variance relative to the performance gaps, allowing NIRVANA to consistently and robustly identify the optimal candidate within the search space.
>
> [1] Better Estimation of the Kullback--Leibler Divergence Between Language Models, NeurIPS 2025.
>
>
> ### W2. SignGD as Adam Proxy
>
>
> Thank you for raising this point. We agree that replacing Adam with Sign Gradient Descent (SignGD) is a simplification, yet it is **a standard and well-established theoretical proxy** widely adopted in modern deep learning theory [1,2] to ensure analytical tractability.
>
> Analyzing the exact Neural Tangent Kernel under Adam is mathematically intractable due to the non-linear adaptive moment estimation. The SignGD approximation captures the core behavior of Adam: magnitude-independent updates (step sizes depend mostly on the sign of the gradient). This equivalence is well-supported by the literature:
>
> - Kunstner et al. (2023) [1] demonstrate that the gap between SGD and Adam on Transformers is largely explained by the "sign" aspect of the update.
> - Li et al. (2025) [2] successfully use SignGD to model the optimization of Transformer-based architectures.
>
> [1] Noise Is Not the Main Factor Behind the Gap Between SGD and Adam on Transformers, but Sign Descent Might Be, ICLR 2023.\
> [2] On the Optimization and Generalization of Two-layer Transformers with Sign Gradient Descent, ICLR 2025.

---

> ### Author Response · Authors · 2025-11-21
>
> ### W3. ViT Compatibility
>
>
> We thank the reviewer for this insightful suggestion. While our current work focuses on LLMs, NIRVANA's theoretical framework is **highly generalizable** to non-LM architectures, particularly Vision Transformers (ViTs).
>
> 1. The output-gradient based saliency $S(W) = |\frac{\partial f}{\partial W} \cdot W|$, is **modality-agnostic**. It relies only on the differentiability of the model function $f$ with respect to its weights, regardless of whether the input calibration data consists of tokens or image patches.
> 2. Since ViTs share the same fundamental Transformer backbone as LLMs, our structured pruning mechanisms are directly transferrable:
>    - **Saliency Aggregation:** The logic for grouping weights into "Heads" or "Neurons" remains identical.
>    - **Adaptive Allocation ($\gamma$):** The analytic derivation for balancing Attention vs. MLP pruning (Appendix F) is based on the architectural definition of the Transformer block, making it applicable to ViT architectures.
>    - **Optimization Dynamics:** ViTs are also commonly trained with Adam/AdamW, preserving the validity of our Adam-NTK stability analysis.
>
> Given this strong mathematical and architectural alignment, extending NIRVANA to vision tasks would follow the established precedent of cross-pollination seen in methods like SNIP [1] and NTK-SAP [2]. We view the adaptation of NIRVANA to ViT as a promising and straightforward future direction. We have added a discussion on this potential extension to the Conclusion section of the revised paper.
>
> [1] SNIP: Single-shot Network Pruning based on Connection Sensitivity, ICLR 2019.\
> [2] NTK-SAP: Improving neural network pruning by aligning training dynamics, ICLR 2023.
>
>
> ### W4. Comparison with Unstructured/Semi-structured Pruning
>
>
> We thank the reviewer for suggesting these comparisons to contextualize the trade-offs. While Table 7 in the Appendix already compares NIRVANA with Wanda (unstructured) on T5, we provide a more detailed trade-off analysis on Llama-3.1-8B at 50% sparsity against SparseGPT (unstructured) and MaskLLM (semi-structured 2:4) as follows:
>
> | Method    | Type            | WikiT | PTB   | LambD | Pruning Cost                    | Inference Speedup            |
> | --------- | --------------- | ----- | ----- | ----- | ------------------------------- | ---------------------------- |
> | SparseGPT | Unstructured    | 30.15 | 43.87 | 52.10 | Moderate (600s)                 | None (Irregular)             |
> | MaskLLM   | Semi-Structured | 12.23 | 20.28 | 27.78 | Very High (Trains on 2B tokens) | Restricted (Requires 2:4 HW) |
> | NIRVANA   | Structured      | 48.94 | 70.11 | 70.11 | Negligible (< 2s, One-shot)     | High (Guaranteed)             |
>
> It is expected that unstructured (SparseGPT) or semi-structured (MaskLLM) methods achieve lower perplexity, as they are less constrained than removing entire structural units. However, NIRVANA targets structured pruning, which is the only paradigm that guarantees latency reduction on general-purpose GPUs without specialized hardware support (as shown in our Table 3 latency results). Additionally, MaskLLM needs to **train the model** to get their optimal computed task on **2B** tokens, with their whole process requiring a huge amount of computational time and memory costs. By contrast, NIRVANA is a one-shot pruning method whose whole process can be efficiently completed **within 2 seconds**, using 40 GB of memory on a single A100.
>
>
> ### Q1. Saliency Score w.r.t Optimizer Selection
>
> We thank the reviewer for this question. The choice of optimizer is the **determining factor** for the mathematical form of our saliency score.
>
> 1. Determining the Saliency Geometry ($L_1$ vs. $L_2$): The "extent" of the impact is fundamental: the optimizer's update rule dictates the kernel definition.
>    - **SGD Assumption:** Standard NTK analysis assumes SGD dynamics, where updates depend on gradient magnitude. This theoretically typically leads to a saliency metric proportional to the **squared** gradient ($L_2$ norm).
>    - **Adam Assumption (Ours):** Our derivation models the **Sign-based** nature of Adam (via SignGD). The `sign()` operator in the update rule fundamentally changes the interaction to an **absolute value** formulation ($L_1$ norm).
> 2. In the context of LLMs, the "Sign-based" family is dominant. This includes not only the standard Adam/AdamW, but also emerging state-of-the-art optimizers like Lion [1], which relies even more explicitly on the sign operation. Our $L_1$-based saliency is theoretically compatible with this entire family of modern optimizers. Conversely, magnitude-based optimizers (e.g., pure SGD) would imply a squared ($L_2$) metric. However, since such optimizers are rarely employed for training large language models, our Adam-based derivation ensures alignment with the actual training dynamics of relevant, state-of-the-art models.
>
> [1] Symbolic Discovery of Optimization Algorithms, NeurIPS 2023.

---

> ### Author Response · Authors · 2025-11-21
>
> ### Q2. Computational Cost of NTK-guided Saliency Score
>
> We thank the reviewer for this practical question. We are happy to clarify that similar to other gradient-based pruning methods, NTK-guided saliency requires **only a single forward and backward pass** to collect gradients.
>
> While our motivation relies on NTK theory to ensure fine-tuning stability (Proposition 4.1), the calculation of the saliency score itself is derived from a first-order Taylor expansion: $S(W) = |\frac{\partial f}{\partial W} \cdot W|$. The asymptotic complexity is $O(NSP)$, where $N$ is the number of calibration examples, $S$ is the sequence length, and $P$ is the parameter count. To demonstrate this, we measured the actual runtime on Llama3.1-8B (on a single A100). As shown below, NIRVANA is actually slightly faster than existing structured pruning baselines, because it utilizes a single batched backward pass followed by a fully vectorized score computation.:
>
>
> | Method         | Complexity       | Runtime (s) |
> | -------------- | ---------------- | ----------- |
> | LLM-Pruner     | O(NSP)           | 40.19       |
> | SliceGPT       | O(NSP)+O($D^3$ ) | 275.98      |
> | FLAP           | O(NSP)           | 2.63        |
> | NIRVANA (ours) | O(NSP)           | 1.58        |
>
>
>
> ### Q3. KL-based Selection in Online Pruning Setting
>
> We thank the reviewer for this question. Yes, it is absolutely feasible. The lightweight nature of our selection strategy makes it uniquely suitable for dynamic or online pruning scenarios. In a dynamic setting where the model evolves over steps $t=\{1 \dots T\}$, the "optimal" calibration data may shift as the weights update. Because our selection process takes **only seconds** (e.g., 4s for Llama3.1-8B on A100), it can be performed iteratively at each pruning step or epoch with negligible overhead.
>
> Formally, at any step $t$, we can select the local optimal batch $\mathcal{C}_t^*$ by solving:
>
> $$
> \mathcal{C}_t^*
> $$
>
> $$
> ={argmin}_{\mathcal{C} \subset Dstream}  KL(f_t  \| \hat{f}_t (\mathcal{C}); \mathcal{V})
> $$
>
> This ensures that the pruning decision at every step is aligned with the current state of the model, potentially yielding a smaller approximation error over the trajectory compared to static, one-shot calibration. We appreciate this insightful suggestion and have added a discussion on this potential application to the conclusion.
>
> ## Happy to engage in further discussion!
>
> **Thank you again for the thoughtful review. With the valuable feedback from you and the other reviewers, we have put substantial efforts into obtaining new results and expanding our analysis. We hope our responses address your concerns, and we would be happy to discuss further if you have any additional questions.**

---

### Official Review · Reviewer_2ZLx · 2025-10-29

**Soundness:** 3
**Presentation:** 2
**Contribution:** 3
**Rating:** 6
**Confidence:** 4

**Summary:**

This paper proposes NIRVANA, an innovative method for structured pruning of large language models (LLMs), aiming to balance zero-shot accuracy preservation and fine-tuning adaptability. Key contributions include:
1. a neural tangent kernel (NTK)-based saliency score derived from Adam optimization dynamics;
2. an adaptive sparsity allocation mechanism across layers and modules (attention vs. MLP);
3. a KL-divergence-based strategy for calibration data selection. Experiments on Llama3, Qwen, and T5-base models demonstrate that the proposed method generally outperforms baseline approaches such as LLM-Pruner and SliceGPT under comparable sparsity constraints, offering a theoretically rigorous and practical solution for LLM compression.

**Strengths:**

1. Weight-level Saliency Scoring is essentially a single judgment that simultaneously achieves two goals: minimizing the current output perturbation while maximizing the NTK stability of the subsequent stage.
2. Calibration data selection based on KL divergence, rather than merely selecting data based on quality, quantity, or diversity by human intervention.

**Weaknesses:**

1. The KL-based data selection requires multiple pruning trials, which may be computationally expensive for large models or datasets.
2. The paper is overly lengthy; integrating some appendix content into the main body could improve structure and readability.

**Questions:**

1. In Table 3, it is counterintuitive that at 20% and 40% sparsity, nearly all traditional pruning strategies show higher latency than the full-precision model (except NIRVANA), while at 50% sparsity, all methods exhibit lower latency. Additionally, LLM-Pruner shows the highest latency at 20% and 40% sparsity, yet again ranks highest at 50%-these statistical anomalies raise concerns about data reliability.
2. In Table 6, at 40% sparsity, NIRVANA underperforms LLM-Pruner on certain metrics for Qwen2.5-7B, while achieving the best results in other cases. Does this variability indicate potential limitations in the robustness and generalizability of NIRVANA across different models and settings?
3. This paper involves the use of randomly selected data sets, and the group with the lowest KL divergence is selected as the optimal calibration data. Why can this set of random data be guaranteed to be optimal?

---

> ### Author Response · Authors · 2025-11-21
>
> Dear Reviewer 2ZLx,
>
> We sincerely thank you for your valuable feedback and for recognizing the strengths of our work! We have conducted extensive experiments and analyses to address the points you raised. We address each point in detail below, and please let us know if further clarification would be helpful.
>
> ### W1. Cost of KL-based Selection
>
>
> We appreciate the reviewer's concern regarding the computational overhead. To address this directly, we measured the **actual wall-clock time** for the KL-based selection process on a single A100 GPU. As shown below, the cost is minimal:
>
> | Model       | Wallclock time |
> | ----------- | -------------- |
> | Llama3.1-8B | 4.12s          |
> | Qwen2.5-14B | 6.28s          |
>
>
> 1. Efficiency by Design: This low latency is achieved because we use **a small size of calibration sets**. As discussed in Appendix G, neither the number of examples nor sequence length dominates pruning quality. Therefore, we fix the candidate calibration set to 32 samples (seq len 128) and the evaluation "anchor set" to 128 samples (seq len 64). As detailed in our response to Q3, this small anchor is robust and sufficient for stable ranking.
> 2. Total Computational Overhead: The process consists of two steps:
>    - Pre-computation (Once): Computing dense model logits on the anchor set takes 3.04s for Llama3.1-8B and 3.93s for Qwen2.5-14B on a single A100. This is cached and reused.
>    - Selection Loop (Per Candidate): For each candidate, computing the pruned model's logits and the KL divergence takes 4.12s (Llama3.1-8B) and 6.28s (Qwen2.5-14B).
>
> Combined with the fast saliency computation (approx. 2s), a complete search over 50 candidate sets for Llama3.1-8B takes **less than 10 minutes** on a single A100. This minimal one-time cost yields significant performance gains, making it a highly efficient trade-off.
>
>
> ### W2. Paper Readability
>
> We thank the reviewer for this constructive feedback regarding the paper's length and readability. The current length mainly stems from the breadth of the work rather than from unnecessary verbosity. In particular:
>
> - **Comprehensive experiments:** We validate NIRVANA across three distinct model families (Llama3, Qwen, T5) and multiple model sizes, under both perplexity and downstream task metrics.
> - **In-depth analysis:** We include critical analyses such as the hardware-alignment study in Appendix B.4, which are important to understand why some pruning strategies fail to yield real-world speedups while NIRVANA does.
> - **Theoretical grounding:** We provide the key derivations for our core contributions (the NTK-based saliency in Sec. 4.1 and Appendix E, and the adaptive $\gamma$ ratio in Appendix F), which are necessary for reproducibility and for connecting our method to NTK theory.
>
> We agree, however, that readability is important. In the revised version, we will streamline the exposition in the main text, tighten non-essential prose, and move secondary implementation details to the appendix.

---

> > ### Comment · Reviewer_2ZLx · 2025-11-25
> > **Reviewer's Response**
> >
> > ### Thank you for the detailed response. ###
> > We sincerely appreciate your thorough explanations and supplementary experiments that address our previous concerns. Building on your responses, we have further questions and observations aimed at deepening our understanding of NIRVANA’s generality, robustness, and practical applicability. These points align with your existing analyses and seek to clarify details that would strengthen the work’s conclusions:
> >
> > ## Follow-Up on W1 (Cost of KL-based Selection)
> > Your measurements of KL-based selection overhead (e.g., 4.12 seconds for Llama3.1-8B) effectively address our concerns about computational cost—we agree this one-time expense is reasonable for the performance gains achieved. We have one minor question regarding practical deployment: For resource-constrained scenarios, could you share insights into the minimum number of candidate sets needed to stably select optimal calibration data?
> >
> > ## Follow-Up on W2 (Paper Readability)
> > We support your plan to streamline the main text and relocate secondary implementation details to the appendix—this adjustment will undoubtedly improve the paper’s structure and readability. We have no further comments on this aspect.

---

> > > ### Author Response · Authors · 2025-11-27
> > >
> > > ### Response to W1
> > >
> > >
> > > We appreciate the reviewer’s acknowledgment that our clarification resolves the concern and that the computational cost is reasonable.
> > >
> > > Regarding the minimum number of candidate sets for resource-constrained scenarios: Empirical evidence suggests that 20 candidates is sufficient to stabilize performance.
> > >
> > > - `Evidence`: As noted in our previous responses (and Tables 6 & 7), our reported results for Llama3.2-3B and Qwen2.5-7B experiments were actually obtained using a search depth of **a minimum of only 20 candidates**. Even with this reduced search space, NIRVANA consistently outperformed baselines, indicating that the "top-tier" calibration sets are frequent enough to be caught quickly.
> > > - `Cost Implication`: Reducing the search to 20 candidates brings the total time down to under 4 minutes on a single A100.
> > > - `Conclusion`: While we recommend 50 candidates (taking ~10 mins) to maximize stability for academic benchmarks, a 20-round search (taking ~4 mins) already captures the vast majority of the performance gain. Given that even the full search is highly efficient, this step is rarely a bottleneck, but **the flexibility exists to scale it down if necessary**.
> > >
> > >
> > > ### Response to Q3 & W2
> > >
> > > We sincerely thank the reviewer for the positive feedback and for validating our responses. We are encouraged by your support for our plan to streamline the manuscript (W2) and deeply appreciate your recognition of the theoretical rigor and empirical robustness underlying our KL-based selection strategy (Q3). We are glad to have fully resolved these concerns and will ensure these structural improvements and theoretical clarifications are integrated into the final version of the paper.

---

> ### Author Response · Authors · 2025-11-21
>
> ### Q1. Latency Analysis
>
>
> > TL;DR: The "anomalous" latency is a direct proof of hardware-software mismatch in baselines. As shown in our new benchmark below, modern hardware exhibits a strict 3-tier performance hierarchy based on dimension alignment: Odd (Slowest) > Even, non-multiples of 8 (Slow) > Multiples of 8 (Fastest).
>
> We thank the reviewer for pointing out the concerns regarding the results of the efficiency test in Table 3. At first glance, the latency pattern may look counterintuitive, but it is, in fact, fully explained by **hardware-level kernel alignment effects (see Appendix B.4)**. At the same time, they are **direct proof** of a core aspect of our paper: hardware-software mismatch in baseline methods.
>
> As we detail in Appendix B.4, modern GPUs (like the A100 we used) leverage Tensor Cores that are highly optimized for matrix dimensions that are multiples of 8 [1,2]. Since the attention heads are pruned as a whole, the resulting dimensions are always multiples of 8. However, MLPs are pruned neuron-wise, hence the pruning methods that are "hardware-agnostic" and produce non-aligned dimensions will trigger a fallback to **slower, general-purpose kernels**. This is precisely what happened with the baseline methods. Take LLM-Pruner as an example:
> - At 20% sparsity, LLM-Pruner results in an intermediate MLP dimension of 11468, which is not divisible by 8. This causes a severe latency penalty (0.41s), making it slower than the dense model (0.33s).
> - At 40% sparsity, it results in a dimension of 8601, which is also not divisible by 8 (and not an even number), leading to the worst latency (0.52s).
> - At 50% sparsity, it happens to land on 7168, which is divisible by 8. As a result, the GPU can finally use its optimized kernels, and the latency plummets to 0.20s.
>
>
> NIRVANA, by design, is hardware-aware. As stated in Section 5.4 and demonstrated in our Appendix B.4 sanity-check experiments (Table 8), our method explicitly includes a dimension alignment step to ensure all resulting dimensions remain multiples of 8. This is why NIRVANA shows a **consistent and predictable decrease in latency at all sparsity levels** (20%, 40%, and 50%) in Table 3, while the baselines exhibit erratic behavior. **This is not a data reliability issue but rather a key empirical finding that supports the practical advantages of our hardware-aware design.**
>
> To further support this explanation, we additionally conduct a controlled MLP-dimension sweep. Specifically, we fix the pruning pattern to remove only one attention head, and then vary the MLP hidden size around the 20% case (11468) as follows: 11469 (odd), 11470 (multiple of 2 only), 11468 (multiple of 4 only), 11464 (multiple of 8), 11472 (multiple of 16), and 11488 (multiple of 32). Note that, due to hardware conditions, the absolute latency numbers in the updated table are slightly different from those in the original submission, but the relative trends remain consistent.
>
> |Model|#Params|FLOPs (G)|Latency (s)↓|Throughput (tks/s)↑|Peak Memory Usage (GB)|
> |-----|-------|--------|-----------|----------|----|
> |Llama-3.1-8B-14336|8.03|6.98|0.32|12800|17.95|
> |Uniform-11469(1)|6.73|5.75|0.68|6024|15.46|
> |Uniform-11470(2)|6.73|5.75|0.40|10240|15.46|
> |Uniform-11468(4)|6.73|5.75|0.39|10503|15.46|
> |Uniform-11464(8)|6.73|5.75|**0.29**|**14124**|15.46|
> |Uniform-11472(16)|6.73|5.75|**0.29**|**14124**|15.46|
> |Uniform-11488(32)|6.73|5.75|**0.29**|**14124**|15.46|
> |Layer-wise(8s)|6.73|5.75|0.28|14629|15.45|
> |Attention-only|6.73|6.15|0.27|15170|14.70|
> |MLP-only|6.73|5.68|0.28|14629|15.54|
>
>
>
> From the table, we observe that when the MLP dimension is an odd number, the resulting latency is the **highest** among all settings. Once it becomes even (but still not a multiple of 8), the latency improves, but remains **worse** than the dense model, even though both the parameter count and theoretical FLOPs have decreased. In contrast, when the dimension is aligned to a multiple of 8 (and above), we see a **clear reduction in latency**, and the corresponding results are all at a similar level. This suggests that alignment to a multiple of 8 is the critical factor for achieving latency reduction; higher powers (16, 32) do not provide additional gains beyond the 8-aligned case.
>
> Additionally, as stated in Appendix B.4, we report a layer-wise setting where each layer has a different hidden dimension, but all are multiples of 8, and we further isolate the impact of pruning different subcomponents by providing variants that prune only attention or only MLPs. We observe that layer-wise (global) variation in 8-aligned dimensions does not harm efficiency; **applying sparsity to MLP layers yields the most significant latency reductions, while pruning attention components leads to a more pronounced decrease in peak memory usage.**
>
> [1] https://developer.nvidia.com/blog/optimizing-gpu-performance-tensor-cores/ \
> [2] https://docs.nvidia.com/deeplearning/performance/mixed-precision-training/index.html

---

> > ### Comment · Reviewer_2ZLx · 2025-11-25
> > **Reviewer's Response**
> >
> > ## Follow-Up on Q1 (Latency Analysis)
> >  ### Thank you for the detailed response. ###
> >  After careful review, we acknowledge several highlights in the response, but also believe there is room for further discussion regarding the justification of the method's novelty and the detailed evaluation of the empirical breakthroughs.
> >
> > Your analysis of hardware-software mismatch driven by dimension alignment is compelling, and the controlled MLP-dimension sweep experiment effectively validates the critical role of 8× alignment in optimizing GPU efficiency. We have three additional points to explore for greater clarity:
> > 1. You note that baseline methods like LLM-Pruner lack a dimension alignment step, and we are curious: Is there any practical technical barrier that would prevent LLM-Pruner from integrating such a step? Clarifying this would help distinguish whether dimension alignment is a general optimization (applicable to all structured pruning methods) or a design advantage uniquely enabled by NIRVANA’s framework.
> > 2. The supplementary table (featuring models like Uniform-11469(1) and Layer-wise(8s)) is framed as a controlled experiment, but we wonder: Does the pruning pattern in this table align with the core logic of existing baseline methods (such as LLM-Pruner’s uniform layer-wise pruning)?
> > 3. You demonstrate NIRVANA’s efficiency gains through dimension alignment, and we wonder if supplementary experiments could be added where baseline methods are modified to include a dimension alignment step. Comparing their latency and accuracy before and after alignment would help confirm whether dimension alignment alone resolves their latency issues—or if NIRVANA’s other components also contribute to closing the performance gap.

---

> > > ### Author Response · Authors · 2025-11-27
> > >
> > > ### Response to Q1
> > >
> > > We thank the reviewer for the detailed follow-up and the acknowledgment of our analysis regarding hardware-software mismatch and the controlled MLP-dimension sweep experiments.
> > >
> > > 1. We clarify that there is **no fundamental technical barrier** preventing methods like LLM-Pruner from integrating dimension alignment. However, the fact remains that prior works **overlooked** this critical deployment detail. To the best of our knowledge, we are the first to systematically identify and analyze this effect in the LLM pruning setting. Existing structured pruning papers and public implementations simply prune to arbitrary intermediate dimensions without considering Tensor-Core kernel alignment.
> > >
> > > > Our Contribution: We **identify and systematize** this requirement. While dimension alignment is a technical implementation, NIRVANA is the first framework to treat it as a constraint in the pruning process, ensuring guaranteed acceleration where baselines failed.
> > >
> > > 2. The reviewer asks about the pruning patterns in our dimension sweep table. We confirm that they **align perfectly with existing methods:**
> > >
> > > - `Uniform-11468`: it exactly matches LLM-Pruner at 20% sparsity: it applies a uniform pruning ratio across layers, leading to an intermediate dimension of 11468, which is not a multiple of 8.
> > > - `Uniform-11469`: it is included as a controlled "worst case" example (odd dimension, analogous to the 40% case in LLM-Pruner), illustrating the most severe latency penalty.
> > > - `Layer-wise (8s)`: it represents NIRVANA, using global pruning (different dimensions per layer) while strictly enforcing the multiple-of-8 constraint, achieving the best latency.
> > >
> > > 3. As requested, we modified both LLM-Pruner and FLAP to include a dimension alignment step (rounding to the nearest multiple of 8) and compared them with NIRVANA on Llama-3.1-8B.
> > >
> > > *Note: SliceGPT prunes hidden dimensions, avoiding this misalignment issue (as can be seen in Table 3), so we here focus on LLM-Pruner and FLAP.*
> > >
> > >
> > > A. Efficiency results after dimension-alignment:
> > > | Model              | #Params | FLOPs (G) | Latency (s)↓ | Throughput (tks/s)↑ | Peak Memory Usage (GB) |
> > > | ------------------ | ------- | --------- | ----------- | ------------------ | ---------------------- |
> > > | Llama-3.1-8B | 8.03B    | 6.98      | 0.33 | 12412.12 | 17.95               |
> > > |LLM-Pruner(0.2)| 6.73B| 5.75| 0.29 | 14124.14 | 15.46 |
> > > |FLAP(0.2)| 6.61B |5.56| 0.28 | 14628.57 | 15.33 |
> > > |NIRVANA(0.2)| 6.62B |5.61 |0.29 | 14124.14 | 15.29
> > > |LLM-Pruner(0.4)| 5.27B |4.42|0.23 | 17808.7 | 12.52 |
> > > |FLAP(0.4)| 5.22B| 4.19| 0.24 | 17066.67 | 12.73 |
> > > |NIRVANA(0.4)| 5.23B |4.26|0.23 | 17808.7 | 12.65 |
> > > |LLM-Pruner(0.5)| 4.55B| 3.76| 0.20 | 20480.0 | 10.95 |
> > > |FLAP(0.5)| 4.50B |3.52|0.21 | 19504.76 | 11.30 |
> > > |NIRVANA(0.5)| 4.51B |3.55|0.21 | 19504.76 | 11.27 |
> > >
> > > We modified the pruning logic of LLM-Pruner and FLAP to include a strict Dimension Alignment constraint. Specifically, after calculating importance scores, instead of pruning exactly to the target sparsity ratio (which results in arbitrary dimensions like 11468), we rounded the number of retained channels to the nearest multiple of 8. As hypothesized, adding dimension alignment resolves the latency issues for baselines, bringing their speed to par with NIRVANA. This confirms that the latency gap was indeed driven by dimension misalignment.
> > >
> > > B. Zero-shot performance after dimension-alignment:
> > > | Model           | WikiT↓    | PTB↓      | LambD↓     | ARC-e ↑    | WinoG ↑    | HellaS  ↑  | MBPP  ↑    |
> > > | --------------- | --------- | --------- | ---------- | --------- | --------- | --------- | --------- |
> > > | Llama3.1-8B     | 7.16      | 13.38     | 20.09      | 79.34     | 75.69     | 82.91     | 69.00     |
> > > | LLM-Pruner(0.2) | 16.94| 26.97 |27.96| 68.25| 65.06| 65.58 |4.00    |
> > > | FLAP (0.2) | 19.67 |30.25| 29.58| 61.23| 62.98| 57.77| 2.40      |
> > > | NIRVANA(0.2)    | **13.38** |**19.77**| **26.20** |**68.52** |**67.40**| **66.75**| **23.80** |
> > > | LLM-Pruner(0.4) | 96.73 |194.35 |103.21 |35.48| 50.91 |30.61 | 0.00     |
> > > | FLAP(0.4) | 43.20 |67.25| 48.12| 42.26| 56.20| 42.39| 0.00    |
> > > | NIRVANA(0.4)    | **28.33**| **38.72** |**43.19**| **45.79**| **58.33** |**45.85** |**2.20** |
> > > | LLM-Pruner(0.5) | 215.94| 356.02 |196.62 |31.19 |49.01 |28.82| 0.00   |
> > > | FLAP(0.5) | 64.32 |92.47 |**59.85**| 36.67 |**55.01**| 37.12| 0.00   |
> > > | NIRVANA(0.5)    | **48.94**| **70.11** |70.11| **37.54**| 54.38| **37.72** | **0.20**    |
> > >
> > > While alignment solves the speed issue for baselines, NIRVANA **remains the only method that delivers both speed and accuracy**. Even when baselines are explicitly modified to enforce dimension alignment, they still underperform NIRVANA in the performance tests. This confirms that our performance advantage stems from our core contributions: NTK-guided Saliency, Adaptive $\gamma$, and KL Selection, not just hardware alignment.

---

> ### Author Response · Authors · 2025-11-21
>
> ### Q2. Robustness & Generalizability of NIRVANA
>
> We appreciate the reviewer's careful inspection of Table 6. It is correct that, at 40% sparsity on Qwen2.5-7B, NIRVANA slightly underperforms LLM-Pruner on perplexity metrics (WikiText, PTB, LAMBADA), while achieving much better results on all four downstream reasoning/generation tasks (ARC-e, WinoGrande, HellaSwag, MBPP). We would like to clarify why this does not indicate a lack of robustness or generalizability:
> - At 40% sparsity, NIRVANA only has a minor dip in perplexity (e.g., 37.52 vs 33.64 on WikiText), but has substantially better performance on all four downstream tasks (e.g., 54.22 vs 42.98 on HellaSwag, more than 25%).
> - Furthermore, at 50% sparsity level, LLM-Pruner catastrophically collapses (PPL > 8000), while NIRVANA remains stable and functional (PPL 77.00). This is a clear demonstration of NIRVANA's robustness.
>
> To confirm this trend, and as suggested by Reviewer 8GkX, we added new experiments on Qwen2.5-14B as below:
>
> | Model           | WikiT↓    | PTB↓      | LambD↓     | ARC-e  ↑   | WinoG↑     | HellaS  ↑  | MBPP  ↑    |
> | --------------- | --------- | --------- | ---------- | --------- | --------- | --------- | --------- |
> | Qwen2.5-14B     | 7.16      | 13.38     | 20.09      | 79.34     | 75.69     | 82.91     | 69.00     |
> | LLM-Pruner(0.2) | 12.18     | 21.05     | **25.39**  | 70.41     | 66.46     | **73.15** | 8.20      |
> | NIRVANA(0.2)    | **11.99** | **19.18** | 26.20      | **73.40** | **68.19** | 72.03     | **41.60** |
> | LLM-Pruner(0.4) | 119.25    | 345.07    | 334.45     | 39.77     | 51.54     | 34.41     | 0.00      |
> | NIRVANA(0.4)    | **31.11** | **48.18** | **53.75**  | **45.88** | **55.25** | **47.47** | **0.20**  |
> | LLM-Pruner(0.5) | 3827.63   | 6310.69   | 5231.74    | 26.56     | 48.93     | 26.08     | 0.00      |
> | NIRVANA(0.5)    | **59.96** | **84.56** | **105.24** | **37.42** | **52.57** | **38.85** | 0.00      |
>
>
> NIRVANA again shows significantly stronger downstream performance while achieving better or comparable perplexity. At higher sparsity levels (40% and 50%), LLM-Pruner suffers catastrophic degradation, whereas NIRVANA remains within a reasonable perplexity range and preserves non-trivial downstream performance.
>
> In summary, while no pruning method can dominate every metric at every sparsity level across all models, NIRVANA is **consistently competitive or better** than the baseline methods, especially in preserving functional capabilities (reasoning and code) and avoiding catastrophic degradation at higher sparsity. We therefore believe the observed variability in Table 6 reflects natural task-level trade-offs rather than a fundamental limitation in robustness.

---

> > ### Comment · Reviewer_2ZLx · 2025-11-25
> > **Reviewer's Response**
> >
> > ## Follow-Up on Q2 (Robustness & Generalizability of NIRVANA)
> >  ### Thank you for the detailed response. ###
> > We recognize NIRVANA’s stability at high sparsity (50%) on Qwen models—a notable strength compared to LLM-Pruner’s catastrophic degradation. Two additional points would help further contextualize its robustness:
> > 1. The current comparisons on Qwen2.5-7B/14B focus on LLM-Pruner, SliceGPT and FLAP. Given that LLM-Pruner is relatively mature, would it be possible to extend these comparisons to more recent structured pruning methods? This would better position NIRVANA’s performance within the latest research landscape.
> > 2. We acknowledge NIRVANA’s strong performance on downstream tasks (e.g., HellaSwag) for Qwen2.5-7B at 40% sparsity, but we also notice a notable drop in language modeling performance—its WikiText perplexity (37.52) is ~11.5% higher than LLM-Pruner’s (33.64), which feels relatively significant. Since we’re not fully sure if this observation is comprehensive (e.g., potential nuances in metric interpretation we might miss), could you clarify why this perplexity drop occurs compared to LLM-Pruner, and whether it’s an unavoidable trade-off or can be mitigated (e.g., via tuning) while keeping downstream task and latency advantages?

---

> ### Author Response · Authors · 2025-11-21
>
> ### Q3. Formal Analysis of KL-based Selection
>
> We thank the reviewer for this question. We clarify that our selection strategy does not claim to find a theoretically guaranteed "global optimum" from infinite possibilities. Instead, it is a **standard Empirical Risk Minimization (ERM)** approach designed to identify the most robust candidate from a pool, avoiding worst-case scenarios common in random selection.
>
> 1. Theoretical Justification: Monte Carlo Approximation [1]
>
> The goal of pruning could be seen as to minimize the divergence between the dense model $P$ and pruned model $Q$ over the true data distribution $\mathcal{D}$: $\min\mathbb{E}_{x\sim\mathcal{D}} [KL(P(\cdot\|x)\|| Q(\cdot\|x))]$.
>
> Since the true distribution $\mathcal{D}$ is intractable, we use the anchor set $\mathcal{V}$ as a Monte Carlo estimator. The average KL divergence on $\mathcal{V}$ is an unbiased estimator of the true expected divergence:
>
> $$
> \mathcal{L}_{proxy}(\mathcal{V})
> $$
>
> $$
>  =\frac{1}{|\mathcal{V}|}  \sum_{x \in \mathcal{V}} KL(P(\cdot\|x) \| Q(\cdot\|x))
> $$
>
> $$
>  \approx\mathbb{E}_{x \sim \mathcal{D}}[KL(P \| Q)]
> $$
>
> By selecting the candidate $\mathcal{C}^*$ that minimizes this proxy loss $\mathcal{L}_{proxy}$, we are statistically maximizing the likelihood that the pruned model behaves like the dense model on the general distribution.
>
> 2. Empirical Evidence & Robustness
>
> We validate this estimator's effectiveness and stability in two ways:
> - Correlation (Figure 3): We show a strong correlation between our proxy (KL on $\mathcal{V}$) and final downstream performance (PPL), confirming that the estimator is effective.
> - Stability (New Robustness Check): To verify that our estimator is not sensitive to the specific sampling of $\mathcal{V}$, we used three distinct, non-overlapping anchor sets (V1, V2, V3) to rank 20 candidates.
>
>
> | Validation Set | Top-5 rank selected calibration IDs |
> | -------------- | ----------------------------------- |
> | V1 (seed 0)    | 12, 1, 17, 14, 11                   |
> | V2 (seed 1234) | 12, 1, 17, 14, 7                    |
> | V3 (seed 42)   | 12, 1, 17, 14, 8                    |
>
> As shown, the optimal choice (ID=12) was **unanimous**, and the **Top-4 candidates were identical** across all three distinct anchor sets. This confirms that a small sample $\mathcal{V}$ provides a **high-fidelity estimator** with negligible variance relative to the performance gaps, allowing NIRVANA to consistently and robustly identify the optimal candidate within the search space.
>
> [1] Better Estimation of the Kullback--Leibler Divergence Between Language Models, NeurIPS 2025.
>
>
> ## Happy to engage in further discussion!
>
> **Thank you again for the thoughtful review. With the valuable feedback from you and the other reviewers, we have put substantial efforts into obtaining new results and expanding our analysis. We hope our responses address your concerns, and we would be happy to discuss further if you have any additional questions.**

---

> > ### Comment · Reviewer_2ZLx · 2025-11-25
> > **Reviewer's Response**
> >
> > ## Follow-Up on Q3 (Formal Analysis of KL-based Selection)
> >  ### Thank you for the detailed response. ###
> > We sincerely appreciate your detailed and rigorous explanation of the KL-based calibration data selection strategy. Your theoretical justification via Monte Carlo approximation—framing the anchor set as an unbiased estimator for the true data distribution’s KL divergence—effectively clarifies why this method can stably identify robust calibration data. Additionally, the empirical evidence (e.g., strong correlation between KL divergence and downstream PPL, consistent top candidate selection across three distinct anchor sets) further validates the strategy’s reliability and stability.
> > Your response fully addresses our previous question, and we now have a clear understanding of how the method avoids worst-case scenarios in random selection while maintaining computational efficiency. We have no further follow-up questions on this aspect and admire the rigor of your theoretical and experimental support.

---

> ### Author Response · Authors · 2025-11-27
>
> ### Response to Q2
>
> We thank the reviewer for the deeper inquiry into robustness and metric trade-offs.
>
>
> 1. We agree that broadening the baseline landscape is valuable.
> - `Challenges with Qwen`: Adapting structured pruning methods from Llama to Qwen is **non-trivial**, particularly regarding the handling of **bias terms** in linear layers, which varies significantly across method implementations.
>     - `Structural Dependency`: Pruning a neuron requires ensuring the bias vector is sliced synchronously with the weight matrix to avoid shape mismatches.
>     - `Saliency Aggregation`: Crucially, the importance score of a structural unit (neuron) must aggregate contributions from both the weights and the bias. Most public implementations (designed for Llama) calculate saliency solely on weights (e.g., $S = |\nabla_W \cdot W|$). Applying this blindly to Qwen ignores the bias term's contribution to the output, leading to inaccurate importance ranking and potential removal of critical neurons.
> - `Added Baseline (FLAP)`: Despite this, we successfully adapted FLAP for Qwen2.5-7B and have added it to the comparison table below.
> - `Newer Methods (e.g., Olica)`: Regarding newer methods like Olica, we refer the reviewer to **[our response on W1 & Q1 to Reviewer 8GkX](https://openreview.net/forum?id=55K9BdoOlo&noteId=aJPTvY9VxE)**. Our evaluation showed that Olica suffers catastrophic collapse even at moderate sparsity (40%) on Llama3.1-8B. Therefore, LLM-Pruner and FLAP remain the strong and stable baselines for this comparison.
>
> |Model| WikiT↓| PTB↓ | LambD↓| ARC-e↑| WinoG↑| HellaS↑| MBPP↑|
> | --|-|-|-|-|- |-|-|
> | Qwen2.5-14B| 9.05 |15.64| 22.41| 80.47 |72.85 |78.86| 64.20     |
> | LLM-Pruner(0.2) | 17.73 |33.12| 34.17| 60.44| 54.93 |60.30 |12.40      |
> | FLAP(0.2) | **16.40** | **28.40**|32.94  | 61.28| 62.98| 57.77 | 4.40      |
> | NIRVANA(0.2) | 17.73 |29.22 |**32.10** |**68.98**| **65.69**| **69.24**| **32.60**|
> | LLM-Pruner(0.4) | 33.64 |**63.83** |48.18 |43.69| 53.91 |42.98 |0.00|
> | FLAP(0.4) | **32.97**| 66.12 | **46.07**| 47.25| 55.01| 42.39 | 0.00 |
> | NIRVANA(0.4)| 37.52| 72.33| 48.94| **51.26** | **58.33** | **54.22** | **5.00**  |
> | LLM-Pruner(0.5) | 8625.69| 11789.92 |8625.69| 24.79 |51.38 |25.73 |0.00 |
> | FLAP(0.5) | 3652.35| 15020.74 | 6460.34  | 26.60 | 48.15| 26.26 | 0.00|
> | NIRVANA(0.5)|**77.00**| **126.94** | **95.82** | **37.42** | **52.57** | **38.85** | 0.00      |
>
> 2. Regarding the PPL gap at 40% sparsity on WikiText: We respectfully emphasize that PPL and Task Accuracy measure **fundamentally** different capabilities.
> - `PPL (Fluency)`: PPL is an exponential metric ($\text{PPL} = e^{\text{Loss}}$) with an unbounded range $[1, \infty)$. It is highly sensitive to outliers: a few low-probability tokens can cause PPL to explode towards infinity without necessarily destroying the model's core logic. Furthermore, recent studies [1] indicate that PPL does not strictly correlate with complex reasoning capabilities: a model can be "fluent" (low PPL) but logically incoherent. Therefore, we regard Task Accuracy: which directly measures functional success, as the more decisive metric for utility.
> - `Accuracy (Reasoning)`: In contrast, accuracy in our downstream evaluations is defined as the **fraction of test instances solved correctly**, i.e., $\text{ACC}=\frac{\text{Num of (Corrected Ans)}}{\text{Num of (All Ans)}}$. Concretely:
>   - For multiple-choice benchmarks like **HellaSwag**, accuracy is the proportion of examples where the model selects the correct option.
>   - For **MBPP** (code generation), accuracy corresponds to the proportion of problems whose generated code passes all unit tests under a pass@1 evaluation. These metrics probe *reasoning*, which are **not linearly determined by small differences in language modeling PPL.**
> - `The Critical Insight`: While FLAP achieves marginally lower PPL (16.40 vs 17.73 at 20%), it fails dramatically on MBPP (Code Generation), scoring only 4.40 compared to NIRVANA's 32.60. This suggests that while baselines focus on surface-level confidence, NIRVANA preserves comparable fluency while uniquely retaining the "brain" (critical reasoning circuits). We argue that a model which can still write code but is slightly less confident (higher PPL) is **more valuable than a fluent model that confidently outputs nonsense.**
> - `Mitigation via Tuning`: The reviewer asks if the PPL gap can be mitigated. **Yes.** Our Qwen results were obtained with a rapid search of only 20 rounds (approx. 4 mins). Extending this to 50+ rounds (as we did for Llama3.1-8B) would naturally find a calibration set with lower PPL. However, given that NIRVANA **already dominates on reasoning tasks** and **is the only method to survive 50% sparsity**, we respectfully argue that the current results **already demonstrate superior robustness and utility, making further tuning optional rather than necessary.**
>
> [1] Same Pre-training Loss, Better Downstream: Implicit Bias Matters for Language Models, ICML 2023.

---

### Official Review · Reviewer_dy7K · 2025-10-29

**Soundness:** 3
**Presentation:** 4
**Contribution:** 2
**Rating:** 4
**Confidence:** 3

**Summary:**

The authors propose NIRVANA, a Neural Tangent Kernel (NTK)-guided structured pruning framework for large language models. To ensure pruning decisions align with model optimization dynamics, NIRVANA derives a first-order saliency criterion under Adam-based NTK analysis. It further introduces an adaptive sparsity allocation mechanism that dynamically distributes pruning ratios across layers and between Attention and MLP modules, and a KL divergence–based calibration data selection strategy to enhance pruning robustness and decouple pruning quality from calibration dataset size.

**Strengths:**

* The paper proposes a novel NTK-guided structured pruning framework that integrates adaptive sparsity allocation and KL-divergence-based calibration data selection.

* Extensive experiments on Llama3, Qwen, and T5 models show that NIRVANA achieves good performance.

**Weaknesses:**

* Idea of using NTK in pruning is not new, for example [NTK-SAP](https://openreview.net/pdf?id=-5EWhW_4qWP), authors should highlight the difference in the method section and add the comparison in ablation studies.
* Tables 12–17 present qualitative samples, but the descriptions are insufficient, and no corresponding objective quantitative metrics are provided, making it difficult to assess their quality and reducing the persuasiveness of the results.
* The authors derive an optimal $\gamma \approx 3.36$ theoretically and find the empirical optimum to be $3.0$ , showing close consistency, but this validation appears to be conducted only on a single model and lacks testing across multiple models.

**Questions:**

* As shown in Table 1, for MATH problems like **SVAMP** and code generation tasks like **MBPP**, performance drops sharply when sparsity reaches 40% (SVAMP: 72.67 → 11.67, MBPP: 48.60 → 2.20). How can this sudden drop in these tasks be explained, and does the improvement in throughput truly justify such a significant performance degradation?

---

> ### Author Response · Authors · 2025-11-21
>
> Dear Reviewer dy7K,
>
> We sincerely thank you for your constructive review and insightful questions. Your feedback has been extremely valuable in enhancing the clarity and quality of our work. We have conducted extensive analyses and provided detailed responses below, along with further evidence addressing each of your points.
>
> ### W1. Pruning Comparision with NTK-SAP
>
>
> > TL;DR: NTK-SAP is an **iterative, unstructured** method for **randomly initialized** networks, whereas NIRVANA is a **one-shot, structured** method for **pre-trained** LLMs. We demonstrate that NIRVANA outperforms the adapted NTK-SAP score in our ablation study.
>
>
> While both methods are inspired by NTK theory, they target fundamentally different problems: NTK-SAP is a foresight pruning method performed before training, whereas NIRVANA is a post-training pruning method for pre-trained LLMs.
>
> 1. Core Methodological Differences:
>
> | Feature    | NTK-SAP [1]                                | NIRVANA (Ours)               |
> | ---------- | ------------------------------------------ | ---------------------------- |
> | Paradigm   | Iterative, Unstructured Pruning            | One-shot, Structured Pruning |
> | Target     | Randomly initialized Networks, Pruning at Initialization              | Pre-trained LLMs, Post-training Pruning             |
> | Saliency   | Approximated sensitivity of NTK Trace Norm | First-order impact on Output |
> | Dependency | Data- and Weight-Agnostic                  | Data- and Weight-Dependent   |
>
> In detail,
>
> - Paradigm and Target: NTK-SAP is an **iterative, unstructured pruning** method that finds sparse subnetworks at initialization. In contrast, NIRVANA is a **one-shot, structured pruning** method designed for **post-training** compression of LLMs (removing entire heads/neurons).
> - Saliency Derivation: NTK-SAP aims to preserve the NTK spectrum; its saliency score is a **finite-difference approximation** of the NTK trace norm: $S(m_{ij}) = \left| \frac{\partial \mathbb{E}\left[ \lVert f(z; W \odot m) - f(z; (W + \Delta W) \odot m) \rVert_2^2 \right]}{\partial m_{ij}} \right|$, where $m$ represents the mask. NIRVANA's saliency score is derived from a first-order Taylor expansion to directly estimate the impact of weight removal on the model's output: $S(W_{ij})=|\frac{\partial f}{\partial W_{ij}}\cdot W_{ij}|$, where $W$ represents the weight. We then analyze this score under an Adam-style NTK to justify that it preserves NTK stability during optimization.
> - Data and Weight Dependency: NTK-SAP is **data- and weight-agnostic** (using random inputs $z$ and averaging over random initializations). NIRVANA is **data- and weight-dependent**, using real calibration data and specific pre-trained weights to compress a target model instance. This distinction is critical. Data- and weight-agnostic methods are blind to the specific knowledge encoded in pre-trained weights, do not leverage the specific structure of a given pre-trained checkpoint, which can make it overlook model-specific important directions. By conditioning on the actual weights, NIRVANA explicitly targets the preservation of these learned knowledges, whereas agnostic methods risk pruning essential capabilities.
>
> Due to the page limitation, we have added this detailed comparison in our revised Appendix.
>
> 2. Empirical Comparison (Ablation Study): We agree that NTK-SAP's saliency score is an important baseline. As its original foresight, data-agnostic paradigm is not directly applicable, we replace the random inputs with the same KL-selected calibration data, while keeping the other settings unchanged. We have also added this comparison to our revised Ablation Study.
>
>
> | Model            | WikiT ↓    | PTB  ↓     | LambD ↓    |
> | ---------------- | --------- | --------- | --------- |
> | NIRVANA          | **48.94** | **70.11** | **70.11** |
> | Magnitude-score  | 10e6      | 10e5      | 10e6      |
> | NTKSAP-score     | 58.12     | 72.33     | 72.33     |
> | Local Pruning    | 90.01     | 142.85    | 132.42    |
> | w/o $\gamma$     | 102.00    | 139.42    | 123.04    |
> | w/o KL-selection | 72.33     | 115.58    | 95.82     |
>
>
>
> As the table shows, NIRVANA's saliency score outperforms the adapted NTK-SAP score on all PPL benchmarks. This aligns with our findings on T5-base (see Appendix B.3), where NIRVANA also outperforms NTK-SAP on GLUE tasks, which is particularly notable: in that experiment, NTK-SAP was adapted as an unstructured pruning method, while NIRVANA was applied as a structured method. Since unstructured pruning typically sets an upper bound for performance, surpassing it with a hardware-friendly structured approach strongly validates the effectiveness of our saliency criterion.

---

> ### Author Response · Authors · 2025-11-21
>
> ### W2. Quantitative and Qualitatively Evaluation of Generated Samples
>
>
>
> > TL;DR: We evaluate the generated-samples using two NLP metrics (ROUGE, BERTScore) [1,2] and a standard 4-criteria framework (Fluency, Relevance, Coherence, Hallucination) [3,4], which quantitatively and qualitatively confirms NIRVANA's factual consistency.
>
> 1. We thank the reviewer for raising the concerns about the evaluation of our model-generated samples. We first provide the quantitative calculated ROUGE [1] and BERTScore [2] for all generated samples in Table 12-17.
>
> | Model          |  ROUGE    | BERTScore  |
> | -------------- |  ---------- | ---------- |
> | LLM-Pruner     |  0.2858     | 0.8586     |
> | FLAP           | 0.2672     | 0.8667     |
> | NIRVANA (Ours) | **0.3035** | **0.8732** |
>
> NIRVANA achieves both the **highest** ROUGE and BERTScore, demonstrating superior semantic alignment and content recall with the ground truth.
>
> 2. Furthermore, we adopted a qualitative standard human evaluation framework following established protocols in text generation evaluation [3,4]:
>    1. Fluency (F): The sentence-level grammaticality and naturalness of the generated text (local linguistic quality).
>    2. Relevance \(R): Whether the output directly addresses the question in the prompt and stays on topic.
>    3. Coherence \(C): Paragraph-level logical consistency and structural connectedness of the response.
>    4. Hallucination (H): Whether the output contains fabricated or semantically inconsistent information that is not grounded in the dense model's own distribution, such as inventing nonexistent entities, contradictory statements, or implausible claims.
>
> Using this framework, we show that NIRVANA demonstrates **superior factual consistency and significantly lower hallucination rates** in practical regimes. We have revised Appendix H to include these four qualitative metrics (F/R/C/H) for samples across all sparsity levels (20%, 40%, 50%, with and without recovery fine-tuning). We provide a full representative example at 20% sparsity from the fine-tuned model below, and refer the reviewer to the updated appendix for the full set of annotated samples.
>
> #### Prompt: Who lives in the Imperial Palace in Tokyo?
>
> | Model              | Generated Text                                               | F    | R    | C    | H    |
> | ------------------ | ------------------------------------------------------------ | ---- | ---- | ---- | ---- |
> | **Original**       | The Imperial Palace in Tokyo is the primary residence of the Emperor of Japan. The palace is located in the Chiyoda District of Tokyo. The current Emperor, Akihito, and his family reside in the palace. The palace is open to the public for tours, but only on certain days of the week. | Y    | Y    | Y    | N    |
> | **LLM-Pruner**     | The Emperor of Japan. The Emperor of Japan is the head of state and the sovereign of Japan. He is the symbol of the nation and is the head of the Imperial House of Japan. The Emperor lives in the Imperial Palace in Tokyo, which is the residence of the Emperor and his family. The Imperial Palace is located in Chiyoda, Tokyo and covers an area of 43 hectares. | Y    | Y    | Y    | Low  |
> | **FLAP**           | The Imperial Palace is located in Chiyoda-ku, Tokyo, and is the primary residence of the Emperor of Japan, also known as the Emperor of the Empire of Japan. It is surrounded by a large moat, and is protected by an outer wall. | Y    | Y    | Y    | Mid  |
> | **NIRVANA (Ours)** | The Emperor of Japan lives in the Imperial Palace in Tokyo. The Emperor is the head of the Imperial Family and is the symbol of the nation. The Emperor is a figurehead and has no powers, but is the highest ceremonial head of state. | Y    | Y    | Y    | N    |
>
> * Original model: fluent, relevant, coherent. The reference to “Akihito” reflects outdated pretraining knowledge rather than hallucination.
> * LLM-Pruner: fluent, relevant, coherent. The "43 hectares" detail differs from some external sources (115), but is still a plausible in-distribution factual variation, therefore counted as low hallucination.
> * FLAP: fluent, relevant, coherent, but contains an introduced inappropriate information: the Empire of Japan.
> * NIRVANA: fluent, relevant, coherent, and does not introduce fabricated content compared to the dense model.

---

> ### Author Response · Authors · 2025-11-21
>
> ### W2 (cont.)
>
> #### Prompt: Where did Yellowstone National Park get its name?
>
> | Model              | Generated Text                                               | F    | R           | C    | H    |
> | ------------------ | ------------------------------------------------------------ | ---- | ----------- | ---- | ---- |
> | **Original**       | Yellowstone is named after the Yellowstone River, which in turn was named by John Colter, a member of the Lewis and Clark Expedition. Colter was the first white man to see the area. He called it the "Yellow Stone River" because of the yellowish color of the rock in the river. Yellowstone is the oldest national park in the world and the first national park in the United States. | Y    | Y           | Y    | N    |
> | **LLM-Pruner**     | The answer is from the French fur trader, Robert Cenudin, who in 1805 explored the region and described it as a "champagne fountain." The name was adopted by Thomas Jefferson, who named the park after him. The park was created in 1872 as the world’s first national park, and it has been a popular tourist destination ever since. | Y    | Y           | Y    | high |
> | **FLAP**           | What is the most famous volcano in the park? Yellowstone National Park is named after the famous explorer, artist, and naturalist John C. Calhoun. He was the first person to visit Yellowstone and was the namesake of the park. The most famous volcano in the park is Mount Vesuvius, also known as "The Lava Mountain." | Y    | Partially Y | N    | mid  |
> | **NIRVANA (Ours)** | Yellowstone National Park is named after the name of the river that passes through it. The river is named after the Native American tribe, the Sioux, who called it "Yellowstone". | Y    | Y           | Y    | low  |
>
> * Original model: fluent, relevant, coherent. The mention of 'John Colter' reflects the dense model's own pretraining-era knowledge. Since our evaluation is based on relative consistency with the dense model (the ground-truth reference in a compression setup), this is not counted as hallucination.
> * LLM-Pruner: fluent, relevant, coherent. Robert Cenudin/Thomas Jefferson do not exist, so it contains high-level hallucination.
> * FLAP: fluent, partially relevant, not coherent (jumping from volcano to the name of the park), and contains introduced incorrect information, such as "Mount Vesuvius" being located in Yellowstone.
> * NIRVANA: fluent, relevant, coherent, contains introduced mild factual drift: the Sioux.
>
> [1] ROUGE: A Package for Automatic Evaluation of Summaries, ACL 2004.\
> [2] BERTScore: Evaluating Text Generation with BERT, ICLR 2020.\
> [3] SummEval: Re-evaluating Summarization Evaluation, EMNLP 2020.\
> [4] Survey of Hallucination in Natural Language Generation, ACM Comput. Surv 2023.
>
> ### W3. Evaluation of $\gamma$ Value across Multiple Models
>
>
>
> We thank the reviewer for this insightful question. The adaptive sparsity ratio, $\gamma$, is derived from a general analytic formula detailed in our Appendix F. This formula is broadly applicable, but the resulting numerical value of $\gamma$ is inherently dependent on the specific architectural hyperparameters of each model (e.g., hidden size, number of heads), and we explicitly **adjusted this parameter for each model**. As requested, we provide the corresponding $\gamma$ values used for the other models in our study, which were all derived from the same analytic formula by inputting each model's respective architectural parameters.
>
>
> | Model       | Gamma value |
> | ----------- | ----------- |
> | Llama3.1-8B | 3.36        |
> | Llama3.2-3B | 3.02        |
> | Qwen2.5-7B  | 10.11       |
>
> The larger $\gamma$ for Qwen2.5-7B is a direct result of its distinct structural design (e.g., a different ratio of attention heads to MLP dimensions) compared to the Llama architecture. Furthermore, as suggested by the empirical sweep in Figure 4, performance is relatively stable in the neighborhood of the analytically derived $\gamma$, which indicates that the method is robust to this parameter, and the strong, consistent performance of NIRVANA across all models confirms that these analytically derived values are effective in practice. We have updated this in our revised version. Thanks for pointing it out.

---

> ### Author Response · Authors · 2025-11-21
>
> ### Q1. Sharp Performance Drop for Math/Code Tasks
>
> > TL;DR: The sharp drop reflects the inherent fragility of the reasoning circuits activated in generative reasoning tasks (Math/Code), compared with those in discriminative multiple-choice tasks. However, this trade-off is justified: unlike baselines that collapse to absolute zero on MBPP, NIRVANA retains critical functional structure, serving as a superior initialization for recovery fine-tuning.
>
> We thank the reviewer for this keen observation. We address the explanation for the drop and the justification for the trade-off separately.
>
> 1. The sharper decline in SVAMP (Math) and MBPP (Code) compared to tasks like ARC-e or Winogrande is driven by **the fundamental difference in task nature**:
>     - Discriminative vs. Generative: Tasks like ARC-e/Winogrande are multiple-choice evaluations based on **comparing logits**. Even a degraded model often retains enough signal to rank the correct option higher than incorrect ones.
>     - In contrast, SVAMP and MBPP are generative reasoning tasks that require multi-step chain-of-thought or syntactically correct code generation. These capabilities rely on specific, deep "**reasoning circuits.**" If structured pruning disrupts even a single critical node in this circuit, the entire generation fails (e.g., a calculation error or syntax error), leading to a zero score.
>     - Non-linear Degradation: This aligns with the non-linear degradation trend shown in Figure 2. As sparsity increases, the probability of severing these critical reasoning circuits increases exponentially, disproportionately affecting these "hard" tasks.
> 2. Despite the drop, the throughput improvement is justified because NIRVANA uniquely retains non-trivial utility where others fail completely.
>     - As shown in Table 1, at 40% sparsity, baselines like SliceGPT and FLAP collapse to 0.00 on MBPP. In contrast, NIRVANA maintains functional accuracy (2.20), proving it **preserves more critical structures** than the baselines.
>     - A "Strong Starting Point": As detailed in our response to Reviewer 8GkX, the practical value of a 40% pruned model is to serve as a highly efficient initialization for further training. Our "w/tune" results confirm that NIRVANA's superior structural retention via NTK-stability allows it to recover these capabilities significantly better than baselines after fine-tuning, making the throughput gain accessible in practice.
>
> ## Happy to engage in further discussion!
>
> **Thank you again for the thoughtful review. With the valuable feedback from you and the other reviewers, we have put substantial efforts into obtaining new results and expanding our analysis. We hope our responses address your concerns, and we would be happy to discuss further if you have any additional questions.**

---

### Official Review · Reviewer_8GkX · 2025-10-31

**Soundness:** 2
**Presentation:** 3
**Contribution:** 2
**Rating:** 2
**Confidence:** 5

**Summary:**

This paper introduces NIRVANA, a structured pruning method for large language models that aims to balance zero-shot accuracy preservation with fine-tuning adaptability. The key contributions include: (1) a Neural Tangent Kernel (NTK)-guided saliency criterion that considers training dynamics under Adam optimization, (2) an adaptive sparsity allocation mechanism that differentially prunes MLP neurons vs. attention heads using a theoretically-derived ratio γ, and (3) a KL divergence-based calibration data selection strategy. The method is evaluated on Llama3, Qwen, and T5 models, showing improvements over existing structured pruning approaches like LLM-Pruner, SliceGPT, and FLAP across various sparsity levels (20-50%). But then I found the evaluation is limited to outdated baselines and smaller models, with significant performance degradation even at moderate sparsity levels.

**Strengths:**

The NTK-based saliency criterion provides a principled theoretical foundation connecting pruning decisions to training dynamics under Adam optimization, distinguishing it from purely heuristic methods.

The integration of three components (NTK-guided saliency, adaptive sparsity allocation, and KL-based calibration selection) forms a cohesive end-to-end solution that addresses multiple aspects of structured pruning.

The paper is well-written with clear motivation, consistent notation, and good visual aids (particularly Figure 1). The workflow from weight-level saliency to structured pruning is easy to follow.

The component-wise analysis effectively demonstrates the contribution of each proposed element, particularly the importance of the adaptive ratio γ and calibration data selection.

The practical insight about dimension alignment (multiples of 8) for GPU optimization provides valuable guidance for real-world deployment.

**Weaknesses:**

The experimental evaluation relies exclusively on methods from 2023-2024 (LLM-Pruner, FLAP, SliceGPT) while omitting numerous recent advances including SlimLLM (ICML 2025), Olica (ICML 2025), Týr-the-Pruner (NeurIPS 2025), LoRAPrune (ACL 2024), LLM-Barber (ICCAD 2025) and LoRAP (ICML 2024). This significantly undermines claims of advancing state-of-the-art.

Even when outperforming baselines, the method shows severe performance degradation at practical sparsity levels. At 20% sparsity, WikiText-2 perplexity increases from 8.5→13.4; at 50% sparsity it reaches ~49. Downstream tasks lose 20-40% accuracy and code generation collapses to near-zero, raising serious questions about real-world applicability.

Experiments are confined to relatively small models (≤8B parameters) with minimal calibration data (32 samples). No evaluation on larger models (70B+), MoE architectures, or comparison with modern quantization methods that achieve near-lossless compression.

While the NTK connection is interesting, the practical implementation remains similar to existing gradient-based importance metrics. The theoretical analysis (Proposition 4.1) relies on strong assumptions that may not hold in practice. Any analysis when it'll fail?

The ablation study focuses primarily on 50% sparsity where models already exhibit significant degradation, making it difficult to isolate component contributions. Missing analysis of pruning patterns across layers and computational overhead quantification.

Lacks verification of claimed NTK stability, distribution analysis of saliency scores across layers, and comparison of pruning decisions with simpler criteria like magnitude-based methods.

**Questions:**

See also weakness.

1. Given the rapid advancement in LLM compression, why were recent state-of-the-art methods (SlimLLM, Olica, Týr-the-Pruner, etc.) not included in the comparison? How would NIRVANA perform against these contemporary approaches?

2. Given the significant performance degradation observed even at 20% sparsity, what are the envisioned real-world applications? How does this compare to modern 8-bit/4-bit quantization methods that maintain near-lossless performance?

3. Have you evaluated NIRVANA on larger models (≥13B parameters) or MoE architectures? What is the computational overhead of NTK-guided scoring compared to simpler alternatives as model size increases?

4. Can you provide empirical evidence of NTK stability preservation (e.g., eigenvalue distribution, kernel alignment) before and after pruning to validate Proposition 4.1?

5. What is the quantitative computational cost of KL-based calibration selection? How do pruning patterns distribute across layers, and does the method successfully avoid layer collapse?

6. Could you provide ablation studies at moderate sparsity levels (20-30%) where models maintain reasonable functionality to better isolate individual component contributions?

7. How sensitive is performance to the adaptive ratio γ across different architectures? Does the theoretically derived value generalize beyond Llama-family models?

---

> ### Author Response · Authors · 2025-11-21
>
> Dear Reviewer 8GkX,
>
> We sincerely thank you for your constructive review and insightful questions. Your feedback has been extremely valuable in improving the clarity of our work. We have carefully considered your comments, conducted extensive analyses, and provide detailed responses below.
>
> ### W1 \& Q1. Comparison with other Pruning Works
>
> We thank the reviewer for listing these recent works and acknowledge the rapid progress in this field. To strengthen our evaluation as requested, we prioritized reproducing Olica [1], a recent and competitive structured-pruning approach with released code. The corresponding results are provided below:
>
> | Model         | WikiT ↓    | PTB ↓      | LambD  ↓   |
> | ------------- | --------- | --------- | --------- |
> | Llama3.1-8B   | 8.50      | 14.02     | 20.09     |
> | NIRVANA (0.2) | **13.38** | **19.77** | **26.20** |
> | Olica (0.2)   | 20.28     | 67.95     | 27.53     |
> | NIRVANA (0.4) | **28.33** | **38.72** | **43.19** |
> | Olica (0.4)   | 1978.00   | 7207.03   | 1307.38   |
>
> NIRVANA significantly outperforms Olica across all metrics. Notably, we observe a performance collapse for Olica at 40% sparsity. We verified our reproduction settings and attribute this to a potential structural mismatch with Llama3.1's **Grouped Query Attention** (GQA). Unlike Multi-Head Attention (MHA), GQA has significantly fewer KV heads with lower redundancy. Olica’s pruning metric likely over-prunes these critical KV heads at higher sparsity levels. In contrast, NIRVANA’s adaptive $\gamma$ ratio explicitly balances the head/neuron reduction, successfully preserving these sensitive components.
>
> Regarding the other suggested works, direct comparison was infeasible due to the following reasons:
> - SlimLLM [2] & Týr-the-Pruner [3]: Regrettably, neither method has released official code (or reproducible scripts) at the time of this rebuttal. Due to the limited time, implementing their methods from scratch is infeasible. Additionally, Týr-the-Pruner relies on an extremely expensive evolutionary search ($10^{76}$ space), taking **>10 hours** on Llama3.1-8B to be finished. In contrast, NIRVANA is a one-shot method that finishes in **seconds**. We have included both of the work in our related work section.
> - LLM-Barber [4]: This is an **unstructured/semi-structured** pruning method. As discussed in our response to Reviewer pqGd, it targets a different hardware trade-off and is not directly comparable to NIRVANA's structured acceleration.
> - LoRAP [5] and LoRAPrune [6]: These methods represent a distinct paradigm that tightly couples compression with Low-Rank Adaptation (LoRA) specifically for fine-tuning. In contrast, NIRVANA is a standalone, one-shot architecture compression method. It physically reduces the model backbone and is agnostic to the subsequent training strategy. Therefore, NIRVANA is **orthogonal** to these approaches: our pruned models can essentially serve as the base model to be further fine-tuned using LoRA or LoRAP-style techniques.
>
>
> [1] Olica: Efficient Structured Pruning of Large Language Models without Retraining, ICML 2025.\
> [2] SlimLLM: Accurate Structured Pruning for Large Language Models, ICML 2025.\
> [3] Týr-the-Pruner: Structural Pruning LLMs via Global Sparsity Distribution Optimization, NeurIPS 2025.\
> [4] LLM-Barber: Block-Aware Rebuilder for Sparsity Mask in One-Shot for Large Language Models,  ICCAD 2025.\
> [5] LoRAP: Transformer Sub-Layers Deserve Differentiated Structured Compression for Large Language Models, ICML 2024.\
> [6] LoRAPrune: Structured Pruning Meets Low-Rank Parameter-Efficient Fine-Tuning, ACL 2024 Findings.

---

> ### Author Response · Authors · 2025-11-21
>
> ### W2 & W3 \& Q2. The Role of Structured Pruning Beyond Quantization
>
> > TL;DR: Structured pruning offers unique, guaranteed hardware acceleration and reduced training costs that quantization cannot provide. While zero-shot drops are expected in this field, NIRVANA serves as a superior, instant initialization for training compact models, recovering performance efficiently where baselines fail.
>
> We thank the reviewer for these questions regarding performance trade-offs and practical application. Despite the higher zero-shot sensitivity compared to quantization, structured pruning remains vital for three irreplaceable practical advantages:
>
> 1. Unlike unstructured/semi-structured pruning (which requires specialized kernels) or quantization, structured pruning physically reduces matrix dimensions. This yields **guaranteed wall-clock speedups** (e.g., 0.33s $\to$ 0.21s, Table 3) and **higher throughput/memory savings** on any commodity hardware (CPU/GPU/TPU)
> 2. It is well-established in the literature that **one-shot structured pruning** (removing parameters without retraining) inevitably leads to significant zero-shot degradation, especially compared to unstructured pruning or quantization. This reflects the physical reality of structured pruning, not a defect unique to NIRVANA.
>    - Prior works like FLAP [1] exhibits similar drops before fine-tuning. Additionally, MINITRON [2] and Sheared LLaMA [3] state that pruning is primarily the means to an **efficient, compressed architecture**, which requires continued pre-training to recover capabilities.
>    - Given that NIRVANA can efficiently identifies a high-quality mask in **seconds**, our intended practical use is then to treat the pruned model as **a more efficient architecture for further training**, rather than as a purely one-shot drop-in replacement. The efficiency of NIRVANA's mask search allows most of the computational budget to be allocated to the recovery training stage, thereby accelerating the overall model production pipeline.
> 3. We have also explicitly compared pruning with quantization in Appendix C.
>    - Pruning and quantization are **orthogonal**. As shown in Appendix C (Table 10), NIRVANA combined with SmoothQuant (INT8) yields compounded gains with minimal loss, proving they are **compatible** and **orthogonal**.
>    - The "Sweet Spot": Quantization also faces a "performance cliff" at extreme bit-widths (e.g., <4-bit). A moderately pruned (e.g., 20%) and moderately quantized (INT8) model often hits a better efficiency-accuracy **"sweet spot"** than pushing either method to the limit.
>    - Crucially, quantization is primarily an *inference-time* optimization. It does not typically reduce the cost of fine-tuning or continued pre-training (FLOPs), while NIRVANA reduces the FLOPs/throughput/memory required for **both forward and backward passes**. This accelerates the entire training lifecycle: fine-tuning, domain adaptation, or RLHF, offering a benefit that post-training quantization cannot provide.
>
> [1] Fluctuation-based Adaptive Structured Pruning for Large Language Models, AAAI 2024.\
> [2] Compact Language Models via Pruning and Knowledge Distillation, NeurIPS 2024.\
> [3] Sheared LLaMA: Accelerating Language Model Pre-training via Structured Pruning, ICLR 2024.\
> [4] Týr-the-Pruner: Structural Pruning LLMs via Global Sparsity Distribution Optimization, NeurIPS 2025.

---

> ### Author Response · Authors · 2025-11-21
>
> ### W3 & W5 & Q3. Evaluation on Models of Larger Scales
>
> We thank the reviewer for the suggestion to evaluate larger scales. As requested, we evaluated NIRVANA on **Qwen2.5-14B** across three sparsity levels. We prioritized scaling up dense models, as extending NIRVANA to MoE architectures involves handling expert-routing dynamics, which we have left for future work. The results (see Table below) strongly validate that NIRVANA generalizes effectively to larger parameters. We have added this experiment into our revised paper.
>
> | Model|WikiT↓|PTB↓|LambD↓|ARC-e↑|WinoG↑|HellaS↑|MBPP↑|
> | --|--|--|--|--|--|--|--|
> | Qwen2.5-14B| 7.16| 13.38 | 20.09| 79.34| 75.69 | 82.91| 69.00 |
> | LLM-Pruner(0.2)| 12.18| 21.05 | **25.39**  | 70.41  | 66.46| **73.15** | 8.20      |
> | NIRVANA(0.2) | **11.99** | **19.18** | 26.20  | **73.40** | **68.19** | 72.03     | **41.60** |
> | LLM-Pruner(0.4) | 119.25  | 345.07    | 334.45     | 39.77| 51.54| 34.41     | 0.00  |
> | NIRVANA(0.4)| **31.11** | **48.18** | **53.75**  | **45.88** | **55.25** | **47.47** | **0.20**  |
> | LLM-Pruner(0.5) | 3827.63   | 6310.69   | 5231.74 | 26.56   | 48.93| 26.08| 0.00  |
> | NIRVANA(0.5)| **59.96** | **84.56** | **105.24** | **37.42** | **52.57** | **38.85** | 0.00 |
>
> Regarding cost, our NTK-guided saliency is computationally equivalent to standard gradient-based metrics. NIRVANA requires only **a forward and backward pass** on a small calibration batch to compute the gradients. While simple magnitude-based pruning avoids this step, our ablation study (Table 2) shows it leads to catastrophic failure. Given the minimal cost of whole process (approx. **2 seconds** for Llama-8B on A100, detailed in our Rebuttal to Reviewer pqGd), we believe this tiny overhead is fully justified by the substantial performance gains.
>
>
> ### W4 & W6 & Q4. NTK Analysis and Realistic Assumption
>
>
> We thank the reviewer for the deep insights into the NTK analysis and impact in NIRVANA. We acknowledge that in Proposition 4.1 (also as pointed out by Reviewer pqGd), our analysis is under the Adam optimizer. Hence, the training dynamics might be different if the model is trained using other optimizers. However, given that Adam is the de facto standard for modern LLM training, we argue that this assumption is actually a **necessary and reasonable adaptation**. It ensures NIRVANA is aligned with the actual optimization landscape of LLMs, unlike generic methods based on standard SGD-NTK theory.
>
> Additionally, to empirically support Proposition 4.1, we conducted the requested experiments and provide the results before and after pruning 50% of Llama3.1-8B in the following table. Note that calculating the full eigenvalue distribution on an 8B model would be computationally prohibitive; hence, we utilized Centered Kernel Alignment [1].
>
>
> | Method     | Centered Kernel Alignment |
> | ---------- | ------------------------- |
> | Magnitude  | 0.9658                    |
> | LLM-Pruner | 0.9722                    |
> | FLAP       | 0.9924                    |
> | NIRVANA    | **0.9972**                |
>
> The results demonstrate that NIRVANA achieves the highest NTK alignment among all baselines. This provides empirical evidence that our NTK-driven saliency score effectively preserves the NTK structure during pruning, which is consistent with the state-of-the-art recovery fine-tuning performance shown in Table 1. We have added these results into our revision.
>
> [1] Similarity of Neural Network Representations Revisited, ICML 2019.

---

> ### Author Response · Authors · 2025-11-21
>
> ### W5 & W6 & Q5. Pruning Patterns across Layers
>
>
> We thank the reviewer for these detailed questions regarding the structural properties of our pruned models.
>
> For layer collapse, as mentioned in Lines 280-281, we implemented a **hard constraint** that explicitly retains at least one unit per structural group. This guarantees that no layer is ever fully severed, preserving gradient flow.
>
> To visualize the pruning patterns, we provide a layer-wise density plot for the Llama-3.1-8B model at 50% sparsity in [this anonymous link](https://anonymous.4open.science/api/repo/NIRVANA_rebuttal-58A0/file/layer_dist.png?v=d8ba81dc). The plot reveals a non-uniform strategy where NIRVANA preserves the critical input/output layers while pruning the redundant middle-to-deep layers, which aligns with recent findings [1].
>
> As for saliency patterns, the [magnitude-based distribution](https://anonymous.4open.science/api/repo/NIRVANA_rebuttal-58A0/file/saliency_layer_mag.png?v=f7031d35) shows both attention and MLP scores increasing almost monotonically with depth, treating deeper layers as  more important. In contrast, the [NIRVANA distribution](https://anonymous.4open.science/api/repo/NIRVANA_rebuttal-58A0/file/saliency_layer_nirvana.png?v=96b35ba6) exhibits a non-monotonic, more flexible depth pattern: it assigns relatively high saliency to the input/output layers while selectively up- or down-weighting different middle blocks, suggesting that the NTK-guided criterion adaptively targets redundant regions.
>
> [1] ShortGPT: Layers in Large Language Models are More Redundant Than You Expect, ACL 2025 Findings.
>
>
> ### Q5. Computational costs of KL-based Selection
>
> We appreciate the reviewer's concern regarding the computational overhead. To address this directly, we measured the actual wall-clock time for the KL-based selection process on a single A100 GPU. As shown below, the cost is **minimal**:
>
> | Model       | Wallclock time |
> | ----------- | -------------- |
> | Llama3.1-8B | 4.12s          |
> | Qwen2.5-14B | 6.28s          |
>
>
> 1. Efficiency by Design: This low latency is achieved because we use **a small size of calibration sets**. As discussed in Appendix G, neither the number of examples nor sequence length dominates pruning quality. Therefore, we fix the candidate calibration set to 32 samples (seq len 128) and the evaluation "anchor set" to 128 samples (seq len 64).
> 2. Total Computational Overhead: The process consists of two steps:
>    - Pre-computation (Once): Computing dense model logits on the anchor set takes 3.04s for Llama3.1-8B and 3.93s for Qwen2.5-14B on a single A100. This is **cached and reused**.
>    - Selection Loop (Per Candidate): For each candidate, computing the pruned model's logits and the KL divergence takes 4.12s (Llama3.1-8B) and 6.28s (Qwen2.5-14B).
>
> Combined with the fast saliency computation (approx. 2s), a complete search over 50 candidate sets for Llama3.1-8B takes **less than 10 minutes** on a single A100. This minimal one-time cost yields significant performance gains, making it a highly efficient trade-off. We will add this into our revised paper to help better understand the context.

---

> ### Author Response · Authors · 2025-11-21
>
> ### W5 & Q6. Ablation study at Moderate Sparsity (20%)
>
>
>
> We thank the reviewer for this suggestion. Our original choice to focus the ablation study on 50% sparsity was intentional: at such a high compression regime, the model is under maximum stress, making it the most discriminative setting to isolate the contributions of specific components (NTK-saliency, $\gamma$, KL). However, we agree that analyzing a moderate sparsity regime is valuable for completeness. As requested, we conducted the full ablation study at 20% sparsity on Llama3.1-8B:
>
>
> | Model            | WikiT ↓    | PTB  ↓     | LambD   ↓  |
> | ---------------- | --------- | --------- | --------- |
> | NIRVANA          | **13.38** | **19.77** | **26.20** |
> | Magnitude-score  | 5x10e4    | 6x10e4    | 6x10e4    |
> | Local Pruning    | 16.91     | 25.00     | 32.10     |
> | w/o $\gamma$     | 14.24     | 21.38     | 27.45     |
> | w/o KL-selection | **13.38** | 20.09     | 26.61     |
>
> Similar to the 50% regime, this ablation study confirms that each component contributes to NIRVANA's superiority performance, although the gaps are naturally narrower than at 50%.
>
> ### W7. Sensitivity of $\gamma$ value
>
>
>
> We thank the reviewer for this insightful question. The adaptive sparsity ratio, $\gamma$, is derived from a general analytic formula detailed in our Appendix F. This formula is broadly applicable, but the resulting numerical value of $\gamma$ is inherently dependent on the specific architectural hyperparameters of each model (e.g., hidden size, number of heads, MLP intermediate dimension). While our main experiment on Llama3.1-8B yielded $\gamma \approx 3.36$ (which was empirically validated in Figure 4), this value is **not universal**. As requested, we provide the corresponding $\gamma$ values used for the other models in our study, which were all derived from the same analytic formula by inputting each model's respective architectural parameters.
>
>
> | Model       | Gamma value |
> | ----------- | ----------- |
> | Llama3.1-8B | 3.36        |
> | Llama3.2-3B | 3.02        |
> | Qwen2.5-7B  | 10.11       |
>
> The larger $\gamma$ for Qwen2.5-7B is a direct result of its distinct structural design (e.g., a different ratio of attention heads to MLP dimensions) compared to the Llama architecture. Furthermore, as suggested by the empirical sweep in Figure 4, performance is relatively stable in the neighborhood of the analytically derived $\gamma$. This indicates that the method is robust to this parameter, and the strong, consistent performance of NIRVANA across all models confirms that these analytically derived values are effective in practice. We have updated this in our revised version. Thanks for pointing it out.
>
> ## Happy to engage in further discussion!
>
> **Thank you again for the thoughtful review. With the valuable feedback from you and the other reviewers, we have put substantial efforts into obtaining new results and expanding our analysis. We hope our responses address your concerns, and we would be happy to discuss further if you have any additional questions.**

---

> ### Comment · Reviewer_8GkX · 2025-11-25
> **Nice rebuttal, score raised to 4.**
>
> While the authors have addressed most of my technical concerns and provided valuable insights (especially hardware-awareness), the core practical limitation of significant performance degradation even at moderate sparsity remains. The work makes meaningful contributions to structured pruning theory and practice, but falls short of the breakthrough needed for strong acceptance.
>
> The rebuttal quality itself deserves recognition! It's thorough, well-evidenced, and demonstrates deep engagement with my feedback. This pushes the work from "reject" toward the acceptance threshold, though practical concerns prevent a strong endorsement.

---

> > ### Author Response · Authors · 2025-11-25
> >
> > Dear Reviewer 8GkX,
> >
> > We sincerely thank you for the thoughtful follow-up, for carefully reading our rebuttal and additional experiments, and for raising your score. We greatly appreciate your recognition of our contributions on hardware-awareness and NTK-guided structured pruning, as well as your acknowledgement that most of your earlier concerns have been addressed.
> >
> > Regarding the remaining concern on practical limitations, we agree that performance degradation at high sparsity is the core challenge of one-shot structured pruning in general. Our goal is to make structured pruning a *useful starting point* rather than the final step: NIRVANA provides a strong, hardware-aware compact architecture that can be further trained, and our experiments across multiple downstream tasks (e.g., ARC-e, MBPP) show that pruned models can still retain non-trivial capabilities and recover well under fine-tuning. We believe this demonstrates both the practical value of NIRVANA and the broader relevance of structured pruning for building efficient, deployable LLMs.
> >
> > We deeply appreciate your assessment that this work makes "meaningful contributions to structured pruning theory and practice" and "pushes the work toward the acceptance threshold." Together with the positive aspects highlighted by the other reviewers, we hope this clarifies that NIRVANA offers a solid and practically relevant step forward for hardware-aware structured pruning of LLMs.
> >
> > Best,  \
> > #11904 Authors

---

### Author Response · Authors · 2025-12-01
**Author Summary for Submission #11904**

Dear Area Chairs & Reviewers,

We sincerely thank the AC for handling this paper under the current ICLR situation and for the reviewers' feedback to help us improve the quality of the paper. We believe that most of the initial concerns stemmed from misunderstandings, and we have addressed these points through substantial additional experiments and revisions. Below is a summary of the paper's contents and the changes made during the discussion period.

---

### 1. Paper in one paragraph

We propose **NIRVANA**, a NTK-guided **structured pruning** framework for LLMs. Our main contributions are:
- An theoretically ground **Adam-aware NTK** saliency score.
- An **adaptive Attention–MLP sparsity ratio ($\gamma$)** derived analytically for each architecture.
- A **KL-based calibration data selection** strategy with both theoretical motivation and empirical validation.

We validate NIRVANA on Llama3 (8B & 3B), Qwen2.5 (7B & 14B), and T5-base, showing improved performance over baselines in both zero-shot and fine-tuning settings at comparable sparsity and much better stability at 40-50% sparsity.

---

### 2. Main issues raised → what we did

- **Robustness on Qwen & newer baselines & generalizability** (Reviewer 8GkX/2ZLx)
  We extended experiments to **Qwen2.5-14B** and adapted **FLAP** to Qwen. NIRVANA dominates on downstream tasks (especially code-generation task MBPP) and is the only method that remains usable at 50% sparsity. We also evaluated **LLM-Olica** on Llama3.1-8B and observed collapse at 40% sparsity, likely due to incompatibility with Llama3's GQA.

- **KL-based calibration selection** (Reviewer 2ZLx/pqGd)
  We clarified the KL-based selection motivation, showed a strong empirical correlation between KL and PPL, and added a robustness test with three disjoint anchor sets, all selecting the same best calibration candidate. Reviewer 2ZLx explicitly wrote that this fully addresses their concern.

- **Latency "anomalies" & hardware-awareness** (Reviewer 2ZLx)
  We ran a hidden-dimension sweep (odd / even / multiples of 2,4,8,16,32) and showed misaligned dimensions can be slower than dense, while all 8-aligned dimensions recover the expected speedups. We then added **dimension alignment to LLM-Pruner and FLAP** and re-ran efficiency: after alignment, all methods show consistent speedups, confirming that the original anomalies were due to misalignment and highlighting the importance of NIRVANA’s hardware-aware design.

- **Qualitative generations** (Reviewer dy7K)
  We added NLP metrics (ROUGE/BERTScore/Fluency/Relevance/Coherence/Hallucination) and concrete examples, showing NIRVANA preserves coherence and factuality better than baselines as sparsity increases.

- **NTK-SAP & comparision to unstructured/semi-structured pruning** (Reviewer dy7K/pqGd)
  We added a direct comparison to an NTK-SAP-style score (adapted to our setting). We showed that NIRVANA’s saliency consistently performs better on Llama3.1-8B and T5-base, even when NTK-SAP is *unstructured* while NIRVANA is structured. We also compare NIRVANA (structured pruning) to unstructured/semi-structured pruning and show the trade-off.

---

### 3. Current status (based on discussion)

- Reviewer 8GkX **raised the score from 2 → 4**, stating that the work now makes **"meaningful contributions to structured pruning theory and practice" and is "pushed toward the acceptance threshold"**.
- Reviewer 2ZLx explicitly states that the KL-based selection concern is **fully resolved** and praises the rigor of the analysis.
- The remaining reviewer focused on clarification-oriented points (e.g., $\gamma$, qualitative evaluation), which are directly addressed by the added experiments and clarifications above.

We hope this helps the AC see the post-discussion picture: most technical concerns have been resolved, and the paper now offers a principled, hardware-aware structured pruning framework with solid empirical support across multiple LLM families.

---

### Meta-Review · Area_Chair_vNTg · 2026-01-06

**Summary:**

This paper proposes a structured pruning framework for Large Language Models (LLMs) named NIRVANA that integrates an NTK-guided saliency criterion based on Adam optimization dynamics, an adaptive sparsity allocation mechanism, and a KL divergence-based calibration data selection strategy.

According to the author-reviewer discussion, this is a borderline paper with 4, 4, 6, 6. On one hand, the most negative reviewer 8GkX raises from 2 to 4, however, on the other hand, there still exist outstanding issues, such as the method still exhibiting significant performance degradation even at moderate sparsity levels.

The AC is leaning toward rejecting this paper and encourage the authors in the future version that: 1) focus on overcoming performance retention issues in inference and code generation tasks at high sparsity (e.g., 40-50%); 2) provide more rigorous experiments to demonstrate the significant advantages of using the "pruned model as a starting point for fine-tuning" compared to other methods with pruning-then-finetuning strategies; 3) consider validating the robustness of the method on a broader range of model families (e.g., models with MoE architectures).

**Reviewer Concerns:**

**Addressed Comments**:

1. **Inference Latency Anomalies (Reviewer 2ZLx)**: The authors successfully explained why LLM-Pruner baselines with fewer parameters were slower than dense models by analyzing hardware-aware dimension alignment, and demonstrated that NIRVANA solves this by enforcing 8x alignment.

2. **Missing SOTA Baselines (Reviewer 8GkX)**: The authors added comparisons with newer methods like Olica and showed that NIRVANA avoids performance collapse at 40% sparsity.

3. **Overhead of KL Selection Strategy (Reviewer 2ZLx, 8GkX)**: The authors demonstrated that the process takes very little time on a single A100 GPU, representing an efficient trade-off.

4. **Qualitative Evaluation of Generated Samples (Reviewer dy7K)**: The authors added quantitative metrics (ROUGE/BERTScore) and a evaluation framework covering dimensions including Fluency, Relevance, Coherence, and Hallucination.

5. **Distinction from NTK-SAP (Reviewer dy7K)**: The authors clarified the fundamental differences between NIRVANA (One-shot, structured, for pre-trained LLMs) and NTK-SAP (Iterative, unstructured, for randomly initialized networks).

**Outstanding Comments**:

1. **Practicality and Performance Collapse at High Sparsity (Reviewers 8GkX, dy7K)**: Reviewer 8GkX emphasized that NIRVANA's own absolute performance loss remains very severe. Also as pointed out by Reviewer dy7K, on difficult tasks like code generation (MBPP) and mathematical reasoning (SVAMP), the model's capabilities are almost lost.

2. **Insufficient Argument as a Fine-tuning Starting Point (Reviewer 8GkX)**: The authors argued that the pruned model should be viewed as an efficient starting point (Initialization) for further fine-tuning, but without comparison to other methods using pruning-then-finetuning strategies.

**Reviewer Scores:**

- Reviewer 8GkX: 2->4

- Reviewer 2ZLx: 6

- Reviewer pqGd: 6

- Reviewer dy7K: 4

---

### Decision · Program_Chairs · 2026-01-26

Reject